# Sample complexity of data-driven tuning model hyperparameters in neural networks with piecewise polynomial dual functions

## Abstract

Modern machine learning algorithms, especially deep learning-based techniques, typically involve careful hyperparameter tuning to achieve the best performance. Despite the surge of intense interest in practical techniques like Bayesian optimization and random search-based approaches to automating this laborious and compute-intensive task, the fundamental learning-theoretic complexity of tuning hyperparameters for deep neural networks is poorly understood. Inspired by this glaring gap, we initiate the formal study of hyperparameter tuning complexity in deep learning through a recently introduced data-driven setting. We assume that we have a series of deep learning tasks, and we have to tune hyperparameters to do well on average over the distribution of tasks. A major difficulty is that the utility function as a function of the hyperparameter is very volatile and furthermore, it is given implicitly by an optimization problem over the model parameters. This is unlike previous work in data-driven design, where one can typically explicitly model the algorithmic behavior as a function of the hyperparameters. To tackle this challenge, we introduce a new technique to characterize the discontinuities and oscillations of the utility function on any fixed problem instance as we vary the hyperparameter; our analysis relies on subtle concepts including tools from differential/algebraic geometry and constrained optimization. This can be used to show that the learning-theoretic complexity of the corresponding family of utility functions is bounded. We instantiate our results and provide the sample complexity bounds for concrete applications—tuning a hyperparameter that interpolates neural activation functions and setting the kernel parameter in graph neural networks.

## 1 Introduction

Developing deep neural networks that work best for a given application typically corresponds to a tedious selection of hyperparameters and architectures over extremely large search spaces. This process of adapting a deep learning algorithm or model to a new application domain takes up significant engineering and research resources, and often involves unprincipled techniques with limited or no theoretical guarantees on the effectiveness. While the success of pre-trained (foundation) models have shown the usefulness of transferring effective parameters (weights) of learned deep models across tasks (Devlin, 2018; Achiam et al., 2023), it is less clear how to leverage prior experience of "good" hyperparameters to new tasks. In this work, we develop a principled framework for tuning continuous hyperparameters in deep networks by leveraging similar problem instances and obtain sample complexity guarantees for learning provably good hyperparameter values.

The vast majority of practitioners still use a naive "grid search" based approach which involves selecting a finite grid of (often continuous-valued) hyperparameters and selecting the one that performs the best. A lot of recent literature has been devoted to automating and improving this hyperparameter tuning process, prominent techniques include Bayesian optimization (Hutter et al., 2011; Bergstra et al., 2011; Snoek et al., 2012; 2015) and random search based methods (Bergstra & Bengio, 2012; Li et al., 2018). While these approaches work well in practice, they either lack a formal basis or enjoy limited theoretical guarantees only under strong assumptions. For example, Bayesian optimization assumes that the performance of the deep network as a function of the hyperparameter can

be approximated as a noisy evaluation of an expensive function, typically making assumptions on the form of this noise, and requires setting several hyperparameters and other design choices including the amount of noise, the acquisition function which determines the hyperparameter search space, the type of kernel and its bandwidth parameter. Other techniques, including random search methods and spectral approaches (Hazan et al., 2018) make fewer assumptions but only work for a discrete and finite grid of hyperparameters.

We approach the problem of hyperparameter tuning in deep networks using the lens of data-driven algorithm design, initially introduced in the context of theory of computing for algorithm configuration (Gupta & Roughgarden, 2016; Balcan, 2020). A key idea is to treat a parameterized family of algorithms as the hypothesis space and input instances to the algorithm as the data, reducing hyperparameter tuning to a learning problem. While the approach has been successfully applied to tune fundamental machine learning algorithms including clustering (Balcan et al., 2018b; 2019), semi-supervised learning (Balcan & Sharma, 2021), low-rank approximation (Bartlett et al., 2022), regularized linear regression (Balcan et al., 2022a; 2024a), decision tree learning (Balcan & Sharma, 2024), among others, our work is the only one to focus on analyzing deep network hyperparameter tuning under this data-driven paradigm. A key technical challenge that we overcome is that varying the hyperparameter even slightly can lead to a significantly different learned deep network (even for the same training set) with completely different parameters (weights) which is hard to characterize directly. This is very different from a typical data-driven method where one is able to show closed forms or precise structural properties for the variation of the learning algorithm's behavior as a function of the hyperparameter (Balcan et al., 2021a). We elaborate further on our technical novelties in Section 1.1. We note that our theoretical advances are potentially useful beyond deep networks, to algorithms with a tunable hyperparameter and several learned parameters.

We instantiate our novel framework for hyperparameter tuning in deep networks in some fundamental deep learning techniques with active research interest. Our first application is to tuning an interpolation hyperparameter for the activation function used at each node of the neural network. Different activation functions perform well on different datasets (Ramachandran et al., 2017; Liu et al., 2019). We analyze the sample complexity of tuning the best combination from a pair of activation functions by learning a real-valued hyperparameter that interpolates between them. We tune the hyperparameter across multiple problem instances, an important setting for multi-task learning. Our contribution is related to neural architecture search (NAS). NAS (Zoph & Le, 2017; Pham et al., 2018; Liu et al., 2018) automates the discovery and optimization of neural network architectures, replacing human-led design with computational methods. Several techniques have been proposed (Bergstra et al., 2013; Baker et al., 2017; White et al., 2021), but they lack principled theoretical guarantees (see additional related work in Appendix A), and multi-task learning is a known open research direction (Elsken et al., 2019). We also instantiate our framework for tuning the graph kernel parameter in Graph Neural Networks (GNNs) (Kipf & Welling, 2017) designed for more effectively deep learning with structured data. Hyperparameter tuning for graph kernels has been studied in the context of classical models (Balcan & Sharma, 2021; Sharma & Jones, 2023), in this work we provide the first provable guarantees for tuning the graph hyperparameter for the more effective modern approach of graph neural networks.

**Our contributions.** In this work, we provide an analysis for the learnability of parameterized algorithms involving both parameters and hyperparameters in the data-driven setting, which captures model hyperparameter tuning in deep networks with piecewise polynomial dual functions. A key ingredient of our approach is to show that the dual utility function $u_{\boldsymbol{x}}^*(\alpha)$, measuring the performance of the deep network on a fixed dataset $\boldsymbol{x}$ and when the parameters are trained to optimality using hyperparameter $\alpha$, admits a specific piecewise structure. We show that in many cases of interest, the dual utility function $u_{\boldsymbol{x}}^*$ is piecewise polynomial, and we bound the number of discontinuities and number of local maxima within each piece. Concretely,

- We introduce tools of independent interest, connecting the discontinuities and local maxima of a piecewise continuous function with its learning-theoretic complexity (Lemma 3.1, Lemma 3.2).

- We demonstrate that when the function $f_{\boldsymbol{x}}(\alpha, \boldsymbol{w})$ computed by a deep network is *piecewise constant* over at most $N$ *connected components* in the space $\mathcal{A} \times \mathcal{W}$ of hyperparameter $\alpha$ and parameters $\boldsymbol{w}$, the function $u_{\boldsymbol{x}}^*$ is also piecewise constant. This structure occurs in classification tasks

with a 0-1 loss objective. Using our proposed tools, we then establish an upper-bound for the pseudo-dimension of $\mathcal{U}$, which automatically translate to learning guarantee for $\mathcal{U}$ (Theorem 4.2).

- We further prove that when the function $f_{\boldsymbol{x}}(\alpha, \boldsymbol{w})$ exhibits a *piecewise polynomial* structure, under mild regularity assumptions, we can establish an upper bound for the number of discontinuities and local extrema of the dual utility function $u_{\boldsymbol{x}}^*$. The core technical component is to use ideas from algebraic geometry to give an upper-bound for the number of local extrema of parameter $\boldsymbol{w}$ for each value of the hyperparameter $\alpha$ and use tools from differential geometry to identify the smooth 1-manifolds on which the local extrema $(\alpha, \boldsymbol{w})$ lie. We then use our proposed result (Lemma 3.2) to translate the structure of $u_{\boldsymbol{x}}^*$ to learning guarantee for $\mathcal{U}$ (Theorem 5.1).

- We examine data-driven algorithm configuration for deep networks, focusing on hyperparameter tuning in semi-supervised GCNs (Theorem 6.2) and activation function learning in NAS (Theorem 6.1). Analysis of their dual utility functions reveals piecewise structures that, under our framework, establish the learnability of hyperparameters for both classification and regression tasks.

## 1.1 TECHNICAL CHALLENGES AND INSIGHTS

To analyze the pseudo-dimension of the utility function class $\mathcal{U}$, by using our proposed results (Theorem 3.1), the key challenge is to establish the relevant piecewise structure of the dual utility function class $u_{\boldsymbol{x}}^*$. Different from typical problems studied in data-driven algorithm design, $u_{\boldsymbol{x}}^*$ in our case is not an explicit function of the hyperparameter $\alpha$, but defined implicitly via an optimization problem over the network weights $\boldsymbol{w}$, i.e. $u_{\boldsymbol{x}}^*(\alpha) = \max_{\boldsymbol{w} \in \mathcal{W}} f_{\boldsymbol{x}}(\alpha, \boldsymbol{w})$. In the case where $f_{\boldsymbol{x}}(\alpha, \boldsymbol{w})$ is piecewise constant, we can partition the hyperparameter space $\mathcal{A}$ into multiple segments, over which the set of connected components for any fixed value of the hyperparameter remains unchanged. Thus, the behavior on a fixed instance as a function of the hyperparameter $\alpha$ is also piecewise constant and pseudo-dimension bounds follow. It is worth noting that $u_{\boldsymbol{x}}^*$ cannot be viewed as a simple projection of $f_{\boldsymbol{x}}$ onto the hyperparameter space $\mathcal{A}$, making it challenging to determine the relevant structural properties of $u_{\boldsymbol{x}}^*$.

For the case $f_{\boldsymbol{x}}(\alpha, \boldsymbol{w})$ is piecewise polynomial, the structure is significantly more complicated and we do not obtain a clean functional form for the dual utility function class $u_{\boldsymbol{x}}^*$. We first simplify the problem to focus on individual pieces, and analyze the behavior of $u_{\boldsymbol{x},i}^*(\alpha) = \sup_{\boldsymbol{w}:(\alpha,\boldsymbol{w}) \in R_{\boldsymbol{x},i}} f_{\boldsymbol{x},i}(\alpha, \boldsymbol{w})$ in the region $R_i$ where $f_{\boldsymbol{x}}(\alpha, \boldsymbol{w}) = f_{\boldsymbol{x},i}(\alpha, \boldsymbol{w})$ is a polynomial. We then employ ideas from algebraic geometry to give an upper-bound for the number of local extrema $\boldsymbol{w}$ for each $\alpha$ and use tools from differential geometry to identify the *smooth 1-manifolds* on which the local extrema $(\alpha, \boldsymbol{w})$ lie. We then decompose such manifolds into *monotonic-curves*, which have the property that they intersect at most once with any fixed-hyperparameter hyperplane $\alpha = \alpha_0$. Using these observations, we can finally partition $\mathcal{A}$ into intervals, over which $u_{\boldsymbol{x},i}^*$ can be expressed as a maximum of multiple continuous functions for each of which we have upper bounds on the number of local extrema. Putting together, we are able to leverage a result from Balcan et al. (2021a) to bound the pseudo-dimension.

**Paper positioning.** Our setting requires technical novelty compared to prior work in statistical data-driven algorithm hyperparameter tuning (Balcan et al., 2017; 2020a;b; 2021b;a; 2022a; Bartlett et al., 2022; Balcan & Sharma, 2024). As far as we concern, in most prior research (Balcan et al., 2017; 2020a; 2021a; 2020b; 2021b; Bartlett et al., 2022), the hyperparameter tuning process does not involve the parameter $\boldsymbol{w}$ meaning that given any fixed hyperparameter $\alpha$, the behavior of the algorithm is determined. In some other cases that involves parameter $\boldsymbol{w}$, we can have a precise analytical characterization of how the optimal parameter behaves for any fixed hyperparameter (Balcan et al., 2022a), or at least a uniform approximate characterization (Balcan et al., 2024a). However, our setting does not belong to those cases, and requires a novel proof approach to handle the challenging case of hyperparameter tuning of neural networks (see Appendix B for a detailed discussion).

## 2 PRELIMINARIES

**Setup.** We introduce a novel data-driven hyperparameter tuning framework for algorithms with trainable parameters. Our objective is to optimize a hyperparameter $\alpha \in \mathcal{A} = [\alpha_{\min}, \alpha_{\max}] \subset \mathbb{R}$ for an algorithm that also involves model parameters $\boldsymbol{w} \in [w_{\min}, w_{\max}]^d \subset \mathbb{R}^d$. For a given problem

instance $\boldsymbol{x} \in \mathcal{X}$, we measure the model's performance as $f(\boldsymbol{x}, \boldsymbol{w}; \alpha)$, where $\boldsymbol{w}$ represents the model parameters and $\alpha$ the hyperparameter. We then define a utility function $u_\alpha(\boldsymbol{x})$ to quantify the algorithm's performance with hyperparameter $\alpha$ on problem instance $\boldsymbol{x}$: $u_\alpha(\boldsymbol{x}) = \max_{w \in \mathcal{W}} f(\boldsymbol{x}, \boldsymbol{w}; \alpha)$. This formulation can be interpreted as follows: for a given hyperparameter $\alpha$ and problem instance $\boldsymbol{x}$, we determine the optimal model parameters $\boldsymbol{w}$ that maximize performance.

In the data-driven framework, we assume an underlying, application-specific problem distribution $\mathcal{D}$ over $\mathcal{X}$. The best hyperparameter $\alpha^*$ for $\mathcal{D}$ can be defined as $\alpha^* \in \arg\max_\alpha \mathbb{E}_{\boldsymbol{x} \sim \mathcal{D}}[u_\alpha(\boldsymbol{x})]$. However, since the problem distribution $\mathcal{D}$ is unknown, we instead use a set $S$ of $N$ problem instances at hand, $S = \{\boldsymbol{x}_1, \ldots, \boldsymbol{x}_N\}$ drawn from $\mathcal{D}$. The hyperparameter $\hat{\alpha}_{\text{ERM}}$ is then chosen to maximize the empirical utility: $\hat{\alpha}_{\text{ERM}} \in \arg\max_\alpha \frac{1}{N} \sum_{i=1}^N u_\alpha(\boldsymbol{x}_i)$.

**Main question.** Our goal is to answer the learning-theoretic question: *How good is the tuned hyperparameter compared to the best hyperparameter, for algorithms with trainable parameters?* Specifically, we aim to provide a high-probability guarantee for the difference between the performance of $\hat{\alpha}_{\text{ERM}}$ and $\alpha^*$, expressed as: $|\mathbb{E}_{\boldsymbol{x} \sim \mathcal{D}}[u_{\hat{\alpha}_{\text{ERM}}}(\boldsymbol{x})] - \mathbb{E}_{\boldsymbol{x} \sim \mathcal{D}}[u_{\alpha^*}(\boldsymbol{x})]|$. Let $\mathcal{U} = \{u_\alpha : \mathbb{R} \to [0, H] \mid \alpha \in \mathcal{A}\}$ be the utility function class. Classical theory suggests that the learning-theoretic question at hand is equivalent to analyzing the pseudo-dimension (Pollard, 2012) or Rademacher complexity (Wainwright, 2019) (see Appendix C for further background) of the function class $\mathcal{U}$. However, this analysis poses significant challenges due to two primary factors: (1) the intricate structure of the function class itself, where a small change in $\alpha$ can lead to large changes in the utility function $u_\alpha$, and (2) $u_\alpha$ is computed by solving an optimization problem over the trainable parameters, and its explicit structure is unknown and hard to characterize. These challenges make analyzing the learning-theoretic complexity of $\mathcal{U}$ particularly challenging.

In this work, we demonstrate that when the function $f(\boldsymbol{x}, \boldsymbol{w}; \alpha)$ exhibits a certain degree of structure, we can establish an upper bound for the learning-theoretic complexity of the utility function class $\mathcal{U}$. Specifically, we examine two scenarios: (1) where $f(\boldsymbol{x}, \boldsymbol{w}; \alpha)$ possesses a piecewise constant structure (Section 4), and (2) where it exhibits a piecewise polynomial (or rational) structure (Section 5). These piecewise structures hold in hyperparameter tuning for popular deep learning algorithms (Section 6).

**Remark 1.** Note that our bounds on the learning-theoretic complexity of the dual utility function class implies bounded sample complexity for ERM, but the algorithmic question of actually implementing this ERM efficiently is left open for future research.

**Methodology.** The general approach to analyzing the complexity of the utility function class $\mathcal{U}$ is via analyzing its dual functions. Specifically, for each problem instance $\boldsymbol{x}$, we define the dual utility function $u_{\boldsymbol{x}}^* : \mathcal{A} \to [0, H]$ as follows:

$$u_{\boldsymbol{x}}^*(\alpha) := u_\alpha(\boldsymbol{x}) = \max_{\boldsymbol{w} \in \mathcal{W}} f(\boldsymbol{x}, \boldsymbol{w}; \alpha) = \max_{\boldsymbol{w} \in \mathcal{W}} f_{\boldsymbol{x}}(\alpha, \boldsymbol{w}).$$

Our key technical contribution is to demonstrate that when $f_{\boldsymbol{x}}(\alpha, \boldsymbol{w}) := f(\boldsymbol{x}, \boldsymbol{w}; \alpha)$ exhibits a piecewise structure, $u_{\boldsymbol{x}}^*(\alpha)$ also admits favorable structural properties, which depend on the specific structure of $f_{\boldsymbol{x}}(\alpha, \boldsymbol{w})$. We present some useful results that allow us to derive the learning-theoretic complexity of $\mathcal{U}$ from the structural properties of $u_{\boldsymbol{x}}^*(\alpha)$ (Section 3).

**Oscillations and its connection with pseudo-dimension.** When the function class $\mathcal{U} = \{u_\rho : \mathcal{X} \to \mathbb{R} \mid \rho \in \mathbb{R}\}$ is parameterized by a real-valued index $\rho$, Balcan et al. (2021a) propose a convenient way of bounding the pseudo-dimension of $\mathcal{H}$, via bounding the *oscillations* of the dual function $u_{\boldsymbol{x}}^*(\rho) := u_\rho(\boldsymbol{x})$ corresponding to any problem instance $\boldsymbol{x}$. We recall the notions of oscillation and its connection with the pseudo-dimension of the dual function class.

**Definition 1** (Oscillations, Balcan et al. 2021a). A function $h : \mathbb{R} \to \mathbb{R}$ has at most $B$ oscillations if for every $z \in \mathbb{R}$, the function $\rho \mapsto \mathbb{I}_{\{h(\rho) \geq z\}}$ is piecewise constant with at most $B$ discontinuities.

An illustration of the notion of oscillations can be found in Figure 1. Using the idea of oscillations, one can analyze the pseudo-dimension of parameterized function classes by alternatively analyzing the oscillations of their dual functions, formalized as follows.

**Theorem 2.1** (Balcan et al. 2021a). *Let $\mathcal{U} = \{u_\rho : \mathcal{X} \to \mathbb{R} \mid \rho \in \mathbb{R}\}$, of which each dual function $u_{\boldsymbol{x}}^*(\rho)$ has at most $B$ oscillations. Then $\text{Pdim}(\mathcal{U}) = \mathcal{O}(\ln B)$.*

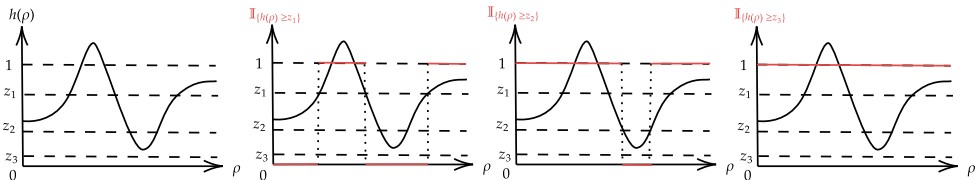

Figure 1: The oscillation of a function $h : \mathbb{R} \to \mathbb{R}$ is defined as the maximum number of discontinuities in the function $\mathbb{I}_{\{h(\rho) \geq z\}}$, as the threshold $z$ varies. When $z = z_1$, the function $\mathbb{I}_{\{h(\rho) \geq z\}}$ exhibits the highest number of discontinuities, which is four. Therefore, $h$ has 4 oscillations.

## 3   OSCILLATIONS OF PIECEWISE CONTINUOUS FUNCTIONS

We first establish connection between the number of oscillations in a piecewise continuous function and its local extrema and discontinuities. It serves as a general tool to upper-bound the pseudo-dimension of function classes via analyzing the piecewise continuous structure their dual functions.

**Lemma 3.1.** *Let $h : \mathbb{R} \to \mathbb{R}$ be a piecewise continuous function which has at most $B_1$ discontinuity points, and has at most $B_2$ local maxima. Then $h$ has at most $\mathcal{O}(B_1 + B_2)$ oscillations.*

*Proof Sketch.* The proof can be found in Appendix D. The idea is to bound the number of solutions of $h(\rho) = 0$, which determines the number of oscillations for $h$. We show that in each interval where $h$ is continuous, we can bound the number of solutions of $h(\rho) = 0$ using the number of local maxima of $h$. Aggregating the number of solutions across all continuous intervals of $h$ yields the desired result. $\square$

From Lemma 3.1 and Theorem 2.1, we have the following result which allows us to bound the pseudo-dimension of a function class $\mathcal{H}$ via bounding the number of discontinuity and local extrema points of any function in its dual function class $\mathcal{H}^*$.

**Corollary 3.2.** *Consider a real-valued function class $\mathcal{U} = \{u_\rho : \mathcal{X} \to \mathbb{R} \mid \rho \in \mathbb{R}\}$, of which each dual function $u_{\boldsymbol{x}}^*(\rho)$ is piecewise continuous, with at most $B_1$ discontinuities and $B_2$ local maxima. Then $\mathrm{Pdim}(\mathcal{H}) = \mathcal{O}(\ln(B_1 + B_2))$.*

We now consider piecewise constant functions with finite discontinuities. Despite infinite local extrema making Lemma 3.1 inapplicable, the function's special structure allows bounding oscillations via its number of discontinuities.

**Lemma 3.3.** *Consider a real-valued function class $\mathcal{U} = \{u_\rho : \mathcal{X} \to \mathbb{R} \mid \rho \in \mathbb{R}\}$, of which each dual function $u_{\boldsymbol{x}}^*(\rho)$ is piecewise constant with at most $B$ discontinuities. Then $\mathrm{Pdim}(\mathcal{U}) = \mathcal{O}(\ln B)$.*

## 4   $f_{\boldsymbol{x}}(\alpha, \boldsymbol{w})$ IS PIECEWISE CONSTANT

We first examine the case where $f_{\boldsymbol{x}}(\alpha, \boldsymbol{w})$ exhibits a *piecewise constant* structure with $N$ pieces. Specifically, we assume there exists a partition $\mathcal{P}_{\boldsymbol{x}} = \{R_{\boldsymbol{x},1}, \ldots, R_{\boldsymbol{x},N}\}$ of the domain $\mathcal{A} \times \mathcal{W}$ of $f_{\boldsymbol{x}}$, where each $R_{\boldsymbol{x},i}$ in $\mathcal{P}_{\boldsymbol{x}}$ is a connected set. Over the region $R_{\boldsymbol{x},i}$, the value of $f_{\boldsymbol{x}}$ is $f_{\boldsymbol{x},i}$ which is a constant value $c_i$ for any $(\alpha, \boldsymbol{w}) \in R_{\boldsymbol{x},i}$. Consequently, we can reformulate $u_{\boldsymbol{x}}^*(\alpha)$ as follows:

$$u_{\boldsymbol{x}}^*(\alpha) = \sup_{\boldsymbol{w} \in \mathcal{W}} f_{\boldsymbol{x}}(\alpha, \boldsymbol{w}) = \max_{R_{\boldsymbol{x},i}} \sup_{\boldsymbol{w}:(\alpha, \boldsymbol{w}) \in R_{\boldsymbol{x},i}} f_{\boldsymbol{x}}(\alpha, \boldsymbol{w}) = \max_{R_{\boldsymbol{x},i}:\exists \boldsymbol{w},(\alpha, \boldsymbol{w}) \in R_{\boldsymbol{x},i}} c_i.$$

This leads to Lemma 4.1, which asserts that $u_{\boldsymbol{x}}^*(\alpha)$ is a piecewise constant function and provides an upper bound for the number of discontinuities in $u_{\boldsymbol{x}}^*(\alpha)$.

**Lemma 4.1.** *Assume that the piece functions $f_i(\alpha, \boldsymbol{w})$ is constant for all $i \in [N]$. Then $u_{\boldsymbol{x}}^*(\alpha)$ has $\mathcal{O}(N)$ discontinuity points, partitioning $\mathcal{A}$ into at most $\mathcal{O}(N)$ regions. In each region, $u_{\boldsymbol{x}}^*(\alpha)$ is a constant function.*

The proof idea is demonstrated in Figure 2, and the detailed proof can be found in Appendix D. By combining Lemma 4.1 and Lemma 3.3, we have the following result, which establishes learning guarantees for the utility function class $\mathcal{U}$ when $f_{\boldsymbol{x}}(\alpha, \boldsymbol{w})$ admits piecewise constant structure.

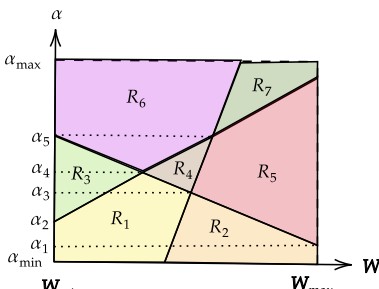

Figure 2: A demonstration of the proof idea for Lemma 4.1: We begin by partitioning the domain $\mathcal{A}$ of the dual utility function $u_{\boldsymbol{x}}^*(\alpha)$ into intervals. This partitioning is formed using two key points for each connected component $R$ in the partition $\mathcal{P}_{\boldsymbol{x}}$ of the domain $\mathcal{A} \times \mathcal{W}$ of $f_{\boldsymbol{x}}(\alpha, \boldsymbol{w})$: $\alpha_{R,\inf} = \inf_\alpha\{\alpha : \exists \boldsymbol{w}, (\alpha, \boldsymbol{w}) \in R\}$ and $\alpha_{R,\sup} = \sup_\alpha\{\alpha : \exists \boldsymbol{w}, (\alpha, \boldsymbol{w}) \in R\}$. Given that $\mathcal{P}$ contains $N$ elements, the number of such points is $\mathcal{O}(N)$. We demonstrate that the dual utility functions $u_{\boldsymbol{x}}^*$ remain constant over each interval defined by these points.

**Theorem 4.2.** *Consider the utility function class $\mathcal{U} = \{u_\alpha : \mathcal{X} \to [0, H] \mid \alpha \in \mathcal{A}\}$. Assume that $f_{\boldsymbol{x}}(\alpha, \boldsymbol{w})$ admits piecewise constant structure with $N$ pieces over $\mathcal{A} \times \mathcal{W}$. Then for any distribution $\mathcal{D}$ over $\mathcal{X}$, and any $\delta \in (0, 1)$, with probability at least $1 - \delta$ over the draw of $S \sim \mathcal{D}$, we have*

$$|\mathbb{E}_{\boldsymbol{x} \sim \mathcal{D}}[u_{\hat{\alpha}_{ERM}}(\boldsymbol{x})] - \mathbb{E}_{\boldsymbol{x} \sim \mathcal{D}}[u_{\alpha^*}(\boldsymbol{x})]| = \mathcal{O}\left(\sqrt{\frac{\log(N/\delta)}{m}}\right).$$

**Remark 2.** The partition of $f_{\boldsymbol{x}}(\alpha, \boldsymbol{w})$ into connected components is defined by $S$ boundary functions $h_i(\alpha, \boldsymbol{w})$, which are typically polynomials of degree $\Delta$ in $d + 1$ variables. For these cases, we can bound the number of connected components in $\mathbb{R}^d - \cup_{i=1}^S Z(h_i)$ using only $\Delta$ and $d$, which is key for applying Theorem 4.2. Further details are in Appendix E.2.

## 5 $f_{\boldsymbol{x}}(\alpha, \boldsymbol{w})$ IS PIECEWISE POLYNOMIAL

In this section, we examine the case where $f_{\boldsymbol{x}}(\alpha, \boldsymbol{w})$ exhibits a piecewise polynomial structure. The domain $\mathcal{A} \times \mathcal{W}$ of $f_{\boldsymbol{x}}$ is divided into $N$ connected components by $M$ polynomials $h_{\boldsymbol{x},1}, \ldots, h_{\boldsymbol{x},M}$ in $\alpha, \boldsymbol{w}$, each of degree at most $\Delta_b$. The resulting partition $\mathcal{P}_{\boldsymbol{x}} = \{R_{\boldsymbol{x},1}, \ldots, R_{\boldsymbol{x},N}\}$ consists of connected sets $R_{\boldsymbol{x},i}$, each formed by a connected component $C_{\boldsymbol{x},i}$ and its adjacent boundaries. Within each $R_{\boldsymbol{x},i}$, $f_{\boldsymbol{x}}$ takes the form of a polynomial $f_{\boldsymbol{x},i}$ in $\alpha$ and $\boldsymbol{w}$ of degree at most $\Delta_p$. The dual utility function $u_{\boldsymbol{x}}^*(\alpha)$ is defined as:

$$u_{\boldsymbol{x}}^*(\alpha) = \sup_{\boldsymbol{w} \in \mathcal{W}} f_{\boldsymbol{x}}(\alpha, \boldsymbol{w}) = \max_{i \in [N]} \sup_{\boldsymbol{w}:(\alpha, \boldsymbol{w}) \in R_i} f_{\boldsymbol{x},i}(\alpha, \boldsymbol{w}) = \max_{i \in [N]} u_{\boldsymbol{x},i}^*(\alpha),$$

where $u_{\boldsymbol{x},i}^*(\alpha) = \sup_{\boldsymbol{w}:(\alpha, \boldsymbol{w}) \in R_{\boldsymbol{x},i}} f_{\boldsymbol{x},i}(\alpha, \boldsymbol{w})$. We begin with the following regularity assumption on the piece and boundary functions $f_{\boldsymbol{x},j}$ and $h_{\boldsymbol{x},i}$.

**Assumption 1.** Assume that for any function $u_{\boldsymbol{x}}^*(\alpha)$, its pieces functions $f_{\boldsymbol{x}}^*$ and boundaries $h_{\boldsymbol{x},1}, \ldots, h_{\boldsymbol{x},M}$: for any piece function $f_{\boldsymbol{x}}, i$ and $S \leq d + 1$ boundaries $h_1, \ldots, h_S$ chosen from $\{h_{\boldsymbol{x},1}, \ldots, h_{\boldsymbol{x},M}\}$, we have $\boldsymbol{0}$ is a regular value of $\overline{k}(\alpha, \boldsymbol{w}, \boldsymbol{\lambda})$. Here $k = (k_1, \ldots, k_{d+S})$, $\overline{k} = (k_1, \ldots, k_{d+S}, \det(J_{k,(\boldsymbol{w}, \boldsymbol{\lambda})}))$, $J_{k,(\boldsymbol{w}, \boldsymbol{\lambda})}$ is the Jacobian of $k$ w.r.t. $\boldsymbol{w}$ and $\boldsymbol{\lambda}$, and $k_1, \ldots, k_{d+S}$ defined as

$$\begin{cases} k_i(\alpha, \boldsymbol{w}, \boldsymbol{\lambda}) = h_i(\alpha, \boldsymbol{w}), & i = 1, \ldots, S, \\ k_{S+j}(\alpha, \boldsymbol{w}, \boldsymbol{\lambda}) = \frac{\partial f_{\boldsymbol{x},i}}{\partial w_j} + \sum_{i=1}^s \lambda_i \frac{\partial h_i}{\partial w_j}, & j = 1, \ldots, d. \end{cases}$$

Intuitively, Assumption 1 states that the preimage $\overline{k}^{-1}(\boldsymbol{0})$, consistently exhibits regular structure (smooth manifolds). This assumption helps us in identifying potential locations of $w^*$ that maximize $f_{\boldsymbol{x},i}(\alpha)$ for each fixed $\alpha$, ensuring these locations have a regular structure. We note that this assumption is both common in constrained optimization theory and relatively mild. For a smooth

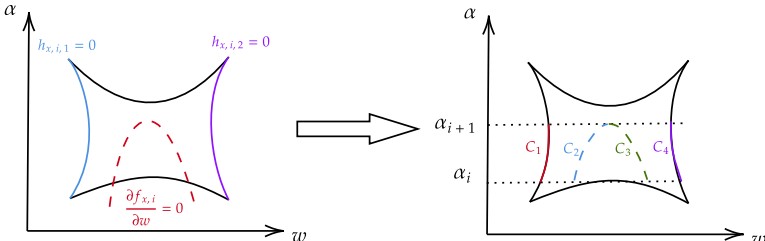

Figure 3: A simplified illustration for the proof idea of Theorem 5.1 where $\boldsymbol{w} \in \mathbb{R}$. Here, our goal is to analyze the number of discontinuities and local maxima of $u^*_{\boldsymbol{x},i}(\alpha)$. The idea is to partition the hyperparameter space $\mathcal{A}$ into intervals such that over each interval, the function $u^*_{\boldsymbol{x},i}(\alpha)$ is the point-wise maximum of $f_{\boldsymbol{x},i}(\alpha, \boldsymbol{w})$ along some fixed set of "monotonic curves" $\mathcal{C}$ (curves that intersect $\alpha = \alpha_0$ at most once for any $\alpha_0$). $u^*_{\boldsymbol{x},i}(\alpha)$ is continuous over such interval; this implies that the interval end points contain all discontinuities of $u^*_{\boldsymbol{x},i}(\alpha)$. In this example, over the interval $(\alpha_i, \alpha_{i+1})$, we have $u^*_{\boldsymbol{x},i}(\alpha) = \max_{C_i}\{f_{\boldsymbol{x},i}(\alpha, \boldsymbol{w}) : (\alpha, \boldsymbol{w}) \in C_i\}$. Then, we can show that over such an interval, any local maximum of $u^{\boldsymbol{x},i}(\alpha)$ is a local extremum of $f_{\boldsymbol{x},i}(\alpha, \boldsymbol{w})$ along a monotonic curve $C \in \mathcal{C}$. Finally, we bound the number of points used for partitioning and local extrema using tools from algebraic and differential geometry.

mapping $\bar{k}$, Sard's theorem (Theorem F.12) asserts that the set of values that are not regular values of $\bar{k}$ has Lebesgue measure zero. This theoretical basis further suggests that the Assumption 1 is reasonable.

Under Assumption 1, we have the following result, which gives us learning-theoretic guarantees for tuning the hyperparameter $\alpha$ for the utility function class $\mathcal{U}$.

**Theorem 5.1.** *Consider the utility function class $\mathcal{U} = \{u_\alpha : \mathcal{X} \to [0, H] \mid \alpha \in \mathcal{A}\}$. Assume that $f_{\boldsymbol{x}}(\alpha, \boldsymbol{w})$ admits piecewise polynomial structure with the piece functions $f_{\boldsymbol{x},i}$ and boundaries $h_{\boldsymbol{x},i}$ satisfies Assumption 1. Then for any distribution $\mathcal{D}$ over $\mathcal{X}$, for any $\delta \in (0, 1)$, with probability at least $1 - \delta$ over the draw of $S \sim \mathcal{D}^m$, we have*

$$|\mathbb{E}_{\boldsymbol{x} \sim \mathcal{D}}[u_{\hat{\alpha}_{ERM}}(\boldsymbol{x})] - \mathbb{E}_{\boldsymbol{x} \sim \mathcal{D}}[u_{\alpha^*}(\boldsymbol{x})]| = \mathcal{O}\left(\sqrt{\frac{\log N + d \log(\Delta M) + \log(1/\delta)}{m}}\right).$$

*Here, $M$ and $N$ are the number of boundaries and connected sets, $\Delta = \max\{\delta_p, \delta_d\}$ is the maximum degree of piece $f_{\boldsymbol{x},i}$ and boundaries $h_{\boldsymbol{x},i}$.*

*Proof Sketch.* We defer the detailed proof to Appendix F.7. The proof is fairly involved and employs many novel ideas, we break it down into the following steps:

1. We first demonstrate that if the piece functions $f_{\boldsymbol{x},i}$ and boundaries $h_{\boldsymbol{x},i}$ satisfy a stronger assumption (Assumption 2), we can bound the pseudo-dimension of $\mathcal{U}$ (Theorem F.19). The details of this step are presented in Appendix F.7.1, with a simplified illustration of the proof idea in Figure 3. The proof follows these steps:

   (a) Using Lemma 3.2, we show that it suffices to bound the number of discontinuities and local maxima of $u^*_{\boldsymbol{x}}$, which is equivalent to bounding those of $u^*_{\boldsymbol{x},i}$.

   (b) We first demonstrate that the domain $\mathcal{A}$ can be partitioned into $\mathcal{O}\left((2\Delta)^{d+1}\left(\frac{eM}{d+1}\right)^{d+1}\right)$ intervals. For each interval $I_t$, there exists a set of subsets of boundaries $\mathbf{S}^1_{\boldsymbol{x},t} \subset \mathbf{H}_{\boldsymbol{x},i}$ such that for any set of boundaries $\mathcal{S} \in \mathbf{S}^1_{\boldsymbol{x},t}$, the intersection of boundaries in $\mathcal{S}$ contains a feasible point $(\alpha, \boldsymbol{w})$ for any $\alpha$ in that interval. The key idea of this step is using the $\alpha$-extreme points (Definition 5) of connected components of such intersection, which can be upper-bounded using Lemma F.10.

(c) We refine the partition of $\mathcal{A}$ into $\mathcal{O}\left((2\Delta)^{2d+2}\left(\frac{eM}{d+1}\right)^{d+1}\right)$ intervals. For each interval $I_t$, there exists a set of subsets of boundaries $\mathbf{S}^2_{\boldsymbol{x},t} \subset \mathbf{H}_{\boldsymbol{x},i}$ such that for any set of boundaries $\mathcal{S} \in \mathbf{S}^2_{\boldsymbol{x},t}$ and any $\alpha$ in such intervals, there exist $\boldsymbol{w}$ and $\boldsymbol{\lambda}$ satisfying Lagrangian stationarity:

$$\begin{cases} h_{\mathcal{S},j}(\alpha, \boldsymbol{w}_\alpha) = 0, j = 1, \ldots, S \\ \frac{\partial f(\alpha, \boldsymbol{w}_\alpha)}{\partial w_i} + \sum_{j=1}^{S} \lambda_j \frac{\partial h_{\mathcal{S},j}(\alpha, \boldsymbol{w}_\alpha)}{\partial w_i}, i = 1, \ldots, d. \end{cases}$$

This defines a smooth 1-manifold $\mathcal{M}^\mathcal{S}$ in $\mathbb{R} \times \mathbb{R}^d \times \mathbb{R}^S$ from Assumption 2. The key idea of this step is using Theorem F.7, and $\alpha$-extreme points of connected components of $\mathcal{M}^\mathcal{S}$, which again can be upper-bounded using Lemma F.10.

(d) We further refine the partition of $\mathcal{A}$ into $\mathcal{O}\left(M(2\Delta)^{2d+2}\left(\frac{eM}{d+1}\right)^{d+1}\right)$ intervals. For each interval $I_t$, there exists a set of subsets of boundaries $\mathbf{S}^3_{\boldsymbol{x},t} \subset \mathbf{H}_{\boldsymbol{x},i}$ such that for any $\alpha$ in that interval and any manifold $\mathcal{M}^\mathcal{S}$, there exists a *feasible* point $(\alpha, \boldsymbol{w}, \boldsymbol{\lambda})$ in $\mathcal{M}^\mathcal{S}$, i.e., $(\alpha, \boldsymbol{w}) \in \overline{R}_{\boldsymbol{x},i}$. The key idea of this step is upper-bounding the number of intersections between $\mathcal{M}^\mathcal{S}$ with any other boundary $h' \notin \mathcal{S}$.

(e) We show that each manifold $\mathcal{M}^\mathcal{S}$ can be partitioned into *monotonic curves* (Definition 12). We then partition $\mathcal{A}$ one final time into $\mathcal{O}\left(\Delta^{4d+2}\left(\frac{eM}{d+1}\right)^{d+1} + M(2\Delta)^{2d+2}\left(\frac{eM}{d+1}\right)^{d+1}\right)$ intervals. Over each interval $I_t$, the function $u^*_{\boldsymbol{x},i}$ can be represented as the value of $f_{\boldsymbol{x},i}$ along a fixed set of monotonic curves (see Figure 3). Hence, $u^*_{\boldsymbol{x},i}$ is continuous over $I_t$. Therefore, the points partitioning $\mathcal{A}$ contain the discontinuities of $u^*_{\boldsymbol{x},i}$. The key idea of this step is using our proposed definition and properties of monotonic curves (Proposition F.18), and Bezout's theorem.

(f) We further demonstrate that in each interval $I_t$, any local maximum of $u^*_{\boldsymbol{x},i}(\alpha)$ is a local maximum of $f_{\boldsymbol{x},i}(\alpha, \boldsymbol{w})$ along a monotonic curve (Lemma F.14)). Again, we can control the number of such points using Bezout's theorem.

(g) Finally, we put together all the potential discontinuities and local extrema of $u^*_{\boldsymbol{x},i}$. Combining with Lemma 3.2 we have the upper-bound for $\mathrm{Pdim}(\mathcal{U})$ (Theorem F.20).

2. We then demonstrate that for any function class $\mathcal{U}$ whose dual functions $u^*_{\boldsymbol{x}}$ have piece functions and boundaries satisfying Assumption 1, we can construct a new function class $\mathcal{V}$. The dual functions $v^*_{\boldsymbol{x}}$ of $\mathcal{V}$ have piece functions and boundaries that satisfy Assumption 2. Moreover, we show that $\|u^*_{\boldsymbol{x}} - v^*_{\boldsymbol{x}}\|_\infty$ can be made arbitrarily small. The details of this construction and proof are presented in Appendix F.7.2.

3. Finally, using the results from Step (1), we establish an upper bound on the pseudo-dimension for the function class $\mathcal{V}$ described in Step (2). Leveraging the approximation guarantee from Step (2), we can then use the results for $\mathcal{V}$ to determine the learning-theoretic complexity of $\mathcal{U}$ by applying Lemma C.3 and Lemma C.4. Standard learning theory literature then allows us to translate the learning-theoretic complexity of $\mathcal{U}$ into its learning guarantee. This final step is detailed in Appendix F.7.3. □

# 6 APPLICATIONS

We demonstrate the application of our results to two specific hyperparameter tuning problems in deep learning. We note that the problem might be presented as analyzing a loss function class $\mathcal{L} = \{\ell_\alpha : \mathcal{X} \to [0, H] \mid \alpha \in \mathcal{A}\}$ instead of utility function class $\mathcal{U} = \{u_\alpha : \mathcal{X} \to [0, H] \mid \alpha \in \mathcal{A}\}$, but our results still hold, just by defining $u_\alpha(\boldsymbol{x}) = H - \ell_\alpha(\boldsymbol{x})$. First, we establish bounds on the complexity of tuning the linear interpolation hyperparameter for activation functions, which is motivated by DARTS (Liu et al., 2019). Additionally, we explore the tuning of graph kernel parameters in Graph Neural Networks (GNNs).

## 6.1 DATA-DRIVEN TUNING FOR INTERPOLATION OF NEURAL ACTIVATION FUNCTIONS

**Problem settings.** We consider a feed-forward neural network $f$ with $L$ layers. Let $W_i$ denote the number of parameters in the $i^{th}$ layer, and $W = \sum_{i=1}^{L} W_i$ the total number of parameters. Besides,

8

we denote $k_i$ the number of computational nodes in layer $i$, and let $k = \sum_{i=1}^{L} k_i$. At each node, we choose between two piecewise polynomial activation functions, $o_1$ and $o_2$. For an activation function $o(z)$, we call $z_0$ a *breakpoint* where $o$ changes its behavior. For example, 0 is a breakpoint of the ReLU activation function. Liu et al. (2019) proposed a simple method for selecting activation functions: during training, they define a general activation function $\sigma$ as a weighted combination of $o_1$ and $o_2$. While their framework is more general, allowing for multiple activation functions and layer-specific activation, we analyze a simplified version. The combined activation function is given by:

$$\sigma(x) = \zeta o_1(x) + (1 - \zeta)o_2(x),$$

where $\zeta \in [0, 1]$ is the interpolation hyperparameter. This framework can express functions like the parametric ReLU, $\sigma(z) = \max\{0, z\} + \alpha \min\{0, z\}$, which empirically outperforms the regular ReLU (i.e., $\alpha = 0$) (He et al., 2015).

**Parametric regression.** In parametric regression, the final layer output is $g(\alpha, \boldsymbol{w}, \boldsymbol{x}) = \hat{y} \in \mathbb{R}^D$, where $\boldsymbol{w} \in \mathcal{W} \subset \mathbb{R}^W$ is the parameter vector and $\alpha$ is the architecture hyperparameter. The validation loss for a single example $(x, y)$ is $\|g(\alpha, \boldsymbol{w}, x) - y\|^2$, and for $T$ examples, we define

$$\ell_\alpha((X, Y)) = \min_{\boldsymbol{w} \in \mathcal{W}} \frac{1}{T} \sum_{(x,y) \in (X,Y)} \|g(\alpha, \boldsymbol{w}, x) - y\|^2 = \min_{\boldsymbol{w} \in \mathcal{W}} f((X, Y), \boldsymbol{w}; \alpha).$$

With $\mathcal{X}$ as the space of $T$-example validation sets, we define the loss function class $\mathcal{L}^{\text{AF}} = \{\ell_\alpha : \mathcal{X} \to \mathbb{R} \mid \alpha \in [\alpha_{\min}, \alpha_{\max}]\}$. We aim to provide a learning-theoretic guarantee for $\mathcal{L}^{\text{AF}}$.

**Theorem 6.1.** *Let $\mathcal{L}^{AF}$ denote loss function class defined above, with activation functions $o_1, o_2$ having maximum degree $\Delta$ and maximum breakpoints $p$. Given a problem instance $\boldsymbol{x} = (X, Y)$, the dual loss function is defined as $\ell_{\boldsymbol{x}}^*(\alpha) := \min_{\boldsymbol{w} \in \mathcal{W}} f(\boldsymbol{x}, \boldsymbol{w}; \alpha) = \min_{\boldsymbol{w} \in \mathcal{W}} f_{\boldsymbol{x}}(\alpha, \boldsymbol{w})$. Then, $f_{\boldsymbol{x}}(\alpha, \boldsymbol{w})$ admits piecewise polynomial structure with bounded pieces and boundaries. Further, if the piecewise structure of $f_{\boldsymbol{x}}(\alpha, \boldsymbol{w})$ satisfies Assumption 1, then for any $\delta \in (0, 1)$, w.p. at least $1 - \delta$ over the draw of problem instances $\boldsymbol{x} \sim \mathcal{D}^m$, where $\mathcal{D}$ is some distribution over $\mathcal{X}$, we have*

$$|\mathbb{E}_{\boldsymbol{x} \sim \mathcal{D}}[\ell_{\hat{\alpha}_{ERM}}(\boldsymbol{x})] - \mathbb{E}_{\boldsymbol{x} \sim \mathcal{D}}[\ell_{\alpha^*}(\boldsymbol{x})]| = \mathcal{O}\left(\sqrt{\frac{L^2W \log \Delta + LW \log(Tpk) + \log(1/\delta)}{m}}\right).$$

A full proof is located in Appendix G. Given a problem instance $(X, Y)$, the key idea is to establish the piecewise polynomial structure for the function $f_{(X,Y)}(\alpha, \boldsymbol{w})$ as a function of both the parameters $\boldsymbol{w}$ and the architecture hyperparameter $\alpha$, and then apply our main result Theorem 5.1. We establish this structure by extending the inductive argument due to Bartlett et al. (1998) which gives the piecewise polynomial structure of the neural network output as a function of the parameters $\boldsymbol{w}$ (i.e. when there are no hyperparameters) on any fixed collection of input examples. We also investigate the case where the network is used for classification task (see Appendix G.1.2).

### 6.2 DATA-DRIVEN HYPERPARAMETER TUNING FOR GRAPH POLYNOMIAL KERNELS

We now demonstrate the applicability of our proposed results in a simple scenario: tuning the hyperparameter of a graph kernel. Here, we consider the classification case and defer the regression case to Appendix.

**Partially labeled graph instance.** Consider a graph $\mathcal{G} = (\mathcal{V}, \mathcal{E})$, where $\mathcal{V}$ and $\mathcal{E}$ are sets of vertices and edges, respectively. Let $n = |\mathcal{V}|$ be the number of vertices. Each vertex in the graph is associated with a $d$-dimensional feature vector, and let $X \in \mathbb{R}^{n \times d}$ denote the matrix that contains all the vertices (as feature vectors) in the graph. We also have a set of indices $\mathcal{Y}_L \subset [n]$ of labeled vertices, where each vertex belongs to one of $C$ categories and $L = |\mathcal{Y}_L|$ is the number of labeled vertices. Let $y \in [F]^L$ be the vector representing the true labels of labeled vertices, where the coordinate $y_l$ of $y$ corresponds to the label of vertex $l \in \mathcal{Y}_L$.

We want to build a model for classifying the remaining (unlabeled) vertices, which correspond to $\mathcal{Y}_U = [n] \setminus \mathcal{Y}_L$. A popular and effective approach for this is to train a graph convolutional network (GCN) Kipf & Welling (2017). Along with the vertex matrix $X$, we are also given the distance

matrix $\boldsymbol{\delta} = [\delta_{i,j}]_{(i,j)\in[n]^2}$ encoding the correlation between vertices in the graph. The adjacency matrix $A$ is given by a polynomial kernel of degree $\Delta$ and hyperparameter $\alpha > 0$

$$A_{i,j} = (\delta(i,j) + \alpha)^\Delta.$$

Let $\tilde{A} = A + I_n$, where $I_n$ is the identity matrix, and $\tilde{D} = [\tilde{D}_{i,j}]_{[n]^2}$ where $\tilde{D}_{i,j} = 0$ if $i \neq j$, and $\tilde{D}_{i,i} = \sum_{j=1}^n \tilde{A}_{i,j}$ for $i \in [n]$. We then denote a problem instance $\boldsymbol{x} = (X, y, \boldsymbol{\delta}, \mathcal{Y}_L)$ and call $\mathcal{X}$ the set of all problem instances.

**Network architecture.** We consider a simple two-layer GCN $f$ (Kipf & Welling, 2017), which takes the adjacency matrix $A$ and vertex matrix $X$ as inputs and outputs $Z = f(X, A)$ of the form

$$Z = \hat{A}\,\mathrm{ReLU}(\hat{A}XW^{(0)})W^{(1)},$$

where $\hat{A} = \tilde{D}^{-1}\tilde{A}$ is the row-normalized adjacency matrix, $W^{(0)} \in \mathbb{R}^{d\times d_0}$ is the weight matrix of the first layer, and $W^{(1)} \in \mathbb{R}^{d_0\times F}$ is the hidden-to-output weight matrix. Here, $z_i$ is the $i^{th}$-row of $Z$ representing the score prediction of the model. The prediction $\hat{y}_i$ for vertex $i \in \mathcal{Y}_U$ is then computed from $Z$ as $\hat{y}_i = \max z_i$ which is the maximum coordinate of vector $z_i$.

**Objective function and the loss function class.** We consider the 0-1 loss function corresponding to hyperparameter $\alpha$ and network parameters $\boldsymbol{w} = (\boldsymbol{w}^{(0)}, \boldsymbol{w}^{(1)})$ for given problem instance $\boldsymbol{x}$, $f(\boldsymbol{x}, \boldsymbol{w}; \alpha) = \frac{1}{|\mathcal{Y}_L|}\sum_{i\in\mathcal{Y}_L} \mathbb{I}_{\{\hat{y}_i\neq y_i\}}$. The dual loss function corresponding to hyperparameter $\alpha$ for instance $\boldsymbol{x}$ is given as $\ell_\alpha(\boldsymbol{x}) = \max_{\boldsymbol{w}} f(\boldsymbol{x}, \boldsymbol{w}; \alpha)$, and the corresponding loss function class is $\mathcal{L}^{\mathrm{GCN}} = \{l_\alpha : \mathcal{X} \to [0,1] \mid \alpha \in \mathcal{A}\}$.

To analyze the learning guarantee of $\mathcal{L}^{\mathrm{GCN}}$, we first show that any dual loss function $\ell_{\boldsymbol{x}}^*(\alpha) := \ell_\alpha(\boldsymbol{x}) = \min_{\boldsymbol{w}} f_{\boldsymbol{x}}(\alpha, \boldsymbol{w})$, $f_{\boldsymbol{x}}(\alpha, \boldsymbol{w})$ has a piecewise constant structure, where: The pieces are bounded by rational functions of $\alpha$ and $\boldsymbol{w}$ with bounded degree and positive denominators. We bound the number of connected components created by these functions and apply Theorem 4.2 to derive our result. The full proof is in Appendix G.2.1.

**Theorem 6.2.** *Let $\mathcal{L}^{GCN}$ denote the loss function class defined above. Given a problem instance $\boldsymbol{x}$, the dual loss function is defined as $\ell_{\boldsymbol{x}}^*(\alpha) := \min_{\boldsymbol{w}\in\mathcal{W}} f(\boldsymbol{x}, \boldsymbol{w}; \alpha)) = \min_{\boldsymbol{w}\in\mathcal{W}} f_{\boldsymbol{x}}(\alpha, \boldsymbol{w})$. Then $f_{\boldsymbol{x}}(\alpha, \boldsymbol{w})$ admits piecewise constant structure. Furthermore, for any $\delta \in (0,1)$, w.p. at least $1 - \delta$ over the draw of problem instances $\boldsymbol{x} = (\boldsymbol{x}_1, \dots, \boldsymbol{x}_m) \sim \mathcal{D}^m$, where $\mathcal{D}$ is some problem distribution over $\mathcal{X}$, we have*

$$|\mathbb{E}_{\boldsymbol{x}\sim\mathcal{D}}[\ell_{\hat{\alpha}_{ERM}}(\boldsymbol{x})] - \mathbb{E}_{\boldsymbol{x}\sim\mathcal{D}}[\ell_{\alpha^*}(\boldsymbol{x})]| = \mathcal{O}\left(\sqrt{\frac{d_0(d+F)\log nF\Delta + \log(1/\delta)}{m}}\right).$$

Our results also bound the sample complexity for learning the GCN graph kernel hyperparameter $\alpha$ when minimizing squared loss in regression (Theorem G.5, Appendix G.2.2).

# 7 CONCLUSION AND FUTURE WORK

In this work, we establish the first principled approach to hyperparameter tuning in deep networks with provable guarantees, by employing the lens of data-driven algorithm design. We integrate subtle concepts from algebraic and differential geometry with our proposed ideas, and establish the learning-theoretic complexity of hyperparameter tuning when the neural network loss is a piecewise constant or piecewise polynomial function of the parameters and the hyperparameter. We demonstrate applications of our results in multiple contexts, including tuning graph kernels for graph convolutional networks and neural architecture search.

This work opens up several directions for future research. While we resolve several technical hurdles to handle the piecewise polynomial case, it would be useful to also study cases where the piecewise functions or boundaries involve logarithmic, exponential, or more generally, Pfaffian functions (Khovanski, 1991). We study the case of tuning a single hyperparameter, a natural next question is to determine if our results can be extended to tuning multiple hyperparameters simultaneously. Finally, while our work primarily focuses on providing learning-theoretic sample complexity guarantees, developing computationally efficient methods for hyperparameter tuning in data-driven settings is another avenue for future research.

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

## A ADDITIONAL RELATED WORK

**Learning-theoretic complexity of deep nets.** A related line of work studies the learning-theoretic complexity of deep networks, corresponding to selection of network parameters (weights) over a single problem instance. Bounds on the VC dimension of neural networks have been shown for piecewise linear and polynomial activation functions (Maass, 1994; Bartlett et al., 1998) as well as the broader class of Pfaffian activation functions Karpinski & Macintyre (1997). Recent work includes near-tight bounds for the piecewise linear activation functions (Bartlett et al., 2019) and data-dependent margin bounds for neural networks (Bartlett et al., 2017).

**Data-driven algorithm design.** Data-driven algorithm design, also known as self-improved algorithms (Balcan, 2020; Ailon et al., 2011; Gupta & Roughgarden, 2020), is an emerging field that adapts algorithms' internal components to specific problem instances, particularly in parameterized algorithms with multiple performance-dictating hyperparameters. Unlike traditional worst-case or average-case analysis, this approach assumes problem instances come from an application-specific distribution. By leveraging available input problem instances, this approach seeks to maximize empirical utilities that measure algorithmic performance for those specific instances. This method has demonstrated effectiveness across various domains, including low-rank approximation and dimensionality reduction (Li et al., 2023; Indyk et al., 2019; Ailon et al., 2021), accelerating linear system solvers (Luz et al., 2020; Khodak et al., 2024), mechanism design (Balcan et al., 2016; 2018c), sketching algorithms (Bartlett et al., 2022), branch-and-cut algorithms for (mixed) integer linear programming (Balcan et al., 2021b), among others.

**Neural architecture search.** Neural architecture search (NAS) captures a significant part of the engineering challenge in deploying deep networks for a given application. While neural networks successfully automate the tedious task of "feature engineering" associated with classical machine learning techniques by automatically learning features from data, it requires a tedious search over a large search space to come up with the best neural architecture for any new application domain. Multiple different approaches with different search spaces have been proposed for effective NAS, including searching over the discrete topology of connections between the neural network nodes, and interpolation of activation functions. Due to intense recent interest in moving from hand-crafted to automatically searched architectures, several practically successful approaches have been developed including framing NAS as Bayesian optimization (Bergstra et al., 2013; Mendoza et al., 2016; White et al., 2021), reinforcement learning (Zoph & Le, 2017; Baker et al., 2017), tree search (Negrinho & Gordon, 2017; Elsken et al., 2017), gradient-based optimization (Liu et al., 2019), among others, with progress measured over standard benchmarks (Dong & Yang, 2020; Mehta et al., 2022). Li et al. (2021) introduce a geometry-aware mirror descent based approach to learn the network architecture and weights simultaneously, within a single problem instance, yielding a practical algorithm but without provable guarantees. Our formulation is closely related to tuning the interpolation parameter for activation parameter in NAS approach of DARTS Liu et al. (2019), which can be viewed as a multi-hyperparameter generalization of our setup. We establish the first learning guarantees for the simpler case of single hyperparameter tuning. Note that we are considering a simplified version of DARTS Liu et al. (2019), where we consider a linear interpolation hyperparameter of activation in each node, while DARTS uses a probabilistic interpolation instead.

**Graph-based learning.** While several classical (Blum & Chawla, 2001; Zhu et al., 2003; Zhou et al., 2003; Zhu, 2005) as well as neural models (Kipf & Welling, 2017; Velic kovic et al., 2018; Wu et al., 2019; Gilmer et al., 2017) have been proposed for graph-based learning, the underlying graph used to represent the data typically involves heuristically set graph parameters. The latter approach is usually more effective in practice, but comes without formal learning guarantees. Our work provides the first provable guarantees for tuning the graph kernel hyperparameter in graph neural networks.

**A detailed comparison to Hyperband (Li et al., 2018).** Hyperband is one of the most notable work in hyperparameter tuning. Specially, the paper provides a theoretical guarantees for the hyperparameter tuning process, but under strong assumptions. Here, we provide a detailed comparison between guarantees presented in Hyperband and our results, and explain how Hyperband and our work are not competing but complementing each others.

1. **Hyperparameter configuration settings:** Theoretical results (Theorem 1, Proposition 4) in Hyperband assumes finitely many distinct arms and guarantees are with respect to the best arm in that set. Even their infinite arm setting considers a distribution over the hyperparameter space from which $n$ arms are sampled. It is assumed that $n$ is large enough to sample a good arm with high probability without actually showing that this holds for any concrete hyperparameter loss landscape. It is not clear why this assumption will hold in our cases. In sharp contrast, we seek optimality over the entire continuous hyperparameter hyperparameter range for concrete loss functions which satisfy a piecewise polynomial dual structure.

2. **Guarantee settings**: The notion of "sample complexity" in Hyperband is very different from ours. Intuitively, their goal is to find the best hyperparameter from learning curves over fewest training epochs, assuming the test loss converges to a fixed value for each hyperparameter after some epochs. By ruling out (successively halving) hyperparameters that are unlikely to be optimal early, they speed up the search process (by avoiding full training epochs for suboptimal hyperparameters). In contrast, we focus on model hyperparameters and assume the network can be trained to optimality for any value of the hyperparameter. We ignore the computational efficiency aspect and focus on the data (sample) efficiency aspect which is not captured in Hyperband analysis.

3. **Learning settings**: Hyperband assumes the problem instance is fixed, and aims to accelerate the random search of hyperparameter configuration for that problem instance with constrained budgets (formulated as a pure-exploration non-stochastic infinite-armed bandit). In contrast, our results assume a problem distribution D (data-driven setting), and bounds the sample complexity of learning a good hyperparameter for the problem distribution D.

**Conclusion.** The Hyperband paper and our work do not compete but complement each other, as the two papers see the hyperparameter tuning problem from different perspectives and our results cannot be compared to theirs.

## B    ON THE CHALLENGE AND NOVELTY OF TECHNIQUES INTRODUCED IN THIS PAPER.

We note that the main and foremost contribution (Lemma 4.2, Theorem 5.1) in this paper is a new technique for analyzing the model hyperparameter tuning in data-driven setting, where the dual utility function of both parameter and hyperparameter $f_{\boldsymbol{x}}(\alpha, \boldsymbol{w})$ admits a specific piecewise polynomial structure. In this section, we will make an in-depth comparison between our setting and settings in prior works in data-driven algorithm hyperparameter tuning, and discuss why our setting is more challenging and requires novel techniques to analyze.

**Novel challenges.**    We note that our setting requires significant technical novelty relative to prior work in data-driven algorithm design. *As far as we know, most prior works on statistical data-driven algorithm design falls into two categories:*

1. The hyperparameter tuning process does not involve the parameter $w$, meaning that given a hyperparameter $\alpha$, the behavior of the algorithm is fixed. Some concrete examples include tuning hyperparameters of hierarchical clustering algorithms (Balcan et al., 2017; 2020a), branch and bound (B&B) algorithms for (mixed) integer linear programming (Balcan et al., 2018a; 2022b), and graph-based semi-supervised learning (Balcan & Sharma, 2021). The typical approach is to show that the utility function $u_{\boldsymbol{x}}^*(\alpha)$ admits specific piecewise structure of $\alpha$, typically piecewise polynomial and rational.

2. The hyperparameter tuning process involves the parameter $\boldsymbol{w}$, for example in tuning regularization hyperparameters in linear regression. However, here the optimal parameter $\boldsymbol{w}^*(\alpha)$ can either have a close analytical form in terms of the hyperparameter $\alpha$ (Balcan et al., 2022a), or can be easily approximated in terms of $\alpha$ with bounded error (Balcan et al., 2024b).

However, in our setting, the utility function $u_{\boldsymbol{x}}^*(\alpha)$ is defined via an optimization problem $u_{\boldsymbol{x}}^*(\alpha) = \max_w f_{\boldsymbol{x}}(\alpha, \boldsymbol{w})$, where $f_{\boldsymbol{x}}(\alpha, \boldsymbol{w})$ admits a piecewise polynomial structure. This involves the param-

eter $\boldsymbol{w}$ so it does not belong to the first case, and also it is not clear how to use the second approach either. This emphasizes that our problem and requires the development of novel techniques.

**New techniques.** Two general approaches are known from prior work to establish a generalization guarantee for $\mathcal{U}$.

1. The first approach is to establish Pseudo-dimension bound for $\mathcal{U}$ via alternatively analyzing the Pseudo/VC-dimension of the piece and boundary function classes, derived when establishing the piecewise structure of $u_{\boldsymbol{x}}^*(\alpha)$ (following the Theorem 3.3 (Balcan et al., 2021a)). *We build on this ideas, however, in order to apply it we need significant innovation to analyze the structure of the function $u_{\boldsymbol{x}}^*$ in our case.*

2. The second approach is specialized to the case where the computation of $u_x^*(\alpha)$ can be described as the GJ algorithm (Bartlett et al., 2022), where we can do four basic operators $(+, -, \times, \div)$ and the conditional statements. However, it is obviously not applicable to our case as well due to the use of a max operation in the definition.

*As mentioned above, we follow the first approach though we have* to develop new techniques to analyze our setting. Here, we choose to analyze $u_x^*(\alpha)$ via indirectly analyzing $f_{\boldsymbol{x}}(\alpha, \boldsymbol{w})$, which is some case shown to admit piecewise polynomial structure. To do that, we have to develop the following things:

1. The connection between number of discontinuities and local maxima and generalization guarantee of $\mathcal{U}$.

2. The approach to upper-bound the number of discontinuities and local extrema of $u_{\boldsymbol{x}}^*(\alpha)$. This is done via using ideas from differential/algebraic geometry, and constrained optimization. We note that even the tools from differential geometry are not readily available, but we have to identify and develop those tools (e.g. Monotonic curves and its properties, see Definition 12 and Lemma 18).

That corresponds to the main contribution of our papers (Lemma 4.2, Theorem 5.2). We then demonstrate the applicability of our results to two concrete problems in hyperparameter tuning in machine learning (Section 6).

**The need for the ERM oracle.** In our work, we assume the ERM oracle when defining the function $u_{\boldsymbol{x}}^*(\alpha) = \max_w f_{\boldsymbol{x}}(\alpha, w)$. This is the important first step for a clean theoretical formulation, allowing $u_{\boldsymbol{x}}^*(\alpha)$ to have deterministic behavior given a hyperparameter $\alpha$, and independent of the optimization technique.

## C  ADDITIONAL BACKGROUND ON LEARNING THEORY

**Definition 2** (Shattering and pseudo-dimension, Pollard (2012))**.** Let $\mathcal{U}$ be a real-valued function class, of which each function takes input in $\mathcal{X}$. Given a set of inputs $S = (\boldsymbol{x}_1, \ldots, \boldsymbol{x}_N) \subset \mathcal{X}$, we say that $S$ is *pseudo-shattered* by $\mathcal{H}$ if there exists a set of real-valued thresholds $r_1, \ldots, r_N \in \mathbb{R}$ such that

$$|\{(\text{sign}(u(\boldsymbol{x}_1) - r_1), \ldots, \text{sign}(u(\boldsymbol{x}_N) - r_N)) \mid u \in \mathcal{U}\}| = 2^N.$$

The pseudo-dimension of $\mathcal{H}$, denoted as $\text{Pdim}(\mathcal{U})$, is the maximum size $N$ of a input set that $\mathcal{H}$ can shatter.

**Theorem C.1** (Pollard (2012))**.** *Given a real-valued function class $\mathcal{U}$ whose range is $[0, H]$, and assume that $\text{Pdim}(\mathcal{U})$ is finite. Then, given any $\delta \in (0, 1)$, and any distribution $\mathcal{D}$ over the input space $\mathcal{X}$, with probability at least $1 - \delta$ over the drawn of $S \sim \mathcal{D}^n$, we have*

$$\left| \frac{1}{n} \sum_{i=1}^N u(\boldsymbol{x}_i) - \mathbb{E}_{\boldsymbol{x} \sim \mathcal{D}}[u(\boldsymbol{x})] \right| \leq O\left( H \sqrt{\frac{1}{N} \left( \text{Pdim}(\mathcal{U}) + \ln \frac{1}{\delta} \right)} \right).$$

**Theorem C.2** (Pollard (2012))**.** *Given a real-valued function class $\mathcal{U}$ whose range is $[0, H]$, and assume that $\text{Pdim}(\mathcal{U})$ is finite. Then for any $\epsilon > 0$ and $\delta \in (0, 1)$, for any distribution $\mathcal{D}$ and for any set $S$ of $m = O\left( \frac{H^2}{\epsilon^2} (\text{Pdim}(\mathcal{U}) + \log \frac{1}{\delta}) \right)$ samples drawn from $\mathcal{D}$, w.p. at least $1 - \delta$, we have*

$$|L_S^m(f) - L_{\mathcal{D}}(f)| < \epsilon, \quad \text{for all } f \in \mathcal{F}.$$

**Definition 3** (Rademacher complexity, Wainright (2019)). Let $\mathcal{F}$ be a real-valued function class mapping form $\mathcal{X}$ to $[0,1]$. For a set of inputs $S = \{\boldsymbol{x}_1, \boldsymbol{x}_m\}$, we define the *empirical Rademacher complexity* $\hat{\mathscr{R}}_S(\mathcal{F})$ as

$$\hat{\mathscr{R}}_S(\mathcal{F}) = \frac{1}{m}\mathbb{E}_{\epsilon_1,\ldots,\epsilon_m\sim\text{i.i.d unif}\pm1}\left[\sup_{f\in\mathcal{F}}\sum_{i=1}^m\epsilon_i f(\boldsymbol{x}_i)\right].$$

We then define the *Rademacher complexity* $\mathscr{R}_{\mathcal{D}^m}$, where $\mathcal{D}$ is a distribution over $\mathcal{X}$, as

$$\mathscr{R}_{\mathcal{D}^m}(\mathcal{F}) = \mathbb{E}_{S\sim\mathcal{D}^m}[\hat{\mathscr{R}}_S(\mathcal{F})].$$

Furthermore, we define

$$\mathscr{R}_m(\mathcal{F}) = \sup_{S\in\mathcal{X}^m}\hat{\mathscr{R}}_S(\mathcal{F}).$$

The following lemma provides an useful result that allows us to relate the empirical Rademacher complexity of two function classes when the infinity norm between their corresponding dual utility functions is upper-bounded.

**Lemma C.3** (Balcan et al. (2020b)). *Let $\mathcal{F} = \{f_{\boldsymbol{r}} \mid \boldsymbol{r} \in \mathcal{R}\}$ and $\mathcal{G} = \{g_{\boldsymbol{r}} \mid \boldsymbol{r} \in \mathcal{R}\}$ consist of function mapping from $\mathcal{X}$ to $[0,1]$. For any $S \subseteq \mathcal{X}$, we have*

$$\hat{\mathscr{R}}_S(\mathcal{F}) \le \hat{\mathscr{G}}_S(\mathcal{G}) + \frac{1}{|S|}\sum_{\boldsymbol{x}\in S}\|f_{\boldsymbol{x}}^* - g_{\boldsymbol{x}}^*\|_\infty.$$

The following theorem establishes a connection between pseudo-dimension and Rademacher complexity.

**Lemma C.4** (Shalev-Shwartz & Ben-David (2014)). *Let $\mathcal{F}$ is a bounded function class. Then* $\mathscr{R}_m(\mathcal{F}) = \mathcal{O}\left(\sqrt{\frac{\text{Pdim}(\mathcal{F})}{m}}\right)$. *Here $\mathscr{R}_m(\mathcal{F}) = \sup_{S\in\mathcal{X}^m}\hat{\mathscr{R}}_S(\mathcal{F})$.*

The following classical result demonstrates the connection between uniform convergence and learnability with an ERM learner.

**Theorem C.5** (Shalev-Shwartz & Ben-David (2014)). *If $\mathcal{F}$ has a uniform convergence guarantee with $s(\epsilon, \delta)$ samples then it is PAC learnable with ERM and $s(\epsilon/2, \delta)$ samples.*

*Proof.* For $S = \{\boldsymbol{x}_1, \ldots, \boldsymbol{x}_N\}$, let $L_S(f) = \frac{1}{n}\sum_{i=1}^n f(\boldsymbol{x}_i)$, and $L_\mathcal{D}(f) = \mathbb{E}_{\boldsymbol{x}\sim\mathcal{D}}[f(\boldsymbol{x})]$ for any $f \in \mathcal{F}$. Since $\mathcal{F}$ is uniform convergence with $s(\epsilon, \delta)$ samples, w.p. at least $1 - \delta$ for all $f \in \mathcal{F}$, we have $|L_S(f) - L_\mathcal{D}(f)| \le \epsilon$ for any set $S$ with the number of elements $m \ge s(\epsilon, \delta)$. Let $f_{ERM} \in \arg\min_{f\in\mathcal{F}} L_S(f)$ be the hypothesis outputted by the ERM learner, and $f^* \in \arg\min_{f\in\mathcal{F}} L_\mathcal{D}(f)$ be the best hypothesis. We have

$$L_\mathcal{D}(f_{ERM}) \le L_S(f_{ERM}) + \frac{\epsilon}{2} \le L_S(f^*) + \frac{\epsilon}{2} \le L_\mathcal{D}(h^*) + \epsilon,$$

which concludes the proof. $\qquad\square$

# D OMITTED PROOFS FOR SECTION 3

*Lemma 3.1* (restated). Let $h$ be a piecewise continuous function which has at most $B_1$ discontinuity points, and has at most $B_2$ local maxima. Then $h$ has at most $\mathcal{O}(B_1 + B_2)$ oscillations.

*Proof.* For any $z \in \mathbb{R}$, consider the function $g(\rho) = \mathbb{I}_{\{h(\rho)\ge z\}}$. By definition, any discontinuity points of $g(\rho)$ is a root of the equation $h(\rho) = z$. Therefore, it suffices to give an upper-bound for the number of roots that the equation $h(\rho) = z$ can have.

Let $\rho_1 < \rho_2 < \cdots < \rho_N < \rho_{N+1}$ be the discontinuity points of $h$, where $N \le B_1$ from assumption. For convenience, let $\rho_0 = -\infty$ and $\rho_{N+1} = \infty$. For any $i = 1, \ldots, N$, consider an interval $I_i = (\rho_i, \rho_i + 1)$ over which the function $h$ is continuous. Assume that there are $E_i$ local maxima of the function $h$ in between the interval $I_i$, meaning that there are at most $2E_i + 1$ local extrema, we now claim that there are at most $2E_i + 2$ roots of $h(\rho) = z$ in between $I_i$. We prove by contradiction: assume that $\rho_1^* < \rho_2^* < \cdots < \rho_{2E_i+3}^*$ are $2E_i + 3$ roots of the equation $h(\rho) = z$, and there is no other root in between. We have the following claim:

- Claim 1: there is at least 1 local extrema in between $(\rho_j^*, \rho_{j+1}^*)$. Since $h$ has finite number of local extrema, meaning that $h$ cannot be constant over $[\rho_j^*, \rho_{j+1}^*]$. Therefore, there exists some $\rho' \in (\rho_j^*, \rho_{j+1}^*)$ such that $h(\rho') \neq z$, and note that $z = h(\rho_j^*) = h(\rho_{j+1}^*)$. Since $h$ is continuous over $[\rho_j^*, \rho_{j+1}^*]$, from extreme value theorem (Theorem F.11), $h$ (when restricted to $[\rho_j^*, \rho_{j+1}^*]$) reaches minima and maxima over $[\rho_j^*, \rho_{j+1}^*]$. However, since there exists $\rho'$ such that $h(\rho') \neq z$, then $h$ has to achieve minima or maxima in the interior $(\rho_j^*, \rho_{j+1}^*)$. That is also a local extrema of $h$.

- Claim 2: there are at least $2E_i + 2$ local extrema in between $(\rho_1^*, \rho_{E_i+2}^*)$. This claim follows directly from Claim 1.

Claim 2 leads to a contradiction. Therefore, there are at most $2E_i + 2$ roots in between the interval $I_i$. which implies there are $\sum_{i=0}^{N} 2E_i + 2N$ roots in the intervals $I_i$ for $i = 1, \ldots, N$. Note that $\sum_{i=0}^{N} E_i \leq B_2$, $N \leq B_1$ by assumption, and each discontinuity points could also be a root of $h(\rho) = z$, we conclude that there are at most $\mathcal{O}(B_1 + B_2)$ roots of the equation $h(\rho) = z$, for any $z$. □

*Lemma* Lemma 3.3 (restated). Consider a real-valued function class $\mathcal{U} = \{u_\rho : \mathcal{X} \to \mathbb{R} \mid \rho \in \mathbb{R}\}$, of which each dual function $u_{\boldsymbol{x}}^*(\rho)$ is piecewise constant with at most $B$ discontinuities. Then $\mathrm{Pdim}(\mathcal{U}) = \mathcal{O}(\ln B)$.

*Proof.* Consider a dual function $u_{\boldsymbol{x}}^*(\rho)$ which is a piecewise constant function with at most $B$ discontinuities. $\mathbb{I}_{\{u_{\boldsymbol{x}}^*(\rho) \geq z\}}$ is piecewise continuous with at most $B$ continuities for any threshold $z \in \mathbb{R}$. We will show that by contradiction, assume that there exists $z \in \mathbb{R}$ such that $\mathbb{I}_{\{u_{\boldsymbol{x}}^*(\rho) \geq z\}}$ has $N$ discontinuities, where $N \geq B + 1$. Since $u_{\boldsymbol{x}}^*(\rho)$ is piecewise constant, any discontinuities of $\mathbb{I}_{\{u_{\boldsymbol{x}}^*(\rho) \geq z\}}$ is also a discontinuity of $u_{\boldsymbol{x}}^*(\rho)$, meaning that $u_{\boldsymbol{x}}^*(\rho)$ has at least $N$ discontinuities, which leads to a contradiction. Therefore, we conclude that $u_{\boldsymbol{x}}^*(\rho)$ has at most $B$ oscillations, and then $\mathrm{Pdim}(\mathcal{H}) = \mathcal{O}(\log(B))$ following Theorem 2.1. □

# E  ADDITIONAL RESULTS AND OMITTED PROOFS FOR SECTION 4

## E.1  OMITTED PROOFS

In this section, we will present the detailed proof for Theorem 4.1.

*Lemma* 4.1 (restated). Assume that the piece functions $f_i(\alpha, \boldsymbol{w})$ is constant for all $i \in [N]$. Then $u_{\boldsymbol{x}}^*(\alpha)$ has $\mathcal{O}(N)$ discontinuity points, partitioning $\mathcal{A}$ into at most $\mathcal{O}(N)$ regions. In each region, $u_{\boldsymbol{x}}^*(\alpha)$ is a constant function.

*Proof.* For each connected set $R_{\boldsymbol{x},i}$ corresponding to a piece function $f_{\boldsymbol{x},i}(\alpha, \boldsymbol{w}) = c_i$, let

$$\alpha_{R_i, \inf} = \inf_\alpha\{\alpha : \exists \boldsymbol{w}, (\alpha, \boldsymbol{w}) \in R_i\}, \quad \alpha_{R_i, \sup} = \sup_\alpha\{\alpha : \exists \boldsymbol{w}, (\alpha, \boldsymbol{w}) \in R_i\}.$$

There are $N$ connected components, corresponding to $\mathcal{O}(N)$ such points. Reordering those points and removing duplicate points as $\alpha_{\min} = \alpha_0 < \alpha_1 < \alpha_2 < \cdots < \alpha_t = \alpha_{\max}$, where $t = \mathcal{O}(N)$ we claim that for any interval $I_i = (\alpha_i, \alpha_{i+1})$ where $i = 0, \ldots, t-1$, the function $g_{\boldsymbol{x}}(\alpha)$ remains constant.

Consider the any interval $I_i$. By the construction above of $\alpha_i$, for any $\alpha \in I_i$, there exists a *fixed* set of regions $\mathbf{R}_{I_i} = \{R_{I_i,1}, \ldots, R_{I_i,n}\} \subseteq \mathcal{P}_{\boldsymbol{x}} = \{R_{\boldsymbol{x},1}, \ldots, R_{\boldsymbol{x},N}\}$, such that for any connected set $R \in \mathbf{R}_{I_i}$, there exists $\boldsymbol{w}$ such that $(\alpha, \boldsymbol{w}) \in R$. Besides, for any $R \notin \mathbf{R}_{I_i}$, there does not exist $\boldsymbol{w}$ such that $(\alpha, \boldsymbol{w}) \in R$. This implies that for any $\alpha \in I_i$, we can write $u_{\boldsymbol{x}}^*(\alpha)$ as

$$u_{\boldsymbol{x}}^*(\alpha) = \sup_{\boldsymbol{w} \in \mathcal{W}} f_{\boldsymbol{x}}(\alpha, \boldsymbol{W}) = \sup_{R \in \mathbf{R}_{I_i}} \sup_{\boldsymbol{w}:(\alpha, \boldsymbol{W}) \in R} f_{\boldsymbol{x}}(\alpha, \boldsymbol{w}) = \max_{c \in \boldsymbol{C}_{I_i}} c,$$

where $\boldsymbol{C}_{I_i} = \{c_R \mid R \in \mathbf{R}_{I_i}\}$ contains the constant value that $f_{\boldsymbol{x}}(\alpha, \boldsymbol{W})$ takes over $R$. Since the set $\boldsymbol{C}_{I_i}$ is fixed, $u_{\boldsymbol{x}}^*(\alpha)$ remains constant over $I_i$.

Hence, we conclude that over any interval $I_i = (\alpha_i, \alpha_{i+1})$, for $i = 1, \ldots, t-1$, the function $u_{\boldsymbol{x}}^*(\alpha)$ remains constant. Therefore, there are only the points $\alpha_i$, for $i = 0, \ldots, t-1$, at which the function $u_{\boldsymbol{x}}^*$ is not continuous. Since $t = \mathcal{O}(N)$, we have the conclusion. □

*Proof of Theorem 4.2.* From Lemma 4.1, we know that any dual utility function $u_{\boldsymbol{x}}^*$ is piecewise constant and has at most $\mathcal{O}(N)$ discontinuities. Combining with Lemma 3.3, we conclude that $\mathrm{Pdim}(\mathcal{U}) = \mathcal{O}(\log(N))$. Finally, standard learning theory result gives us the final guarantee. □

### E.2 Useful tools for bounding the number of connected components

Here, we will recall some useful tools for bounding the number of connected components created by a set of polynomial equations. It serves as an useful tool to apply our Theorem 4.1.

**Lemma E.1** (Warren (1968)). *Let $p_1, \ldots, p_m$ be real polynomials in $n$ variables, each of degree at most $d$. The number of connected components of the set $\mathbb{R}^n - \cup_{i=1}^m Z(p_i)$ is $\mathcal{O}\left(\left(\frac{md}{n}\right)^n\right)$.*

## F Additional results and omitted proofs for Section 5

### F.1 A simple case: hyperparameter tuning with a single parameter

We provide intuition for our novel proof techniques by first considering a simpler setting. We first consider the case where there is a single parameter and only one piece function. That is, we assume that $N = 1$ and $M = 0$. We first present a structural result for the dual function class $\mathcal{U}^*$, which establishes that any function $u_{\boldsymbol{x}}^*$ in $\mathcal{U}^*$ is piecewise continuous with at most $O(\Delta_p^2)$ pieces. Furthermore, we show that there are at most $O(\Delta_p^3)$ oscillations in $u_{\boldsymbol{x}}^*$ which implies a bound on the pseudo-dimension of $\mathcal{U}^*$ using results in Section 3.

Our proof approach is summarized as follows. We note that the supreme over $\boldsymbol{w} \in \mathcal{W}$ in the definition of $u_{\boldsymbol{x}}^*$ can only be achieved at a domain boundary or along the derivative $h_x(\alpha, \boldsymbol{w}) = \frac{\partial f_x(\alpha, \boldsymbol{w})}{\partial \boldsymbol{w}} = 0$, which is an algebraic curve. We partition this algebraic curve into *monotonic arcs*, which intersect $\alpha = \alpha_0$ at most once for any $\alpha_0$. Intuitively, a point of discontinuity of $u_{\boldsymbol{x}}^*$ can only occur when the set of monotonic arcs corresponding to a fixed value of $\alpha$ changes as $\alpha$ is varied, which corresponds to $\alpha$-extreme points of the monotonic arcs. We use Bezout's theorem to upper bound these extreme points of $h_x(\alpha, \boldsymbol{w}) = 0$ to obtain an upper bound on the number of pieces of $u_{\boldsymbol{x}}^*$. Next, we seek to upper bound the number of local extrema of $u_{\boldsymbol{x}}^*$ to bound its oscillating behavior within the continuous pieces. To this end, we need to examine the behavior of $u_{\boldsymbol{x}}^*$ along the algebraic curve $h_x(\alpha, \boldsymbol{w}) = 0$ and use the Lagrange's multiplier theorem to express the locations of the extrema as intersections of algebraic varieties (in $\alpha, \boldsymbol{w}$ and the Lagrange multiplier $\lambda$). Another application of Bezout's theorem gives us the deisred upper bound on the number of local extrema of $u_{\boldsymbol{x}}^*$.

**Lemma F.1.** *Let $d_{\mathcal{W}} = d_{\mathcal{A}} = 1$ and $N = 1, M = 0$. Assume that $(\alpha, w) \in R = [\alpha_{\min}, \alpha_{\max}] \times [w_{\min}, w_{\max}]$. Then for any function $u_{\boldsymbol{x}}^* \in \mathcal{U}^*$, we have*

> *(a) The hyperparameter domain $\mathcal{A} = [\alpha_{\min}, \alpha_{\max}]$ can be partitioned into $\mathcal{O}(\Delta_p^2)$ intervals such that $u_{\boldsymbol{x}}^*$ is a continuous function over any interval in the partition.*

> *(b) $u_{\boldsymbol{x}}^*$ has $\mathcal{O}(\Delta_p^2)$ local maxima for any $\boldsymbol{x}$.*

*Proof.* (a) Denote $h_x(\alpha, w) = \frac{\partial f_x(\alpha, w)}{\partial \boldsymbol{w}}$. From assumption, $f_x(\alpha, w)$ is a polynomial of $\alpha$ and $\boldsymbol{w}$, therefore it is differentiable everywhere in the compact domain $[\alpha_{\min}, \alpha_{\max}] \times [w_{\min}, w_{\max}]$. Consider any $\alpha_0 \in [\alpha_{\min}, \alpha_{\max}]$, we have $\{(\alpha, w) \mid \alpha = \alpha_0\} \cap [\alpha_{\min}, \alpha_{\max}]$ is an intersection of a hyperplane and a compact set, hence it is also compact. Therefore, from Fermat's interior extremum theorem (Lemma F.8), for any $\alpha_0$, $f_x(\alpha_0, w)$ attains the local maxima $w$ either in $w_{\min}, w_{\max}$, or for $w \in (w_{\min}, w_{\max})$ such that $h_x(\alpha_0, w) = 0$. Note that from assumption, $f_x(\alpha, w)$ is a polynomial of degree at most $\Delta_p$ in $\alpha$ and $w$. This implies $h_x(\alpha, w)$ is a polynomial of degree at most $\Delta_p - 1$.

Denote $C_x = V(h_x)$ the zero set of $h_x$ in $R$. For any $\alpha_0$, $C_x$ intersects the line $\alpha = \alpha_0$ in at most $\Delta_p - 1$ points by Bezout's theorem. This implies that, for any $\alpha$, there are at most $\Delta_p + 1$ candidate

values of $w$ which can possibly maximize $f_x(\alpha, w)$, which can be either $w_{\min}, w_{\max}$, or on some point in $C_x$. We then define the *candidate arc set* $\mathcal{C} : \mathcal{A} \to \mathcal{M}_\alpha(C_x)$ as the function that maps $\alpha_0 \in \mathcal{A}$ to the set of all maximal $\alpha$-monotonic arcs of $C_x$ (12, informally arcs that intersect any line $\alpha = \alpha_0$ at most once) that intersect with $\alpha = \alpha_0$. By the argument above, we have $|\mathcal{C}(\alpha)| \le \Delta_p + 1$ for any $\alpha$.

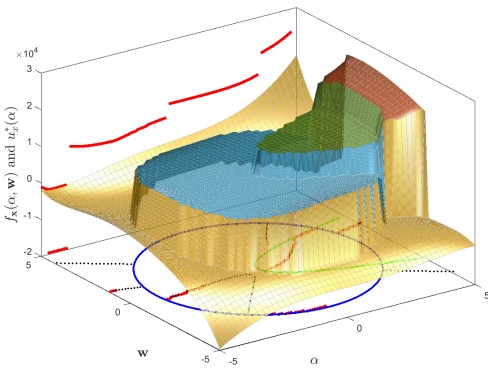

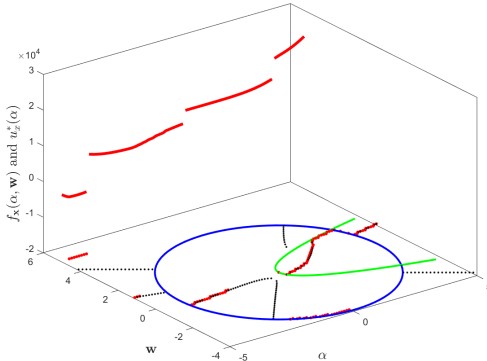

(a) The piecewise structure of $u_x^*(\alpha)$ and piecewise polynomial surface of $f_x(\alpha, w)$ in sheer view.

(b) Removing the surface $f_x(\alpha, w)$ for better view of $u_x^*(\alpha)$, the boundaries, and the derivative curves.

Figure 4: A demonstration of the proof idea for Theorem 5.1 in 2D ($w \in \mathbb{R}$). Here, the domain of $f_x^*(\alpha, w)$ is partitioned into four regions by two boundaries: a circle (blue line) and a parabola (green line). In each region $i$, the function $f_x(\alpha, w)$ is a polynomial $f_{x,i}(\alpha, w)$, of which the derivative curve $\frac{\partial f_{x,i}}{\partial w} = 0$ is demonstrated by the black dot in the plane of $(\alpha, w)$. The value of $u_x^*(\alpha)$ is demonstrated in the red line, and the red dots in the plane $(\alpha, w)$ corresponds to the position where $f_x(\alpha, w) = u_x^*(\alpha)$. We can see that it occurs in either the derivative curves or in the boundary. Our goal is to leverage this property to control the number of discontinuities and local maxima of $u_x^*(\alpha)$, which can be converted to the generalization guarantee of the utility function class $\mathcal{U}$.

We now have the following claims: (1) $\mathcal{C}$ is a piecewise constant function, and (2) any point of discontinuity of $u_x^*$ must be a point of discontinuity of $\mathcal{C}$. For (1), we will show that $\mathcal{C}$ is piecewise constant, with the piece boundaries contained in the set of $\alpha$-extreme points[1] of $C_x$ and the intersection points of $C_x$ with boundary lines $w = w_{\min}, w_{\max}$. Note that if $C_x$ has any components consisting of axis-parallel straight lines $\alpha = \alpha_1$, we do not consider these components to have any $\alpha$-extreme points, and the corresponding discontinuities (if any) are counted in the intersections of $C_x$ with the boundary lines. Indeed, for any interval $I = (\alpha_1, \alpha_2) \subseteq \mathcal{A}$, if there is no $\alpha$-extreme point of $C_x$ in the interval, then the set of arcs $\mathcal{C}(\alpha)$ is fixed over $I$ by Definition 12. Next, we will prove (2) via an equivalent statement: assume that $\mathcal{C}$ is continuous over an interval $I \subseteq \mathcal{A}$, we want to prove that $u_x^*$ is also continuous over $I$. Note that if $\mathcal{C}$ is continuous over $I$, then $u_x^*(\alpha)$ involves a maximum over a fixed set of $\alpha$-monotonic arcs of $C_x$, and the straight lines $w = w_{\min}, w_{\max}$. Since $f_x$ is continuous along these arcs, so is the maximum $u_x^*$.

The above claim implies that the number of discontinuity points of $C_x$ upper-bounds the number of discontinuity points of $u_x^*(\alpha)$. Note that $\alpha$-extreme points $C_x$ satisfies the following equalities: $h_x = 0$ and $\frac{\partial h_x}{\partial w} = 0$. By Bezout's theorem and from assumption on the degree of the polynomial $f_x$, we conclude that there are at most $(\Delta_p - 1)(\Delta_p - 2) = \mathcal{O}(\Delta_p^2)$ $\alpha$-extreme points of $C_x$. Moreover, there are $\mathcal{O}(\Delta_p)$ intersection points between $C_x$ and the boundary lines $w = w_{\min}, w_{\max}$. Thus, the total discontinuities of $\mathcal{C}$, and therefore $u_x^*$, are $\mathcal{O}(\Delta_p^2)$.

(b) Consider any interval $I$ over which the function $u_x^*(\alpha)$ is continuous. By Corollary F.5 and Proposition F.14, it suffices to bound the number of elements of the set of local maxima of $f_x$ along the algebraic curve $C_x$ and the straight lines $w = w_{\min}, w_{\max}$.

---

[1] An $\alpha$-extreme point of an algebraic curve $C$ is a point $p = (\alpha, W)$ such that there is an open neighborhood $N$ around $p$ for which $p$ has the smallest or largest $\alpha$-coordinate among all points $p' \in N$ on the curve.

To bound the set of bound the number of elements of the set of local maxima of $f_x$ along the algebraic curve $C_x$, consider the Lagrangian

$$\mathcal{L}(\alpha, w, \lambda) = f_x(\alpha, w) + \lambda h_x(\alpha, w).$$

From Lagrange's multiplier theorem, any local maxima of $f_x$ along the algebraic curve $C_x$ is also a critical point of $\mathcal{L}$, which satisfies the following equations

$$\frac{\partial \mathcal{L}}{\partial \alpha} = \frac{\partial f_x}{\partial \alpha} + \lambda \frac{\partial h_x}{\partial \alpha} = \frac{\partial f_x}{\partial \alpha} + \lambda \frac{\partial^2 f_x}{\partial \alpha \partial w} = 0,$$

$$\frac{\partial \mathcal{L}}{\partial w} = \frac{\partial f_x}{\partial w} + \lambda \frac{\partial h_x}{\partial w} = \frac{\partial f_x}{\partial w} + \lambda \frac{\partial^2 f_x}{\partial w^2} = 0,$$

$$\frac{\partial \mathcal{L}}{\partial \lambda} = h_x = \frac{\partial f_x}{\partial w} = 0.$$

Plugging $\frac{\partial f_x}{\partial w} = 0$ into the second equation above, we get that either $\lambda = 0$ or $\frac{\partial^2 f_x}{\partial w^2} = 0$. In the former case, the first equation implies $\frac{\partial f_x}{\partial \alpha} = 0$. Thus, we consider two cases for critical points of $\mathcal{L}$.

**Case** $\frac{\partial \mathbf{f_x}}{\partial \mathbf{w}} = \mathbf{0}, \frac{\partial \mathbf{f_x}}{\partial \alpha} = \mathbf{0}$. By Bezout's theorem these algebraic curves intersect in at most $\Delta_p^2$ points, unless the polynomials $\frac{\partial f_x}{\partial w}, \frac{\partial f_x}{\partial \alpha}$ have a common factor. In this case, we can write $\frac{\partial f_x}{\partial w} = g(\alpha, w)g_1(\alpha, w)$ and $\frac{\partial f_x}{\partial \alpha} = g(\alpha, w)g_2(\alpha, w)$ where $g = \gcd\left(\frac{\partial f_x}{\partial w}, \frac{\partial f_x}{\partial \alpha}\right)$ and $g_1, g_2$ have no common factors. Now for any point on $g(\alpha, w) = 0$, we have both $\frac{\partial f_x}{\partial w} = 0, \frac{\partial f_x}{\partial \alpha} = 0$ and therefore $f_x$ is constant along the curve (and therefore has no local maxima). By Bezout's theorem, $g_1, g_2$ intersect in at most $\deg(g_1)\deg(g_2) \leq \Delta_p^2$ points. Thus, the number of local maxima of $u_x^*$ that correspond to this case is $\mathcal{O}(\Delta_p^2)$.

**Case** $\frac{\partial \mathbf{f_x}}{\partial \mathbf{w}} = \mathbf{0}, \frac{\partial^2 \mathbf{f_x}}{\partial \mathbf{w^2}} = \mathbf{0}$. This is essentially the $\alpha$-extreme points computed above, and are at most $\mathcal{O}(\Delta_p^2)$.

Similarly, the equations $f_x(\alpha, w_{\min}) = 0$ and $f_x(\alpha, w_{\max}) = 0$ also have at most $\Delta_p$ solutions each. Therefore, we conclude that the number of local maxima of $u_x^*$ can be upper-bounded by $\mathcal{O}(\Delta_p^2)$. □

**Theorem F.2.** $\text{Pdim}(\mathcal{U}^*) = \mathcal{O}(\log \Delta_p)$.

*Proof.* From Theorem F.1, we conclude that $u_x^*$ has at most $\mathcal{O}(\Delta_p^2)$ oscillations for any $u_x^* \in \mathcal{U}^*$. Therefore, from Theorem 3.3, we conclude that $\text{Pdim}(\mathcal{U}^*) = \mathcal{O}(\log \Delta_p)$. □

**Challenges of generalizing the one-dimensional parameter, single region to high-dimensional parameter, multiple regions.** Recall that in the simple setting above, we assume that $f_x(\alpha, w)$ is a polynomial in the whole domain $[\alpha_{\min}, \alpha_{\max}] \times [w_{\min}, w_{\max}]$. In this case, our approach is to characterize the manifold on which the optimal solution of $\max_{w:(\alpha,w)\in R} f_x(\alpha, w)$ lies, as $\alpha$ varies. We then use algebraic geometry tools to upper bound the number of discontinuity points and local extrema of $u_x^*(\alpha) = \max_{w:(\alpha,w)\in R} f_x(\alpha, w)$, leading to a bound on the pseudo-dimension of the utility function class $\mathcal{U}$ by using our proposed tools in Section 3. However, to generalize this idea to high-dimensional parameters and multiple regions is a much more challenging due to the following issues: (1) handling the analysis of multiple pieces by accounting for polynomial boundary functions is tricky as the $w^*$ maximizing $f_x(\alpha, w)$ can switch between pieces as $\alpha$ is varied, (2) characterizing the optimal solution $\max_{w:(\alpha,w)\in R} f_x(\alpha, w)$ is not trivial and typically requiring additional assumptions to ensure a general position property is achieved, and care needs to be taken to ensure that the assumptions are not too strong and complicated, (3) generalizing the monotonic curve notion to high-dimensions is not trivial and requires a much more complicated analysis invoking tools from differential geometry, and (4) controlling the number of discontinuities and local maxima of $u_x^*$ over the high-dimensional monotonic curves requires more sophisticated techniques.

We now present preliminaries background and our supporting results for Lemma 5.

### F.2  GENERAL SUPPORTING RESULTS

In this section, we recall some elementary results which are crucial in our analysis. The following lemma says that the point-wise maximum of continuous functions is also a continuous function.

**Lemma F.3.** *Let $f_i : \mathcal{X} \to \mathbb{R}$, where $i \in [N]$ be a continuous function over $\mathcal{X}$, and let $f(x) = \max_{i \in [N]}\{f_i(x)\}$. Then we have $f(x)$ is a continuous function over $\mathcal{X}$.*

*Proof.* In the case $N = 2$, we can rewrite $f(x)$ as

$$f(x) = \frac{f_1(x) + f_2(x)}{2} + \frac{1}{2}\left|f_1(x) - f_2(x)\right|,$$

which is sum of continuous function. Hence, $f(x)$ is continous. Assume the claim holds for $N = k$, we then claim that it also holds for $N = k + 1$ by rewriting $f(x)$ as

$$f(x) = \max\{\max_{i \in [k]}\{f_i(x)\}, f_{k+1}(x)\}.$$

Therefore, the claim is proven by induction. $\qquad\square$

The following results are helpful when we want to bound the number of local extrema of point-wise maximum of differentiable functions. In particular, we show that the local extrema of $f(x) = \max\{f_i(x)\}_{i=1}^n$ is the local extrema of one of the functions $f_i(x)$.

**Lemma F.4** (Rockafellar & Wets (2009)). *Let $X$ be a finite-dimensional real Euclidean space and $g_i : X \to \mathbb{R}$ for $i \in [N]$ be continuously differential functions on $X$. Define the function $g(\boldsymbol{x}) = \max_{i \in [N]}\{g_i(\boldsymbol{x})\}$. Let $\overline{\boldsymbol{x}}$ be a point in the interior of $X$, and let $\mathcal{I}_{\overline{\boldsymbol{x}}} = \{i \in [N] \mid g_i(\overline{\boldsymbol{x}}) = g(\overline{\boldsymbol{x}})\}$. Then, for any $d \in X$, the directional derivative of $g$ along the direction $d$ is $g'(\overline{\boldsymbol{x}}; d) = \max_{i \in \mathcal{I}_{\overline{\boldsymbol{x}}}} \langle \nabla g_i(\overline{\boldsymbol{x}}), d \rangle$.*

**Corollary F.5.** *Let $X$ be a finite-dimensional real Euclidean space and $g_i : X \to \mathbb{R}$ for $i \in [N]$ be differential functions on $X$ with the local maxima on $X$ is given by the set $C_i$. Then the function $g(\boldsymbol{x}) = \max_{i \in [N]}\{g_i(\boldsymbol{x})\}$ has its local maxima contained in the union $\cup_{i \in [N]} C_i$.*

*Proof.* Let $\overline{\boldsymbol{x}}$ be a point in the interior of $X$, and let $\mathcal{I}_{\overline{\boldsymbol{x}}} = \{i \in [N] \mid g_i(\overline{\boldsymbol{x}}) = g(\overline{\boldsymbol{x}})\}$. Now suppose $\overline{\boldsymbol{x}} \notin \cup_{i \in [N]} C_i$. If $\mathcal{I}_{\overline{x}}$ consists of a single function $g_i$, then $\overline{x}$ is a local maximum if and only if it is local maximum of $g_i$. By Lemma F.4, if the derivative is non-zero for all $g_i$ with $i \in \mathcal{I}_{\overline{x}}$, then $g(x)$ has a positive derivative in some direction. This implies that $\overline{x}$ cannot be a local maximum in this case. $\qquad\square$

We then recall the wide-known Sauer-Shelah Lemma, which bounds the sum of finite combinatorial series under some conditions.

**Lemma F.6** (Sauer-Shelah Lemma, Sauer (1972)). *Let $1 \le k \le n$, where $k$ and $n$ are positive integers. Then*

$$\sum_{j=0}^{k} \binom{n}{j} \le \left(\frac{en}{k}\right)^k.$$

We recall the Lagrangian multipliers theorem, which allows us to give a necessary condition for the extrema of a function over a constraint.

**Theorem F.7** (Lagrangian multipliers, Rockafellar (1993)). *Let $h : \mathbb{R}^d \to \mathbb{R}$, $f : \mathbb{R}^d \to \mathbb{R}^n$ be $C^1$ functions, $C \in \mathbb{R}^d$, and $M = \{f = C\} \subseteq \mathbb{R}^d$. Assume that for all $x_0 \in M$, $\mathsf{rank}(J_{f,x}(x_0)) = n$. If $h$ attains a constrained local extremum at $a$, subject to the constraint $f = C$, then there exists $\lambda_1, \ldots, \lambda_n \in \mathbb{R}$ such that*

$$\nabla h(a) = \sum_{i=1}^{n} \lambda_i \nabla f_i(a), \quad and \quad f(a) = C,$$

*where $\lambda$ is the Lagrangian multiplier, and $a \in M$ is where $h$ attains its extremum.*

We then recall

**Lemma F.8** (Fermat's interior extremum theorem). *Let $f : D \to \mathbb{R}$, where $D \subseteq \mathbb{R}^n$ is an open set, be a function and suppose that $x_0 \in D$ is a point where $f$ has a local extremum. If $f$ is differentiable at $x_0$, then $\nabla f(x_0) = \mathbf{0}$.*

**Corollary F.9.** *The local extrema of a function $f$ on a domain $D$ occur only at boundaries, non-differentiable points, and stationary points.*

**Definition 4** (Connected components, Anthony & Bartlett (1999)). A connected components of a subset $S \subset \mathbb{R}^d$ is the maximal nonempty subset $A \subseteq S$ such that any two points of $A$ are connected by a continuous curve lying in $A$.

**Definition 5.** Let $S \subset \mathcal{A} \times \mathcal{W}$ where $\mathcal{A} \subset \mathbb{R}$ and $\mathcal{W} \subseteq \mathbb{R}^d$, and let $A$ be a connected component of $S$. We define $\alpha_{A,\inf} = \inf\{\alpha \mid \exists \boldsymbol{w}, (\alpha, \boldsymbol{w}) \in A\}$, and $\alpha_{A,\sup} = \sup\{\alpha \mid \exists \boldsymbol{w}, (\alpha, \boldsymbol{w}) \in A\}$ the $\alpha$-extreme points of $A$.

**Lemma F.10** (Warren (1968)). *Let $p$ be a polynomial in $n$ variables. If the degree of polynomial $p$ is $d$, the number of connected components of $Z(p)$ is at most $2d^n$.*

**Lemma F.11** (Extreme value theorem). *Let $f : D \to \mathbb{R}$ be a continuous function, where $D$ is a non-empty compact set, then $f$ is bounded and there exists $p, q \in D$ such that $f(p) = \sup_{x \in D} f(x)$ and $f(q) = \inf_{x \in D} f(x)$.*

## F.3 BACKGROUND ON DIFFERENTIAL GEOMETRY

In this section, we will introduce some basic terminology of differential geometry, as well as key results that we use in our proofs.

**Definition 6** (Topological manifold, Robbin & Salamon (2022)). A topological manifold is a topological space $M$ such that each point $p \in M$ has an open neighborhood $U$ which is homeomorphic to an open subset of a Euclidean space.

**Definition 7** (Smooth map, Robbin & Salamon (2022)). Let $U \subset \mathbb{R}^n$ and $V \subset \mathbb{R}^m$ be open sets. A map $f : U \to V$ is called smooth iff it is infinitely differentiable, i.e. iff all its partial derivatives

$$\partial^\alpha f = \frac{\partial^{\alpha_1 + \cdots + \alpha_n} f}{\partial x_1^{\alpha_1} \dots \partial x_n^{\alpha_n}}, \quad \alpha = (\alpha_1, \dots, \alpha_n) \in \mathbb{N}_0^n.$$

exists and continuous. Here $\mathbb{N}_0$ is the set of non-negative integers.

**Definition 8** (Regular value, Robbin & Salamon (2022)). Let $U \subset \mathbb{R}^l$ be an open set and let $f : U \to \mathbb{R}^l$ be a smooth map. A value $\boldsymbol{\epsilon} \in \mathbb{R}^l$ is called *regular value* of $f$ iff for any $\boldsymbol{x_0} \in \mathcal{U}$, $J_{f,\boldsymbol{x}}(\boldsymbol{x}_0)$ has full rank. Here, $J_{f,\boldsymbol{x}}(\boldsymbol{x}_0))$ is the Jacobian of $f$ w.r.t $\boldsymbol{x}$ and evaluated at $\boldsymbol{x_0}$.

The following theorem says that for any smooth map $f$, the set of regular value of $f$ has Lebesgue measure zero.

**Theorem F.12** (Sard's theorem, Robbin & Salamon (2022)). *Let $f : \mathbb{R}^k \to \mathbb{R}^l$ is a smooth map. Then the set of non-regular value of $f$ has Lebesgue measure zero in $\mathbb{R}^l$.*

## F.4 SUPPORTING LEMMAS

In this section, we will proof some useful tools that are crucial for our analysis.

**Definition 9** (Open set). A subset $S$ of smooth $n$-manifold $M$ is called open if for any point $x \in S$, there exists a chart $(U, \phi) \in M$ such that $p \in U$ and $\phi(U \cap S)$ is an open set in $\mathbb{R}^n$.

**Definition 10** (Neighborhood). Let $M$ be a smooth $n$-manifold, and let $x$ be a point in $M$. Then $U$ is an (open) neighborhood of $x$ in $M$ if $U$ is an open subset of $M$ that contains $x$.

**Proposition F.13.** *Let $M$ be a smooth $n$-manifold, and let $S$ be an open subset of $M$. Let $x$ be a point in $S$, and assume that $V$ be a neighborhood of $x$ in $S$. Then $x$ is also a neighborhood of $x$ in $M$.*

*Proof.* First, note that $V$ is a neighbor of $x$ in $S$, then $V$ is an open set in the subspace topology $S$. Therefore, there exists an open set $T$ in $M$ such that $V = S \cap T$. However, note that both $S$ and $T$ are open set in $M$, which implies $V$ is also an open set in $M$. And since $V$ contains $x$, meaning that $V$ is a neighborhood of $x$ in $M$. $\square$

**Proposition F.14.** *Let $\mathcal{C} = \{C_1, \ldots, C_n\}$ be a set of $\alpha$-monotonic curve (Definition 12) in the space $\mathcal{A} \times \mathcal{W}$ of $\alpha$ and $\mathbf{W}$ such that for any $\alpha \in (\alpha_1, \alpha_2)$ and any $C \in \mathcal{C}$, there is a point $\mathbf{W}$ such that $(\alpha, \mathbf{W}) \in C$. Let $u^*(\alpha) = \max_{C \in \mathcal{C}}\{f(\alpha, \mathbf{W}) : (\alpha, \mathbf{W}) \in C\}$, where $f(\alpha, \mathbf{W})$ is continuous function and bounded in the domain $\mathcal{A} \times \mathcal{W}$. Then $u^*(\alpha)$ is continuous over $(\alpha_1, \alpha_2)$, and for any local maxima $\alpha'$ of $u^*(\alpha)$, there exist a point $(\alpha', \mathbf{W}')$ that is local maxima of the function $f(\alpha, \mathbf{W})$ restricted on a monotonic curve $C \in \mathcal{C}$.*

*Proof.* We recall the most important properties of monotonic curve $C$: for any $\alpha \in (\alpha_1, \alpha_2)$, there is exactly one point $\mathbf{W}$ such that $(\alpha, \mathbf{W}) \in C$. Since $f(\alpha, \mathbf{W})$ is continuous in the domain $\mathcal{A} \times \mathcal{W}$, hence it is also continuous along the curve $C$ for any $C \in \mathcal{C}$. Therefore, $u^*(\alpha)$ is also continuous.

Now, consider any monotonic curve $C \in \mathcal{C}$ and let $u_C^*(\alpha) = f(\alpha, \mathbf{W})$ where $(\alpha, \mathbf{W}) \in C$. From the property of $C$, consider the continuous invertible mapping $I_C : (\alpha_1, \alpha_2) \to C$, where $I_C(\alpha) = (\alpha, \mathbf{W})$ for any $\alpha \in (\alpha_1, \alpha_2)$. Assume $\alpha'$ is a local extrema of $u_C^*(\alpha)$ in $(\alpha_1, \alpha_2)$, then there exists an open neighbor $V$ of $\alpha'$ such that for any $\alpha \in V$, $u_C^*(\alpha) \leq u_C^*(\alpha')$. Now, $I_C(V)$ is an open set in $C$ that contains $(\alpha', \mathbf{W}')$, hence it is an open neighbor of $(\alpha', \mathbf{W}')$. For any $(\alpha, \mathbf{W}) \in I_C(V)$, we have $f(\alpha, \mathbf{W}) = u_C^*\alpha \leq u_C^*(\alpha') = f(\alpha', \mathbf{W}')$. This means that $(\alpha', \mathbf{W}')$ is a local extrema of $f(\alpha, \mathbf{W})$ in $C$.

Finally, it suffices to give a proof for the case of 2 functions. let $u^*(\alpha) = \max\{u_{C_1}^*(\alpha), u_{C_2}^*(\alpha)\}$. We claim that any local maxima of $u^*(\alpha)$ would be a local maxima of either $u_{C_1}^*(\alpha)$ and $u_{C_2}^*(\alpha)$. Assume that $\alpha'$ is a local maxima of $u^*$, and there exists an open neighbor $V$ of $\alpha'$ in $(\alpha_1, \alpha_2)$ such that for any $\alpha \in V$, $u^*(\alpha) \leq u^*(\alpha')$. WLOG, assume that $u^*(\alpha') = u_{C_1}^*$, therefore $u_{c_1}^*(\alpha') = u^*(\alpha') \geq u^*(alpha) = \max\{u_{C_1}^*(\alpha), u_{C_2}^*(\alpha)\} \geq u_{C_1}^*(\alpha)$ for any $\alpha \in V$. This means that $\alpha'$ is a local extrema of $u_{c_1}^*(\alpha)$ in $(\alpha_1, \alpha_2)$. $\square$

F.5 MONOTONIC CURVES

**Proposition F.15.** *Let $S \subset \mathbb{R}^n$ be a bounded set in $\mathbb{R}^n$, and $f : \overline{S} \to \mathbb{R}$ be a bounded function, where $\overline{S}$ is closure of $S$. Then $\sup_S f$ exists and there is a point $x^* \in \overline{S}$ such that $f(x^*) = \sup_S f$.*

*Proof.* Since $f$ is bounded over $\overline{S}$, then $\sup_S f$ exists and let $a = \sup_S f$. By definition, for any $i > 0$, there exists $x_i \in S$ such that $|f(x_i) - a| < \frac{1}{i}$. Hence, we constructed a sequence $\{x_i\}_{i=1}^\infty$ such that $\lim_{i \to \infty} f(x_i) = a$.

Now, since $S \subset \mathbb{R}^n$ is a bounded subset in $\mathbb{R}^n$, by Bolzano-Weierstrass theorem, there exists a subsequence $\{x_i'\}_{i=1}^n \subseteq \{x_i\}_{i=1}^\infty$ such that the subsequence $\{x_i'\}_{i=1}^n$ converges. In other words, there exists $x^* \in \mathbb{R}^n$ such that $\lim_{i \to \infty} x_i' = x^*$, and since $\{x_i'\}_{i=1}^n \subset S$, then by definition $x^* \in \overline{S}$. Hence, we conclude that there exists $x^* \in \overline{S}$ such that $\sup_S f = f(x^*)$. $\square$

**Definition 11** (Adjacent boundaries). Consider the partition of $\mathbb{R}^n$ by $N$ boundaries $N(h_i)$ for $i = 1, \ldots, N$, where $h_i$ is polynomial of $\mathbf{z}$. Let $C$ be any connected components of $R^n - \cup_{i=1}^n N(h_i)$. Then we say that a boundary $N(p)$ is adjacent to $C$ if $\overline{C} \cap N(p) \neq \emptyset$.

F.6 MONOTONIC CURVE AND ITS PROPERTY

We now present the definition of monotonic curve in high dimension, a key component in our analysis.

**Definition 12** ($x$-Monotonic curve). Let

$$f : \mathbb{R} \times \mathbb{R}^d \quad \to \quad \mathbb{R}^d$$
$$(x, \boldsymbol{y}) \quad \mapsto \quad (f_1(x, \boldsymbol{y}), \ldots, f_d(x, \boldsymbol{y}))$$

be a vector valued function, where each function $f_i$ is a polynomial of $x$ and $\boldsymbol{y}$ for $i = 1, \ldots, d$. Assume that $\mathbf{0} \in \mathbb{R}^d$ is a regular value of $f$, meaning that the set $V_f = \{(x, \boldsymbol{y}) \mid f_i(x, \boldsymbol{y}), i =$

$1, \ldots, d\}$ defines a smooth 1-manifold in $\mathbb{R} \times \mathbb{R}^d$. Let $V' \subset V_f$ be a connected components of $V_f$, and let $C \subset V'$ be an connected open set in $V'$ which is diffeomorphic to $(0, 1)$. The curve $C$ is said to be $x$-*monotonic* if for any point $(a, \boldsymbol{b}) \in C$, we have $\det(J_{f,\boldsymbol{y}}(a, \boldsymbol{b})) \neq 0$, where $J_{f,\boldsymbol{y}}(a, \boldsymbol{b})$ is a Jacobian of $f$ with respect to $\boldsymbol{y}$ evaluated at $(a, \boldsymbol{b})$, defined as

$$J_{f,\boldsymbol{y}}(a, \boldsymbol{b}) = \left[ \frac{\partial f_i}{\partial y_j}(a, \boldsymbol{b}) \right]_{d \times d}.$$

Informally, a key property of an $x$-monotonic curve $C$ is that for any $x_0$, there exists exactly one $\boldsymbol{y}$ such that $(x_0, \boldsymbol{y}) \in C$. We will formalize this claim in Lemma F.18, but first, we will review some fundamental results necessary for the proof.

**Theorem F.16** (Implicit function theorem, Buck (2003)). *Consider the multivariate vector-valued function $f$*

$$f : \mathbb{R}^{n+m} \to \mathbb{R}^m$$
$$(\boldsymbol{x}, \boldsymbol{y}) \mapsto (f_1(\boldsymbol{x}, \boldsymbol{y}), \ldots, f_m(\boldsymbol{x}, \boldsymbol{y})),$$

*and assume that $f$ is continuously differentiable. Let $f(\boldsymbol{a}, \boldsymbol{b}) = \boldsymbol{0}$ for some $(\boldsymbol{a}, \boldsymbol{b}) \in \mathbb{R}^{n+m}$, and the Jacobian*

$$J_{f,\boldsymbol{y}} = \left[ \frac{\partial f_i}{\partial y_j}(\boldsymbol{a}, \boldsymbol{b}) \right]_{m \times m}$$

*is invertible, then there exists a neighborhood $U \subset \mathbb{R}^n$ containing $\boldsymbol{a}$, there exists a neighborhood $V \subset \mathbb{R}^m$ containing $\boldsymbol{b}$, such that there exists an unique function $g : U \to V$ such that $g(\boldsymbol{a}) = \boldsymbol{b}$ and $f(\boldsymbol{x}, g(\boldsymbol{x})) = 0$ for all $x \in U$. We can also say that for $(\boldsymbol{x}, \boldsymbol{y}) \in U \times V$, we have $\boldsymbol{y} = g(\boldsymbol{x})$. Moreover, $g$ is continuously differentiable and, if we denote*

$$J_{f,\boldsymbol{x}}(\boldsymbol{a}, \boldsymbol{b}) = \left[ \frac{\partial f_i}{\partial x_j}(\boldsymbol{a}, \boldsymbol{b}) \right]_{m \times n}$$

*then*

$$\left[ \frac{\partial g_i}{\partial x_j}(\boldsymbol{x}) \right]_{m \times n} = - \left[ J_{f,\boldsymbol{y}}(\boldsymbol{x}, g(\boldsymbol{x})) \right]_{m \times m}^{-1} \cdot \left[ J_{f,\boldsymbol{x}}(\boldsymbol{x}, g(\boldsymbol{x})) \right]_{m \times n}.$$

**Theorem F.17** (Vector-valued mean value theorem). *Let $S \subseteq \mathbb{R}^n$ be open and let $f : S \to \mathbb{R}^m$ be differentiable on all of $S$. Let $\boldsymbol{x}, \boldsymbol{y} \in S$ be such that the line segment connecting these two points contained in $S$, i.e. $L(\boldsymbol{x}, \boldsymbol{y}) \subset S$, where $L(\boldsymbol{x}, \boldsymbol{y}) = \{t\boldsymbol{x} + (1 - t)\boldsymbol{y} \mid t \in [0, 1]\}$. Then for every $\boldsymbol{a} \in \mathbb{R}^m$, there exists a point $\boldsymbol{z} \in L(\boldsymbol{x}, \boldsymbol{y})$ such that $\langle \boldsymbol{a}, f(\boldsymbol{y}) - f(\boldsymbol{x}) \rangle = \langle \boldsymbol{a}, J_{f,\boldsymbol{x}}(z)^\top (\boldsymbol{y} - \boldsymbol{x}) \rangle$.*

We now present a formal statement and proof for the key property of $x$-monotonic curves.

**Lemma F.18.** *Let $C$ be an curve defined as in Definition 12. Then for any $x_0$, the hyperplane $x = x_0$ intersects with $C$ at at most 1 points.*

*Proof.* (of Proposition F.18) Since $C$ is diffeomorphic to $(0, 1)$, there exists a continuously differentiable function $h$, where

$$h : (0, 1) \to C$$
$$t \mapsto (x, \boldsymbol{y}) = (h_0(t), h_1(t), \ldots, h_d(t)) \in C,$$

with correspond inverse function $h^{-1} : C \to (0, 1)$ which is also continuously differentiable.

We will prove the statement by contradiction. Assume that there exists $(x_0, \boldsymbol{y}_1), (x_0, \boldsymbol{y}_2) \in C$ where $\boldsymbol{y_1} \neq \boldsymbol{y_2}$. Then we have two corresponding values $t_1 = h^{-1}(x_0, \boldsymbol{y}_1) \neq t_2 = h^{-1}(x_0, \boldsymbol{y}_2)$. Using Theorem F.17 for the function $h$, for any $\boldsymbol{a} \in \mathbb{R}^d$, there exists $z_a \in (0, 1)$ such that

$$\langle \boldsymbol{a}, (0, \Delta\boldsymbol{y}) \rangle = \langle \boldsymbol{a}, \Delta t J_{h,t}(z_a) \rangle,$$

where $\Delta\boldsymbol{y} = \boldsymbol{y}_2 - \boldsymbol{y}_1 \neq \boldsymbol{0}$, $\Delta t = t_2 - t_1 \neq 0$, and $J_{h,t}(z_a) = (\frac{\partial h_0}{\partial t}(z_{\boldsymbol{a}}), \frac{\partial h_1}{\partial t}(z_{\boldsymbol{a}}), \ldots, \frac{\partial h_d}{\partial t}(z_{\boldsymbol{a}}))$.

Choose $\boldsymbol{a} = \boldsymbol{a}_1 = (1, 0, \ldots, 0)$, then from above, there exists $z_{\boldsymbol{a}_1} \in (0, 1)$ such that $\frac{\partial h_0}{\partial t}\Big|_{t=z_{\boldsymbol{a}_1}} = 0$. Now, consider the point $(x_{\boldsymbol{a}_1}, \boldsymbol{y}_{\boldsymbol{a}_1}) = h(z_{\boldsymbol{a}_1})$. From the assumption, $\det(J_{f,\boldsymbol{y}}(x_{\boldsymbol{a}_1}, \boldsymbol{y}_{\boldsymbol{a}_1})) \neq$

0. Therefore, from Theorem F.16, there exists neighborhoods $U \subset \mathbb{R}$ containing $x_{\boldsymbol{a}_1}$, $V \subseteq \mathbb{R}^d$ containing $\boldsymbol{y}_{\boldsymbol{a}_1}$, such that there exists a continuously differentiable function $g : U \to \mathbb{R}^d$, such that for any $(x, \boldsymbol{y}) \in U \times V$, we have $\boldsymbol{y} = g(x)$. Again, at the point $(x_{\boldsymbol{a}_1}, \boldsymbol{y}_{\boldsymbol{a}_1})$ corresponding to $t = z_{\boldsymbol{a}_1}$, we have

$$\frac{\partial y_i}{\partial t}\bigg|_{t=z_{\boldsymbol{a}_1}} = \frac{\partial g_i}{\partial x} \cdot \frac{\partial x}{\partial t}\bigg|_{t=z_{\boldsymbol{a}_1}} = 0.$$

This means that at the point $t = z_{\boldsymbol{a}_1}$, we have $\frac{\partial x}{\partial t}\big|_{t=z_{\boldsymbol{a}_1}} = \frac{\partial y_i}{\partial t}\big|_{\boldsymbol{a}_1} = 0$.

Note that since $h$ is a diffeomorphism, we have $t = (h^{-1} \circ h)(t)$. From chain rule, we have $1 = J_{h^{-1},h} \cdot J_{h,t}$. However, if we let $t = z_{\boldsymbol{a}_1}$, then $J_{h,t}(\boldsymbol{a}_1) = 0$, meaning that $J_{h^{-1},h} \cdot J_{h,t}(z_{\boldsymbol{a}_1}) = 0$, leading to a contradiction. $\square$

From Definition 13 and Proposition F.18, for each $x$-monotonic curve $C$, we can define their $x$-end points, which are the maximum and minimum of $x$-coordinate that a point in $C$ can have.

**Definition 13** (x-End points of monotonic curve in high dimension). Let $V$ is an monotonic curves as defined in Definition 12. Then we call $\sup\{x \mid \exists \boldsymbol{y}, (x, \boldsymbol{y}) \in V\}$ and $\inf\{x \mid \exists \boldsymbol{y}, (x, \boldsymbol{y}) \in V\}$ the $x$-end points of $V$.

### F.7 MAIN PROOF FOR THEOREM 5.1

**Notation.** We denote $[n] = \{1, \ldots, n\}$. For a polynomial $p(\boldsymbol{x})$, denote $Z(p) = \{\boldsymbol{x} : p(\boldsymbol{x}) = 0\}$ the zero set of $p$. For a set $C \subset \mathbb{R}^d$, denote $\overline{C}$ the closure of $C$, $\text{int}(C)$ the interior of $C$, $\text{bd}(C) = \overline{C} - \text{int}(C)$ the boundary of $C$.

#### F.7.1 A PROOF THAT REQUIRES STRONGER ASSUMPTION

We first give a proof for the case where the piece functions $f_{\boldsymbol{x},i}$ and boundaries $h_{\boldsymbol{x},i}$ satisfies a bit stronger assumption.

**Assumption 2** (Regularity assumption). Assume that for any function $u_{\boldsymbol{x}}^*(\alpha)$, we have the following regularity condition: for any piece function $f_{\boldsymbol{x},i}$ and $S \leq d+1$ boundary functions $h_1, \ldots, h_S$ chosen from $\{h_{\boldsymbol{x},1}, \ldots, h_{\boldsymbol{x},M}\}$, we have

1. For any $(\alpha, \boldsymbol{w}) \in \overline{h}^{-1}(\boldsymbol{0})$, we have $\text{rank}(J_{\overline{h}, \boldsymbol{w}}(\alpha, \boldsymbol{w})) = S$, where $\overline{h} = (h_1(\alpha, \boldsymbol{w}), \ldots, h_S(\alpha, \boldsymbol{w}))$.

2. For any $(\alpha, \boldsymbol{w}, \boldsymbol{\lambda}) \in k^{-1}(\boldsymbol{0})$, we have $\text{rank}(J_{k,(\boldsymbol{w},\boldsymbol{\lambda})}(\alpha, \boldsymbol{w}, \boldsymbol{\lambda})) = d + S$. Here

$$k(\lambda, \boldsymbol{w}, \boldsymbol{\lambda}) = (k_1(\alpha, \boldsymbol{w}, \boldsymbol{\lambda}), \ldots, k_{d+S}(\alpha, \boldsymbol{w}, \boldsymbol{\lambda})),$$

and

$$\begin{cases} k_i(\alpha, \boldsymbol{w}, \boldsymbol{\lambda}) = h_i(\alpha, \boldsymbol{w}), \quad i = 1, \ldots, S, \\ k_{s+j}(\alpha, \boldsymbol{w}, \boldsymbol{\lambda}) = \frac{\partial f_{\boldsymbol{x},i}}{\partial w_j} + \sum_{i=1}^s \lambda_i \frac{\partial h_i}{\partial w_j}, \quad j = 1, \ldots, d. \end{cases}$$

3. For any $(\alpha, \boldsymbol{w}, \boldsymbol{\lambda}, \boldsymbol{\theta}, \boldsymbol{\gamma}) \in \overline{k}^{-1}(\boldsymbol{0})$, we have $\text{rank}(J_{\overline{k},(\alpha,\boldsymbol{w},\boldsymbol{\lambda},\boldsymbol{\theta},\boldsymbol{\gamma})})(\alpha, \boldsymbol{w}, \boldsymbol{\lambda}, \boldsymbol{\theta}, \boldsymbol{\gamma}) = 2d + 2S + 1$. Here

$$\overline{k}(\alpha, \boldsymbol{w}, \boldsymbol{\lambda}, \boldsymbol{\theta}, \boldsymbol{\gamma}) = (\overline{k}_1(\alpha, \boldsymbol{w}, \boldsymbol{\lambda}, \boldsymbol{\theta}, \boldsymbol{\gamma}), \ldots, \overline{k}_{2d+2S+1}(\alpha, \boldsymbol{w}, \boldsymbol{\lambda}, \boldsymbol{\theta}, \boldsymbol{\gamma})),$$

and

$$\begin{cases} \overline{k}_z = h_{\boldsymbol{x},i,z}^S, z = 1, \ldots, S \\ \overline{k}_{S+z} = \sum_{t=1}^d \gamma_t \frac{\partial h_{\boldsymbol{x},i,z}^S}{\partial w_t}, z = 1, \ldots, S \\ \overline{k}_{2S+z} = \frac{\partial f_{\boldsymbol{x},i}}{\partial w_z} + \sum_{j=1}^S \lambda_j \frac{\partial h_{\boldsymbol{x},i,j}^S}{\partial w_z}, z = 1 \ldots, d \\ \overline{k}_{2S+d+z} = \frac{\partial f_{\boldsymbol{x},i}}{\partial w_z} + \sum_{j=1}^S \theta_j \frac{h_{\boldsymbol{x},i,j}^S}{\partial w_z} + \sum_{t=1}^d \gamma_t \left[ \frac{\partial^2 f_{\boldsymbol{x},i}}{\partial w_t \partial w_z} + \sum_{j=1}^S \lambda_j \frac{\partial^2 h_{\boldsymbol{x},i,j}^S}{\partial w_t \partial w_z} \right], z = 1, \ldots, d \\ \overline{k}_{2S+2d+1} = \frac{\partial f_{\boldsymbol{x},i}}{\partial \alpha} + \sum_{j=1}^S \theta_j \frac{h_{\boldsymbol{x},i,j}^S}{\partial \alpha} + \sum_{t=1}^d \gamma_t \left[ \frac{\partial^2 f_{\boldsymbol{x},i}}{\partial w_t \partial \alpha} + \sum_{j=1}^S \lambda_j \frac{\partial^2 h_{\boldsymbol{x},i,j}^S}{\partial w_t \partial \alpha} \right]. \end{cases}$$

**Remark 3.** We note that Assumption 2.3 implies Assumption 2.2, and Assumption 2.2 implies Assumption 2.1. For convenience, we present Assumption 2 with a different sub-assumption is for readability, and because each sub-assumption has its own geometric meaning in our analysis. In particular:

- Assumption 2.1 implies that the intersections of any $S \leq d+1$ boundaries are regular: they are either empty, or are a smooth $(d + 1 - S)$-manifold in $\mathbb{R}^{d+1}$.

- Assumption 2.2 refers to the regularity of the derivative curves.

- Assumption 2.3 implies that the number of local extrema of the piece function along any derivative curve is finite.

**Theorem F.19.** *Assume that Assumption 2 holds, then for any problem instance $\boldsymbol{x} \in \mathcal{X}$, the dual utility function $u_{\boldsymbol{x}}^*$ satisfies the followings:*

*(a) The hyperparameter domain $\mathcal{A}$ can be partitioned into at most*

$$\mathcal{O}\left( N\Delta^{4d+2} \left( \frac{eM}{d+1} \right)^{d+1} + NM(2\Delta)^{2d+2} \left( \frac{eM}{d+1} \right)^{d+1} \right)$$

*intervals such that $u_{\boldsymbol{x}}^*(\alpha)$ is a continuous function over any interval in the partition, where $N$ and $M$ are the upper-bound for the number of pieces and boundary functions, and $\Delta = \max\{\Delta_p, \Delta_b\}$ is the maximum degree of piece $f_{\boldsymbol{x},i}$ and boundary $h_{\boldsymbol{x},i}$ polynomials.*

*(b) $u_{\boldsymbol{x}}^*(\alpha)$ has $\mathcal{O}\left( N\Delta^{4d+3} \left( \frac{eM}{d+1} \right)^{d+1} \right)$ local maxima for any problem instance $\boldsymbol{x}$ overall all such intervals.*

*Proof.* (a) First, note that we can rewrite $u_{\boldsymbol{x},i}^*(\alpha)$ as

$$u_{\boldsymbol{x},i}^*(\alpha) = \max_{\boldsymbol{w}:(\alpha,\boldsymbol{w})\in \overline{R}_{\boldsymbol{x},i}} f_{\boldsymbol{x},i}(\alpha, \boldsymbol{w}).$$

Since $\overline{R}_{\boldsymbol{x},i}$ is connected, let

$$\alpha_{\boldsymbol{x},i,\inf} = \inf\{\alpha \mid \exists \boldsymbol{w} : (\alpha, \boldsymbol{w}) \in R_{\boldsymbol{x},i}\}, \alpha_{\boldsymbol{x},i,\sup} = \sup\{\alpha \mid \exists \boldsymbol{w} : (\alpha, \boldsymbol{w}) \in R_{\boldsymbol{x},i}\}$$

be the $\alpha$-extreme points of $\overline{R}_{\boldsymbol{x},i}$ (Definition 5). Then, for any $\alpha \in (\alpha_{\boldsymbol{x},i,\inf}, \alpha_{\boldsymbol{x},i,\sup})$, there exists $\boldsymbol{w}$ such that $(\alpha, \boldsymbol{w}) \in \overline{R}_{\boldsymbol{x},i}$.

Let $\mathbf{H}_{\boldsymbol{x},i}$ be the set of adjacent boundaries of $R_{\boldsymbol{x},i}$. By assumption, we have $|\mathbf{H}_{\boldsymbol{x},i}| \leq M$. For any subset $\mathcal{S} = \{h_{\mathcal{S},1}, \ldots, h_{\mathcal{S},S}\} \subset \mathbf{H}_{\mathbf{x},i}$, where $|\mathcal{S}| = S$, consider the set of $(\alpha, \boldsymbol{w})$ defined by

$$h_{\mathcal{S},i}(\alpha, \boldsymbol{w}) = 0, \quad i = 1, \ldots, S. \tag{1}$$

If $S > d + 1$, from Assumption 2, the set of $(\alpha, \boldsymbol{w})$ above is empty. Consider $S \leq d + 1$, from Assumption 2, the above defines a smooth $d + 1 - S$ manifolds in $\mathbb{R}^{d+1}$. Note that, the set of above is exactly the set of $(\alpha, \boldsymbol{w})$ defined by

$$\sum_{i=1}^{S} h_{\mathcal{S},i}(\alpha, \boldsymbol{w})^2 = 0.$$

Therefore, from Lemma F.10, the number of connected components of such manifolds is at most $2(2\Delta)^{d+1}$. Each connected components correspond to 2 $\alpha$-extreme points, meaning that there are at most $4(2\Delta)^{d+1}$ $\alpha$-extreme points for all the connected components of the smooth manifolds defined by Equation 1. Taking all possible subset of boundaries of at most $d + 1$ elements, we have total of at most $\mathcal{N}$ $\alpha$-extreme points, where

$$\mathcal{N} \leq (2\Delta)^{d+1} \sum_{S=0}^{d+1} \binom{M}{S} \leq (2\Delta)^{d+1} \left( \frac{eM}{d+1} \right)^{d+1}.$$

Here, the final inequality is from Lemma F.6.

Now, let $\mathcal{A}_1$ be the set of such $\alpha$-extreme points after reordering. For each interval $I_t = (\alpha_t, \alpha_{t+1})$ of consecutive points $\mathcal{A}_1$, the set $\mathbf{S}_t^1 \subset 2^{\mathbf{H}_{\boldsymbol{x},i}}$ of sets of boundaries is fixed. here, the set $\mathbf{S}_t$ consists of all set of boundary $\mathcal{S} = \{h_{\mathcal{S},1}, \ldots, h_{\mathcal{S},S}\}$ such that for any $\alpha \in (\alpha_t, \alpha_{t+1})$, there exists $\boldsymbol{w}$ such that $h_{\mathcal{S},i}(\alpha, \boldsymbol{w}) = 0$ for any $i = 1, \ldots, S$. Here, note that $(\alpha, \boldsymbol{w})$ is not necessarily in $\overline{R}_{\boldsymbol{x},i}$, i.e. it might be infeasible. Now, for any *fixed* $\alpha \in I_t$, assume that $\boldsymbol{w}_\alpha$ is a maxima of $f_{\boldsymbol{x},i}$ in $\overline{R}_{\boldsymbol{x},i}$ (which exists due to the compactness of $\overline{R}_{\boldsymbol{x},i}$), meaning that $(\alpha, \boldsymbol{w}_\alpha)$ is also a local extrema in $\overline{R}_{\boldsymbol{x},i}$. This implies there exists a set of boundaries $\mathcal{S} \in \mathbf{S}_t$ and $\boldsymbol{\lambda}$ such that $(\alpha, \boldsymbol{w}_\alpha)$ satisfies the following due to Theorem F.7

$$\begin{cases} h_{\mathcal{S},j}(\alpha, \boldsymbol{w}_\alpha) = 0, j = 1, \ldots, S \\ \frac{\partial f(\alpha, \boldsymbol{w}_\alpha)}{\partial w_i} + \sum_{j=1}^{S} \lambda_j \frac{\partial h_{\mathcal{S},j}(\alpha, \boldsymbol{w}_\alpha)}{\partial w_i}, i = 1, \ldots, d, \end{cases}$$

which defines a smooth 1-dimensional manifold $\mathcal{M}^{\mathcal{S}}$ in $\mathbb{R}^{d+S+1}$ by Assumption 2. Again, from Lemma F.10, the number of connected components of $\mathcal{M}_{\mathcal{S}}$ is at most $2(2\Delta)^{d+S+1}$, corresponding to at most $4(2\Delta)^{d+S+1}$ $\alpha$-extreme points. Taking all possible subsets $\mathcal{S}$ of at most $d+1$ elements of $\mathbf{H}_{\boldsymbol{x},i}$, we have at most $\mathcal{O}\left((2\Delta)^{2d+2}\left(\frac{eM}{d+1}\right)^{d+1}\right)$ such $\alpha$-extreme points.

Let $\mathcal{A}_2$ be the set contains all the points $\alpha$ in $\mathcal{A}_1$ and the $\alpha$-extreme points above and reordering them. Then in any interval $I_t = (\alpha_t, \alpha_{t+1})$ of consecutive points $\mathcal{A}_2$, the set $\mathbf{S}_t^2$ is fixed. Here, the set $\mathbf{S}_t$ consists of all sets of boundary $\mathcal{S}$ such that for any $\alpha \in (\alpha_t, \alpha_{t+1})$, there exists $\boldsymbol{w}_\alpha$ and $\boldsymbol{\lambda}$ such that $(\alpha, \boldsymbol{w}_\alpha, \boldsymbol{\lambda})$ satisfies

$$\begin{cases} h_{\mathcal{S},j}(\alpha, \boldsymbol{w}_\alpha) = 0, j = 1, \ldots, S \\ \frac{\partial f(\alpha, \boldsymbol{w}_\alpha)}{\partial w_i} + \sum_{j=1}^{S} \lambda_j \frac{\partial h_{\mathcal{S},j}(\alpha, \boldsymbol{w}_\alpha)}{\partial w_i}, i = 1, \ldots, d. \end{cases}$$

Note that the points $(\alpha, \boldsymbol{w}_\alpha, \boldsymbol{\lambda})$ might not be in the feasible region $\overline{R}_{\boldsymbol{x},i}$. For each $\mathcal{S}$, the points $(\alpha, \boldsymbol{w}, \boldsymbol{\lambda})$ in which $\mathcal{M}^{\mathcal{S}}$ can enter or exit the feasible region $\overline{R}_{\boldsymbol{x},i}$ satisfies equation

$$\begin{cases} h_{\mathcal{S},i}(\alpha, \boldsymbol{w}) = 0, i = 1, \ldots, S, \\ h'(\alpha, \boldsymbol{w}) = 0, \text{for some } h' \in \mathbf{H}_{\boldsymbol{x},i} - \mathcal{S} \\ \frac{\partial f(\alpha, \boldsymbol{w})}{\partial w_i} + \sum_{j=1}^{S} \lambda_j \frac{\partial h_{\mathcal{S},j}(\alpha, \boldsymbol{w}_\alpha)}{\partial w_i}, i = 1, \ldots, d. \end{cases}$$

of which the number of solution is finite due to Assumption 2. The number of such points is $2(2\Delta)^{d+S+1}$ for each $\mathcal{S} \subset \mathbf{H}_{\boldsymbol{x},i}$, $|\mathcal{S}| \leq d+1$ and each $h' \in \mathbf{H}_{\boldsymbol{x},i} - \mathcal{S}$, meaning that there are at most $2M(2\Delta)^{d+S+1}$ such points for each $\mathcal{S}$. Taking all possible sets $\mathcal{S}$, we have at most $\mathcal{O}\left(M(2\Delta)^{2d+2}\left(\frac{eM}{d+1}\right)^{d+1}\right)$.

Let $\mathcal{A}_3$ be the set contains all the points in $\mathcal{A}_2$ and the $\alpha$ points above and reordering them. Then for any interval $I_t = (\alpha_t, \alpha_{t+1})$, the set $\mathbf{S}_t^3$ is fixed. Here, the set $\mathbf{S}_t^3$ consists of all sets of boundary $\mathcal{S}$ such that for any $\alpha \in (\alpha_t, \alpha_{t+1})$ *fixed*, there exists $\boldsymbol{w}_\alpha$ and $\boldsymbol{\lambda}$ such that $(\alpha, \boldsymbol{w}_\alpha, \boldsymbol{\lambda})$ satisfies

$$\begin{cases} h(\alpha, \boldsymbol{w}_\alpha) = 0, h \in S \\ \frac{\partial f(\alpha, \boldsymbol{w}_\alpha)}{\partial w_i} + \sum_{h \in \mathcal{S}} \lambda_h \frac{\partial h(\alpha, \boldsymbol{w}_\alpha)}{\partial w_i}, i = 1, \ldots, d, \\ (\alpha, \boldsymbol{w}) \in \overline{R}_{\boldsymbol{x},i}. \end{cases}$$

Finally, we further break the smooth 1-manifold $\mathcal{M}^{\mathcal{S}}$ defined as above into monotonic curves (Definition 12), which we show to have attract property (Proposition F.18): for each monotonic curve $C$ and an $\alpha_0$, there is at most 1 point in $C$ such that the coordinate $\alpha = \alpha_0$. For the smooth 1-manifold $\mathcal{M}^{\mathcal{S}}$, from Definition 12, the points that break $\mathcal{M}^{\mathcal{S}}$ into monotonic curves satisfies

$$\begin{cases} k_i(\alpha, \boldsymbol{w}, \boldsymbol{\lambda}) = h_{\mathcal{S},i}(\alpha, \boldsymbol{w}_\alpha) = 0, i = 1, \ldots, S \\ k_{S+j}(\alpha, \boldsymbol{w}, \boldsymbol{\lambda}) = \frac{\partial f(\alpha, \boldsymbol{w}_\alpha)}{\partial w_i} + \sum_{i=1}^{S} \lambda_i \frac{\partial h_{\mathcal{S},i}(\alpha, \boldsymbol{w}_\alpha)}{\partial w_i}, i = 1, \ldots, d, \\ \det(J_{k,(\boldsymbol{w},\boldsymbol{\lambda})}) = 0. \end{cases}$$

Here, $k = (k_1, \ldots, k_{S+d}) : \mathbb{R}^{d+S+1} \to \mathbb{R}^{d+S+1}$, and $J_{k,(\boldsymbol{w},\boldsymbol{\lambda})}$ is the Jacobian of function $k$ with respect to $\boldsymbol{w}, \boldsymbol{\lambda}$. Note that $J_{k,(\boldsymbol{w},\boldsymbol{\lambda})}$ is a polynomial in $\alpha, \boldsymbol{w}, \boldsymbol{\lambda}$ of degree at most $\Delta^{d+S}$. From Assumption 2 and Bezout's theorem, for each possible choice of $\mathcal{S}$, there are at most $\Delta^{2d+2S}$ such points

$(\alpha, \boldsymbol{w}, \boldsymbol{\lambda})$ satisfies the above. Taking all possible sets $\mathcal{S}$, we have at most $\mathcal{O}\left(\Delta^{4d+2}\left(\frac{eM}{d+1}\right)^{d+1}\right)$ such points.

In summary, there are a set of $\alpha$ points $\mathcal{A}_4$ of at most $\mathcal{O}\left(\Delta^{4d+2}\left(\frac{eM}{d+1}\right)^{d+1} + M(2\Delta)^{2d+2}\left(\frac{eM}{d+1}\right)^{d+1}\right)$ points such such that for any interval $I_t = (\alpha_t, \alpha_{t+1})$ of consecutive points $(\alpha_t, \alpha_{t+1})$ in $\mathcal{A}_4$, there exists a set $\mathcal{C}_t$ of monotonic curves such that for any $\alpha \in (\alpha_t, \alpha_{t+1})$, we have

$$u_{\boldsymbol{x},i}^*(\alpha) = \max_{C \in \mathcal{C}} \{f_{\boldsymbol{x},i}(\alpha, \boldsymbol{w}) \mid \exists \boldsymbol{\lambda}, (\alpha, \boldsymbol{w}, \boldsymbol{\lambda}) \in C\}.$$

In other words, the value of $u_{\boldsymbol{x},i}^*(\alpha)$ for $\alpha \in I_t$ is the point-wise maximum of value of functions $f_{\boldsymbol{x},i}$ along the set of monotonic curves $\mathcal{C}$. From Theorem F.14, we have $u_{\boldsymbol{x},i}^*(\alpha)$ is continuous over $I_t$. Therefore, we conclude that the number of discontinuities of $u_{\boldsymbol{x},i}^*(\alpha)$ is at most

$$\mathcal{O}\left(\Delta^{4d+2}\left(\frac{eM}{d+1}\right)^{d+1} + M(2\Delta)^{2d+2}\left(\frac{eM}{d+1}\right)^{d+1}\right).$$

Finally, recall that

$$u_{\boldsymbol{x}}^*(\alpha) = \max_{i \in [N]} u_{\boldsymbol{x},i}(\alpha),$$

and combining with Theorem F.3, we conclude that the number of discontinuity points of $u_{\boldsymbol{x}}^*(\alpha)$ is at most $\mathcal{O}\left(N\Delta^{4d+2}\left(\frac{eM}{d+1}\right)^{d+1} + NM(2\Delta)^{2d+2}\left(\frac{eM}{d+1}\right)^{d+1}\right).$ $\square$

Combining Theorem F.19 and 3.2, we have the following result.

**Theorem F.20.** *Let* $\mathcal{U} = \{u_\alpha : \mathcal{X} \to [0,1] \mid \alpha \in \mathcal{A}\}$, *where* $\mathcal{A} = [\alpha_{\min}, \alpha_{\max}] \subset \mathbb{R}$. *Assume that any dual utility function* $u_{\boldsymbol{x}}^*$ *admits piecewise polynomial structures that satisfies Assumption 2. Then we have* $\mathrm{Pdim}(\mathcal{U}) = \mathcal{O}(\log N + d\log(\Delta M))$. *Here, $M$ and $N$ are the number of boundaries and functions, and $\Delta$ is the maximum degree of boundaries and piece functions.*

### F.7.2 Relaxing Assumption 2 to Assumption 1

In this section, we show how we can give a relaxation from Assumption 2 to our main Assumption 1. In particular, we show that for any dual utility function $u_{\boldsymbol{x}}^*$ that satisfies Assumption 1, we can construct a function $v_{\boldsymbol{x}}^*$ such that: (1) The piecewise structure of $v_{\boldsymbol{x}}^*$ satisfies Assumption 2, and (2) $\|u_{\boldsymbol{x}}^* - v_{\boldsymbol{x}}^*\|$ can be *arbitrarily* small. This means that, for a utility function class $\mathcal{U}$, we can construct a new function class $\mathcal{V}$ of which each dual function $v_{\boldsymbol{x}}^*$ satisfies Assumption 2. We then can establish pseudo-dimension upper-bound for $\mathcal{V}$ using Theorem F.19, and then recover learning guarantee for $\mathcal{U}$ using Lemma C.4.

First, we recall a useful result regarding sets of regular polynomials. This result states that given a set of regular polynomials and a new polynomial, we can modify the new polynomial by an arbitrarily small amount such that adding it to the set preserves the regularity of the entire set.

**Lemma F.21** (Warren (1968)). *Let* $p(\boldsymbol{x}), q_1(\boldsymbol{x}), \dots, q_m(\boldsymbol{x})$ *be polynomials. Assume that* $\boldsymbol{0}$ *is a regular value of* $\boldsymbol{q} = (q_1, \dots, q_m)$, *then for all but finitely many number of real numbers* $\alpha$, *we have* $\boldsymbol{0}$ *is also a regular value for* $\overline{\boldsymbol{q}} = (q_1, \dots, q_m, p - \alpha)$.

We now present the main claim in this section, which says that for any function $u_{\boldsymbol{x}}^*(\alpha)$ that satisfies Assumption 1, we can construct a function $v_{\boldsymbol{x}}^*(\alpha)$ that satisfies Assumption 2 and that $\|u_{\boldsymbol{x}}^* - v_{\boldsymbol{x}}^*\|_\infty$ can be arbitrarily small.

**Lemma F.22.** *Let* $u_{\boldsymbol{x}}^*$ *be a dual utility function of a utility function class* $\mathcal{U}$. *Assume that the piece-wise polynomial structures of* $u_{\boldsymbol{x}}^*$ *satisfies Assumption 1, then we can construct the function* $v_{\boldsymbol{x}}^*$ *such that* $v_{\boldsymbol{x}}^*$ *has piece-wise polynomial structures that satisfies Assumption 2, and* $\|u_{\boldsymbol{x}}^* - v_{\boldsymbol{x}}^*\|_\infty$ *can be arbitrarily small.*

*Proof.* Consider the functions $\overline{k}$

$$\overline{k}(\alpha, \boldsymbol{w}, \boldsymbol{\lambda}, \boldsymbol{\theta}, \boldsymbol{\gamma}) = (\overline{k}_1(\alpha, \boldsymbol{w}, \boldsymbol{\lambda}, \boldsymbol{\theta}, \boldsymbol{\gamma}), \dots, \overline{k}_{2d+2s+1}(\alpha, \boldsymbol{w}, \boldsymbol{\lambda}, \boldsymbol{\theta}, \boldsymbol{\gamma})),$$

and

$$\begin{cases} \overline{k}_z(\alpha, \boldsymbol{w}, \boldsymbol{\lambda}, \boldsymbol{\theta}, \boldsymbol{\gamma}) = h_{x,i,z}^{\mathcal{S}}(\alpha, \boldsymbol{w}), z = 1, \ldots, S \\ \overline{k}_{S+z}(\alpha, \boldsymbol{w}, \boldsymbol{\lambda}, \boldsymbol{\theta}, \boldsymbol{\gamma}) = \sum_{t=1}^d \gamma_t \frac{\partial h_{x,i,z}^{\mathcal{S}}(\alpha, \boldsymbol{w})}{\partial w_t}, z = 1, \ldots, S \\ \overline{k}_{2S+z}(\alpha, \boldsymbol{w}, \boldsymbol{\lambda}, \boldsymbol{\theta}, \boldsymbol{\gamma}) = \frac{\partial f_{x,i}(\alpha, \boldsymbol{w})}{\partial w_z} + \sum_{j=1}^S \lambda_j \frac{\partial h_{x,i,j}^{\mathcal{S}}(\alpha, \boldsymbol{w})}{\partial w_z}, z = 1 \ldots, d \\ \overline{k}_{2S+d+z}(\alpha, \boldsymbol{w}, \boldsymbol{\lambda}, \boldsymbol{\theta}, \boldsymbol{\gamma}) = \frac{\partial f_{x,i}(\alpha, \boldsymbol{w})}{\partial w_z} + \sum_{j=1}^S \theta_j \frac{h_{x,i,j}^{\mathcal{S}}(\alpha, \boldsymbol{w})}{\partial w_z} \\ \qquad\qquad + \sum_{t=1}^d \gamma_t \left[ \frac{\partial^2 f_{x,i}(\alpha, \boldsymbol{w})}{\partial w_t \partial w_z} + \sum_{j=1}^S \lambda_j \frac{\partial^2 h_{x,i,j}^{\mathcal{S}}(\alpha, \boldsymbol{w})}{\partial w_t \partial w_z} \right] = 0, z = 1, \ldots, d \\ \overline{k}_{2S+2d+1}(\alpha, \boldsymbol{w}, \boldsymbol{\lambda}, \boldsymbol{\theta}, \boldsymbol{\gamma}) = \frac{\partial f_{x,i}(\alpha, \boldsymbol{w})}{\partial \alpha} + \sum_{j=1}^S \theta_j \frac{h_{x,i,j}^{\mathcal{S}}(\alpha, \boldsymbol{w})}{\partial \alpha} + \sum_{t=1}^d \gamma_t \left[ \frac{\partial^2 f_{x,i}(\alpha, \boldsymbol{w})}{\partial w_t \partial \alpha} + \sum_{j=1}^S \lambda_j \frac{\partial^2 h_{x,i,j}^{\mathcal{S}}(\alpha, \boldsymbol{w})}{\partial w_t \partial \alpha} \right]. \end{cases}$$

Since $u_{\boldsymbol{x}}^*$ satisfies Assumption 2.2, then $\mathbf{0}$ is a regular value of $(\overline{k}_1, \ldots, \overline{k}_{2S})$. From Lemma F.21, there exists finitely number of real-valued $\tau$ such that $\mathbf{0}$ is *not* a regular value of $(\overline{k}_1, \ldots, \overline{k}_{2S}, \overline{k}_{2S+1} - \tau)$. Let $\tau^* \neq 0$ be the such $\tau$ such that $|\tau^*|$ is the smallest. Then for any $0 < \tau < |\tau^*|$, we have $\mathbf{0}$ is a regular value of $(\overline{k}_1, \ldots, \overline{k}_{2S}, \overline{k}_{2S+1} - \tau)$. Keep doing so for the all (finite number) polynomials $\hat{k}_{2S+1}, \ldots, \hat{k}_{2S+2d+1}$, we claim that there exists a $\tau^* \neq 0$, such that for any $0 < \tau < |\tau^*|$, we have $\mathbf{0}$ is a regular value of $(\hat{k}_1, \ldots, \hat{k}_{2S}, \hat{k}_{2S+1} - \tau, \ldots, \hat{k}_{2S+2d+1} - \tau)$. We then construct the function $v_{\boldsymbol{x}}^*$ as follow.

- The set of boundary functions is the same as $u_{\boldsymbol{x}}^*$ : $\{h_{\boldsymbol{x},1}, \ldots, h_{\boldsymbol{x},M}\}$.

- In each region $R_{\boldsymbol{x},i}$, the piece function $f'_{\boldsymbol{x},i}(\alpha, \boldsymbol{w})$ of $v_{\boldsymbol{x}}^*$ is defined as:

$$f'_{\boldsymbol{x},i}(\alpha, \boldsymbol{w}) = f_{\boldsymbol{x},i}(\alpha, \boldsymbol{w}) + \tau\alpha + \tau \sum_{z=1}^d w_z,$$

for some $0 < \tau < |\tau^*|$. Then

- $v_{\boldsymbol{x}}^*$ satisfies Assumption 2.

- In any region $R_{\boldsymbol{x},i}$, we have

$$\left| f_{\boldsymbol{x},i}(\alpha, \boldsymbol{w}) - f'_{\boldsymbol{x},i}(\alpha, \boldsymbol{w}) \right| = \left| \tau\alpha + \tau \sum_{z=1}^d w_z \right| \leq \tau C,$$

where $C = (d+1)\max\{|\alpha_{\min}, \alpha_{\max}, w_{\min}, w_{\max}|\}$. This implies

$$\sup_{\boldsymbol{w}:(\alpha,\boldsymbol{w})\in R_{\boldsymbol{x},i}} f_{\boldsymbol{x},i}(\alpha, \boldsymbol{w}) - 2\tau C \leq \sup_{\boldsymbol{w}:(\alpha,\boldsymbol{w})\in R_{\boldsymbol{x},i}} f'_{\boldsymbol{x},i}(\alpha, \boldsymbol{w}) \leq \sup_{\boldsymbol{w}:(\alpha,\boldsymbol{w})\in R_{\boldsymbol{x},i}} f_{\boldsymbol{x},i}(\alpha, \boldsymbol{w}) + 2\tau C,$$

or

$$u_{\boldsymbol{x},i}^*(\alpha) - 2\tau C \leq v_{\boldsymbol{x},i}^*(\alpha) \leq u_{\boldsymbol{x},i}^*(\alpha) + 2\tau C \Rightarrow \|u_{\boldsymbol{x},i}^* - v_{\boldsymbol{x},i}^*(\alpha)\|_\infty \leq 2\tau C.$$

Then we conclude that $\|u_{\boldsymbol{x}}^* - v_{\boldsymbol{x}}^*(\alpha)\|_\infty \leq 2\tau C$, and since $\tau$ can be arbitrarily small, we have the conclusion. $\qquad\square$

### F.7.3 RECOVER THE GUARANTEE UNDER ASSUMPTION 1

We now give the formal proof for the Theorem 5.1.

*Theorem* 5.1 (restated). Consider the utility function class $\mathcal{U} = \{u_\alpha : \mathcal{X} \to [0, H] \mid \alpha \in \mathcal{A}\}$. Assume that the dual utility function $u_{\boldsymbol{x}}^*(\alpha) = \sup_{\boldsymbol{w}\in\mathcal{W}} f_{\boldsymbol{x}}(\alpha, \boldsymbol{w})$, and $f_{\boldsymbol{x}}(\alpha, \boldsymbol{w})$ admits piecewise constant polynomial structure with the piece functions $f_{\boldsymbol{x},i}$ and boundaries $h_{\boldsymbol{x},i}$ satisfies Assumption 1. Then for any distribution $\mathcal{D}$ over $\mathcal{X}$, for any $\delta \in (0, 1)$, with probability at least $1 - \delta$ over the draw of $S \sim \mathcal{D}^m$, we have

$$|\mathbb{E}_{\boldsymbol{x}\sim\mathcal{D}}[u_{\hat{\alpha}}(\boldsymbol{x})] - \mathbb{E}_{\boldsymbol{x}\sim\mathcal{D}}[u_{\alpha^*}(\boldsymbol{x})]| \leq \mathcal{O}\left( \sqrt{\frac{\log N + d\log(\Delta M) + \log(1/\delta)}{m}} \right).$$

Here, $M$ and $N$ are the number of boundaries and connected sets, $\Delta = \max\{\delta_p, \delta_d\}$ is the maximum degree of piece $f_{\boldsymbol{x},i}$ and boundaries $h_{\boldsymbol{x},i}$.

*Proof.* Let $\mathcal{U} = \{u_\alpha : \mathcal{X} \to [0, H] \mid \alpha \in \mathcal{A}\}$ be a function class of which each dual utility $u_{\boldsymbol{x}}^*$ satisfies Assumption 1. From Lemma F.7.2, there exists a function class $\mathcal{V} = \{v_\alpha : \mathcal{X} \to [0, H] \mid \alpha \in \mathcal{A}\}$ such that for any problem instance $\boldsymbol{x}$, we have $\|u_{\boldsymbol{x}}^* - v_{\boldsymbol{x}}^*\|_\infty$ can be arbitrarily small, and any $v_{\boldsymbol{x}}^*$ satisfies Assumption 2. From Theorem F.19, we have $\text{Pdim}(\mathcal{V}) = \mathcal{O}(\log N + d \log(\Delta M))$. From Lemma C.4, we have $\mathscr{R}_m(\mathcal{V}) = \mathcal{O}\left(\frac{\text{Pdim}(\mathcal{V})}{m}\right)$. From Lemma C.3, we have $\hat{\mathscr{R}}_S(\mathcal{U}) = \mathcal{O}\left(\sqrt{\frac{\log N + d \log(\Delta M)}{m}}\right)$, where $S \in \mathcal{X}^m$. Finally, standard learning theory result give us the final claim. $\qquad\square$

# G ADDITIONAL DETAILS FOR SECTION 6

## G.1 TUNING THE INTERPOLATION PARAMETER FOR ACTIVATION FUNCTIONS

### G.1.1 REGRESSION CASE

We now provide a formal proof for Theorem 6.1, which analyzes the generalization guarantee for selecting the interpolation hyperparameter of activation functions in neural architecture search.

**Theorem 6.1 (restated).** *Let $\mathcal{L}^{AF}$ denote loss function class defined above, with activation functions $o_1, o_2$ having maximum degree $\Delta$ and maximum breakpoints $p$. Given a problem instance $(X, Y)$, the dual loss function is defined as $\ell_{(X,Y)}^*(\alpha) := \min_{w \in \mathcal{W}} f((X, Y), \boldsymbol{w}; \alpha) = \min_{w \in \mathcal{W}} f_{(X,Y)}(\alpha, \boldsymbol{w})$, and $f_{(X,Y)}(\alpha, \boldsymbol{w})$ admits piecewise polynomial structure with bounded pieces and boundaries. Assume that the piecewise structure of $f_{(X,Y)}(\alpha, \boldsymbol{w})$ satisfies Assumption 1, then for any $\delta \in (0, 1)$, w.p. at least $1 - \delta$ over the draw of problem instances $S \sim \mathcal{D}^m$, where $\mathcal{D}$ is some distribution over $\mathcal{X}$, we have*

$$\left|\mathbb{E}_{(X,Y)\sim\mathcal{D}}[\ell_{\hat{\alpha}}((X, Y))] - \mathbb{E}_{(X,Y)\sim\mathcal{D}}[\ell_{\alpha^*}((X, Y))]\right| = \mathcal{O}\left(\sqrt{\frac{L^2 W \log \Delta + L W \log(Tpk) + \log(1/\delta)}{m}}\right).$$

*Proof.* Let $x_1, \ldots, x_T$ denote the fixed (unlabeled) validation examples from the *fixed* validation dataset $(X, Y)$. We will show a bound $N$ on a partition of the combined parameter-hyperparameter space $\mathcal{W} \times \mathbb{R}$, such that within each piece the function $f_{(X,Y)}(\alpha, \boldsymbol{w})$ is given by a fixed bounded-degree polynomial function in $\alpha, \boldsymbol{w}$ on the given fixed dataset $(X, Y)$, where the boundaries of the partition are induced by at most $M$ distinct polynomial threshold functions. This structure allows us to use our result Theorem 5.1 to establish learning guarantee for the function class $\mathcal{L}^{AF}$.

The proof proceeds by an induction on the number of network layers $L$. For a single layer $L = 1$, the neural network prediction at node $j \in [k_1]$ is given by

$$\hat{y}_{ij} = \alpha o_1(\boldsymbol{w}_j x_i) + (1 - \alpha)o_2(\boldsymbol{w}_j x_i),$$

for $i \in [T]$. $\mathcal{W} \times \mathbb{R}$ can be partitioned by $2Tk_1p$ affine boundary functions of the form $\boldsymbol{w}_j x_i - t_k$, where $t_k$ is a breakpoint of $o_1$ or $o_2$, such that $\hat{y}_{ij}$ is a fixed polynomial of degree at most $l + 1$ in $\alpha, \boldsymbol{w}$ in any piece of the partition $\mathcal{P}_1$ induced by the boundary functions. By Warren's theorem (Lemma F.10), we have $|\mathcal{P}_1| \leq 2\left(\frac{4eTk_1p}{W_1}\right)^{W_1}$.

Now suppose the neural network function computed at any node in layer $L \leq r$ for some $r \geq 1$ is given by a piecewise polynomial function of $\alpha, \boldsymbol{w}$ with at most $|\mathcal{P}_r| \leq \prod_{q=1}^r 2\left(\frac{4eTk_qp(\Delta+1)^q}{W_q}\right)^{W_q}$ pieces, and at most $2Tp\sum_{q=1}^r k_q$ polynomial boundary functions with degree at most $(\Delta + 1)^r$. Let $j' \in [k_{r+1}]$ be a node in layer $r + 1$. The node prediction is given by $\hat{y}_{ij'} = \alpha o_1(\boldsymbol{w}_{j'}\hat{y}_i) + (1 - \alpha)o_2(\boldsymbol{w}_{j'}\hat{y}_i)$, where $\hat{y}_i$ denotes the incoming prediction to node $j'$ for input $x_i$. By inductive hypothesis, there are at most $2Tk_{r+1}p$ polynomials of degree at most $(\Delta + 1)^r + 1$ such that in each piece of the refinement of $\mathcal{P}_r$ induced by these polynomial boundaries, $\hat{y}_{ij'}$ is a fixed polynomial with degree at most $(\Delta + 1)^{r+1}$. By Warren's theorem, the number of pieces in this refinement is at most $|\mathcal{P}_{r+1}| \leq \prod_{q=1}^{r+1} 2\left(\frac{4eTk_qp(\Delta+1)^q}{W_q}\right)^{W_q}$.

Thus $f_{(X,Y)}(\alpha, \boldsymbol{w})$ is piecewise polynomial with at most $2Tp\sum_{q=1}^{L} k_q = 2mpk$ polynomial boundary functions with degree at most $(\Delta + 1)^{2L}$, and number of pieces at most $|\mathcal{P}_L| \leq \Pi_{q=1}^{L} 2\left(\frac{4eTk_q p(\Delta+1)^q}{W_q}\right)^{W_q}$. Assume that the piecewise polynomial structure of $f_{(X,Y)}(\alpha, \boldsymbol{w})$ satisfies Assumption 1, then applying Theorem 5.1 and standard learning learning theory result gives us the final claim. $\qquad\square$

### G.1.2 BINARY CLASSIFICATION CASE

In the binary classification setting, the output of the final layer corresponds to the prediction $g(\alpha, \boldsymbol{w}, x) = \hat{y} \in \mathbb{R}$, where $\boldsymbol{w} \in \mathcal{W} \subset \mathbb{R}^W$ is the vector of parameters (network weights), and $\alpha$ is the architecture hyperparameter. The 0-1 validation loss on a single validation example $\boldsymbol{x} = (X, Y)$ is given by $\mathbb{I}_{\{g(\alpha, \boldsymbol{w}, x) \neq y\}}$, and on a set of $T$ validation examples as

$$\ell_\alpha^c(\boldsymbol{x}) = \min_{\boldsymbol{w} \in \mathcal{W}} \frac{1}{T} \sum_{(x,y) \in (X,Y)} \mathbb{I}_{\{g(\alpha, \boldsymbol{w}, x) \neq y\}} = \min_{\boldsymbol{w} \in \mathcal{W}} f(\boldsymbol{x}, \boldsymbol{w}, \alpha).$$

For a fixed validation dataset $\boldsymbol{x} = (X, Y)$, the dual class loss function is given by $\mathcal{L}_c^{\mathrm{AF}} = \{\ell_\alpha^c : \mathcal{X} \to [0, 1] \mid \alpha \in \mathcal{A}\}$.

**Theorem G.1.** *Let $\mathcal{L}_c^{AF}$ denote loss function class defined above, with activation functions $o_1, o_2$ having maximum degree $\Delta$ and maximum breakpoints $p$. Given a problem instance $\boldsymbol{x} = (X, Y)$, the dual loss function is defined as $\ell_{\boldsymbol{x}}^*(\alpha) := \min_{\boldsymbol{w} \in \mathcal{W}} f(\boldsymbol{x}, \boldsymbol{w}; \alpha) = \min_{\boldsymbol{w} \in \mathcal{W}} f_{\boldsymbol{x}}(\alpha, \boldsymbol{w})$. Then, $f_{\boldsymbol{x}}(\alpha, \boldsymbol{w})$ admits piecewise constant structure. For any $\delta \in (0, 1)$, w.p. at least $1 - \delta$ over the draw of problem instances $S \sim \mathcal{D}^m$, where $\mathcal{D}$ is some distribution over $\mathcal{X}$, we have*

$$\left| \mathbb{E}_{(X,Y) \sim \mathcal{D}}[\ell_{\hat{\alpha}}((X,Y))] - \mathbb{E}_{(X,Y) \sim \mathcal{D}}[\ell_{\alpha^*}((X,Y))] \right| = \mathcal{O}\left(\sqrt{\frac{L^2 W \log \Delta + LW \log Tpk + \log(1/\delta)}{m}}\right).$$

*Proof.* As in the proof of Theorem 6.1, the loss function $\mathcal{L}_c$ can be shown to be piecewise constant as a function of $\alpha, \boldsymbol{w}$, with at most $|\mathcal{P}_L| \leq \Pi_{q=1}^{L} 2\left(\frac{4eTk_q p(\Delta+1)^q}{W_q}\right)^{W_q}$ pieces. We can apply Theorem 4.2 to obtain the desired learning guarantee for $\mathcal{L}_c^{\mathrm{AF}}$. $\qquad\square$

### G.2 DATA-DRIVEN HYPERPARAMETER TUNING FOR GRAPH POLYNOMIAL KERNELS

### G.2.1 THE CLASSIFICATION CASE

We use the following result due to Warren (1968) to establish the piecewise constant structure of the dual loss function for GCNs.

**Theorem G.2** (Warren 1968). *Suppose $N \geq n$. Consider $N$ polynomials $p_1, \ldots, p_N$ in $n$ variables of degree at most $\Delta$. Then the number of connected components of $\mathbb{R}^n \setminus \cup_{i=1}^{N} \{\boldsymbol{z} \in \mathbb{R}^n \mid p_i(\boldsymbol{z}) = 0\}$ is $\mathcal{O}\left(\frac{N\Delta}{n}\right)^n$.*

To prove Theorem 6.2, we first show that given any problem instance $\boldsymbol{x}$, the function $f(\boldsymbol{x}, \boldsymbol{w}; \alpha) = f_{\boldsymbol{x}}(\alpha, \boldsymbol{w})$ is a piecewise constant function, where the boundaries are rational threshold functions of $\alpha$ and $\boldsymbol{w}$. We then proceed to bound the number of rational functions and their maximum degrees, which can be used to give an upper-bound for the number of connected components, using G.2. After giving an upper-bound for the number of connected components, we then use Theorem 4.2 to recover learning guarantee for $\mathcal{U}$

**Lemma G.3.** *Given a problem instance $\boldsymbol{x} = (X, y, \boldsymbol{\delta}, \mathcal{Y}_L)$ that contains the vertices representation $X$, the label of labeled vertices, the indices of labeled vertices $\mathcal{Y}_L$, and the distance matrix $\boldsymbol{\delta}$, consider the function*

$$f_{\boldsymbol{x}}(\alpha, \boldsymbol{w}) := f(\boldsymbol{x}, \boldsymbol{w}; \alpha) = \frac{1}{|\mathcal{Y}_L|} \sum_{i \in \mathcal{Y}_L} \mathbb{I}_{\{\hat{y}_i \neq y_i\}}$$

*which measures the 0-1 loss corresponding to the GCN parameter $\boldsymbol{w}$, polynomial kernel parameter $\alpha$, and labeled vertices on problem instance $\boldsymbol{x}$. Then we can partition the space of $\boldsymbol{w}$ and $\alpha$ into*

$$\mathcal{O}\left(\left(\frac{(nF^2)(2\Delta+6)}{1+dd_0+d_0F}\right)^{1+dd_0+d_0F}(\Delta+1)^{nd_0}\right)$$

*connected components, in each of which the function $f(\boldsymbol{x}, \boldsymbol{w}; \alpha)$ is a constant function.*

*Proof.* First, recall that $Z = \text{GCN}(X, A) = \hat{A}\text{ReLU}(\hat{A}XW^{(0)})W^{(1)}$, where $\hat{A} = \tilde{D}^{-1}\tilde{A}$ is the row-normalized adjacent matrix, and the matrices $\tilde{A} = [\tilde{A}_{i,j}] = A + I_n$ and $\tilde{D} = [\tilde{D}_{i,j}]$ are calculated as

$$A_{i,j} = (\delta_{i,j} + \alpha)^\Delta,$$

$$\tilde{D}_{i,j} = 0 \text{ if } i \neq j, \text{ and } \tilde{D}_{i,i} = \sum_{j=1}^{n} \tilde{A}_{i,j} \text{ for } i \in [n].$$

Here, recall that $\boldsymbol{\delta} = [\delta_{i,j}]$ is the distance matrix. We first proceed to analyze the output $Z$ step by step as follow:

- Consider the matrix $T^{(1)} = XW^{(0)}$ of size $n \times d_0$. It is clear that each element of $T^{(1)}$ is a polynomial of $W^{(0)}$ of degree at most 1.

- Consider the matrix $T^{(2)} = \hat{A}T^{(1)}$ of size $n \times d_0$. We can see that each element of matrix $\hat{A}$ is a rational function of $\alpha$ of degree at most $\Delta$. Moreover, by definition, the the denominator of each rational functions are strictly positive. Therefore, each element of matrix $T^{(2)}$ is a rational function of $W^{(0)}$ and $\alpha$ of degree at most $\Delta + 1$.

- Consider the matrix $T^{(3)} = \text{ReLU}(T^{(2)})$ of size $n \times d_0$. By definition, we have

$$T_{i,j}^{(3)} = \begin{cases} T_{i,j}^{(2)}, & \text{if } T_{i,j}^{(2)} \geq 0 \\ 0, & \text{otherwise.} \end{cases}$$

  This implies that there are $n \times d_0$ boundary functions of the form $\mathbb{I}_{T_{i,j}^{(2)} \geq 0}$ where $T_{i,j}^{(2)}$ is a rational function of $W^{(0)}$ and $\alpha$ of degree at most $\Delta+1$ with strictly positive denominators. From Theorem G.2, the number of connected components given by those $n \times d_0$ boundaries are $\mathcal{O}\left((\Delta+1)^{nd_0}\right)$. In each connected components, the form of $T^{(3)}$ is fixed, in the sense that each element of $T^{(3)}$ is a rational functions in $W^{(0)}$ and $\alpha$ of degree at most $\Delta+1$.

- Consider the matrix $T^{(4)} = T^{(3)}W^{(1)}$. In connected components defined above, it is clear that each element of $T^{(4)}$ is either 0 or a rational function in $W^{(0)}, W^{(1)}$, and $\alpha$ of degree at most $\Delta + 2$.

- Finally, consider $Z = \hat{A}T^{(4)}$. In each connected components defined above, we can see that each element of $Z$ is either 0 or a rational function in $W^{(0)}, W^{(1)}$, and $\alpha$ of degree at most $\Delta + 3$.

In summary, we proved above that the space of $\boldsymbol{w}, \alpha$ can be partitioned into $\mathcal{O}((\Delta+1)^{nd_0})$ connected components, over each of which the output $Z = \text{GCN}(X, A)$ is a matrix with each element is rational function in $W^{(0}, W^{(1)}$, and $\alpha$ of degree at most $\Delta + 3$. Now in each connected component $C$, each corresponding to a fixed form of $Z$, we will analyze the behavior of $f(\boldsymbol{x}, \boldsymbol{w}; \alpha)$, where

$$f(\boldsymbol{x}, \boldsymbol{w}; \alpha) = \frac{1}{|\mathcal{Y}_L|} \sum_{i \in \mathcal{Y}_L} \mathbb{I}_{\hat{y}_i \neq y_i}.$$

Here $\hat{y}_i = \arg\max_{j \in 1, \dots, F} Z_{i,j}$, assuming that we break tie arbitrarily but consistently. For any $F \geq j > k \geq 1$, consider the boundary function $\mathbb{I}_{Z_{i,j} \geq Z_{i,k}}$, where $Z_{i,j}$ and $Z_{i,k}$ are rational functions in $\alpha$ and $\boldsymbol{w}$ of degree at most $\Delta + 3$, and have strictly positive denominators. This means that the boundary function $\mathbb{I}_{Z_{i,j} \geq Z_{i,k}}$ can also equivalently rewritten as $\mathbb{I}_{\tilde{Z}_{i,j} \geq 0}$, where $\tilde{Z}_{i,j}$ is a polynomial

in $\alpha$ and $\boldsymbol{w}$ of degree at most $2\Delta + 6$. There are $\mathcal{O}(nF^2)$ such boundary functions, partitioning the connected component $C$ into at most $\mathcal{O}\left(\left(\frac{(nF^2)(2\Delta+6)}{1+dd_0+d_0F}\right)^{1+dd_0+d_0F}\right)$ connected components. In each connected components, $\hat{y}_i$ is fixed for all $i \in \{1, \ldots, n\}$, meaning that $f(\boldsymbol{x}, \boldsymbol{w}; \alpha)$ is a constant function.

In conclusion, we can partition the space of $\boldsymbol{w}$ and $\alpha$ into $\mathcal{O}\left(\left(\frac{(nF^2)(2\Delta+6)}{1+dd_0+d_0F}\right)^{1+dd_0+d_0F} \times (\Delta + 1)^{nd_0}\right)$ connected components, in each of which the function $f(\boldsymbol{x}, \boldsymbol{w}; \alpha)$ is a constant function. $\qquad\square$

We now ready to give a proof for Theorem 6.2.

*Theorem* 6.2 (restated). Let $\mathcal{L}^{\text{GCN}}$ denote the loss function class defined above. Given a problem instance $\boldsymbol{x}$, the dual loss function is defined as $\ell_{\boldsymbol{x}}^*(\alpha) := \min_{\boldsymbol{w} \in \mathcal{W}} f(\boldsymbol{x}, \boldsymbol{w}; \alpha)) = \min_{\boldsymbol{w} \in \mathcal{W}} f_{\boldsymbol{x}}(\alpha, \boldsymbol{w})$. Then $f_{\boldsymbol{x}}(\alpha, \boldsymbol{w})$ admits piecewise constant structure. Furthermore, for any $\delta \in (0, 1)$, w.p. at least $1 - \delta$ over the draw of problem instances $S = (\boldsymbol{x}_1, \ldots, \boldsymbol{x}_m) \sim \mathcal{D}^m$, we have

$$|\mathbb{E}_{S\sim\mathcal{D}}[\ell_{\hat{\alpha}_{\text{ERM}}}(S)] - \mathbb{E}_{S\sim\mathcal{D}}[\ell_{\alpha^*}(S)]| = \mathcal{O}\left(\sqrt{\frac{d_0(d+F)\log nF\Delta + \log(1/\delta)}{m}}\right).$$

*Proof.* Given a problem instance $\boldsymbol{x}$, from Lemma G.3, we can partition the space of $\boldsymbol{w}$ and $\alpha$ into $\mathcal{O}\left(\left(\frac{(nF^2)(2\Delta+6)}{1+dd_0+d_0F}\right)^{1+dd_0+d_0F} (\Delta + 1)^{nd_0}\right)$ connected components, over each of which the function $f(\boldsymbol{x}, \boldsymbol{w}; \alpha)$ remains constant. Combining with Theorem 4.2, we have the final claim $\qquad\square$

### G.2.2 THE REGRESSION CASE

The case is a bit more tricky, since our piece function now is not a polynomial, but instead a rational function of $\alpha$ and $\boldsymbol{w}$. Therefore, we need stronger assumption (Assumption 2) to have Theorem G.5.

**Graph instance and associated representations.** Consider a graph $\mathcal{G} = (\mathcal{V}, \mathcal{E})$, where $\mathcal{V}$ and $\mathcal{E}$ are sets of vertices and edges, respectively. Let $n = |\mathcal{V}|$ be the number of vertices. Each vertex in the graph is associated with a feature vector of $d$-dimension, and let $X \in \mathbb{R}^{n \times d}$ is the matrix that contains all the vertices representation in the graph. We also have a set of indices $\mathcal{Y}_L \subset [n]$ of labeled vertices, where each vertex belongs to one of $C$ categories and $L = |\mathcal{Y}_L|$ is the number of labeled vertices. Let $y \in [-R, R]^L$ be the vector representing the true labels of labeled vertices, where the coordinate $y_l$ of $Y$ corresponds to the label vector of vertice $l \in \mathcal{Y}_L$.

**Label prediction.** We want to build a model for classifying the other unlabelled vertices, which belongs to the index set $\mathcal{Y}_U = [n] \setminus \mathcal{Y}_L$. To do that, we train a graph convolutional network (GCN) Kipf & Welling (2017) using semi-supervised learning. Along with the vertices representation matrix $X$, we are also given the distance matrix $\boldsymbol{\delta} = [\delta_{i,j}]_{(i,j)\in[n]^2}$ encoding the correlation between vertices in the graph. Using the distance matrix $D$, we then calculate the following matrices $A, \tilde{A}, \tilde{D}$ which serve as the inputs for the GCN. The matrix $A = [A_{i,j}]_{(i,j)\in[n]^2}$ is the adjacent matrix which is calculated using distance matrix $\boldsymbol{\delta}$ and the polynomial kernel of degree $\Delta$ and hyperparameter $\alpha > 0$

$$A_{i,j} = (\delta(i,j) + \alpha)^\Delta.$$

We then let $\tilde{A} = A + I_n$, where $I_n$ ia the identity matrix, and $\tilde{D} = [\tilde{D}_{i,j}]_{[n]^2}$ of which each element is calculated as

$$\tilde{D}_{i,j} = 0 \text{ if } i \neq j, \text{ and } \tilde{D}_{i,i} = \sum_{j=1}^n \tilde{A}_{i,j} \text{ for } i \in [n].$$

**Network architecture.** We consider a simple two-layer graph convolutional network (GCN) $f$ Kipf & Welling (2017), which takes the adjacent matrix $A$ and vertices representation matrix $X$ as inputs and output $Z = f(X, A)$ of the form

$$Z = \text{GCN}(X, A) = \hat{A}\,\text{ReLU}(\hat{A}XW^{(0)})W^{(1)},$$

where $\hat{A} = \tilde{D}^{-1}\tilde{A}$, $W^{(0)} \in \mathbb{R}^{d \times d_0}$ is the weight matrix of the first layer, and $W^{(1)} \in \mathbb{R}^{d_0 \times 1}$ is the hidden-to-output weight matrix. Here, $z_i$ is the $i^{th}$ element of $Z$ representing the prediction of the model for vertice $i$.

**Objective function and the loss function class.** We consider mean squared loss function corresponding to hyperparameter $\alpha$ and networks parameter $\boldsymbol{w} = (\boldsymbol{w}^{(0)}, \boldsymbol{w}^{(1)})$ when operating the problem instance $\boldsymbol{x}$ as follow

$$f(\boldsymbol{x}, \boldsymbol{w}; \alpha) = \frac{1}{|\mathcal{Y}_L|} \sum_{i \in \mathcal{Y}_L} (z_i - y_i)^2.$$

We then define the loss function corresponding to hyperparameter $\alpha$ when operating on the problem instance $\boldsymbol{x}$ as

$$\ell_\alpha(\boldsymbol{x}) = \min_{\boldsymbol{w}} f(\boldsymbol{x}, \boldsymbol{w}; \alpha).$$

We then define the loss function class for this problem as follow

$$\mathcal{L}_r^{\text{GCN}} = \{\ell_\alpha : \mathcal{X} \to [0, R^2] \mid \alpha \in \mathcal{A}\},$$

and our goal is to analyze the pseudo-dimension of the function class $\mathcal{L}_r^{\text{GCN}}$.

**Lemma G.4.** *Given a problem instance $\boldsymbol{x} = (X, y, \boldsymbol{\delta}, \mathcal{Y}_L)$ that contains the graph $\mathcal{G}$, its vertices representation $X$, the indices of labeled vertices $\mathcal{Y}_L$, and the distance matrix $\boldsymbol{\delta}$, consider the function*

$$f_{\boldsymbol{x}}(\alpha, \boldsymbol{w}) := f(\boldsymbol{x}, \boldsymbol{w}; \alpha) = \frac{1}{|\mathcal{Y}_L|} \sum_{i \in \mathcal{Y}_L} (z_i - y_i)^2.$$

*which measures the mean squared loss corresponding to the GCN parameter $\boldsymbol{w}$, polynomial kernel parameter $\alpha$, and labeled vertices on problem instance $\boldsymbol{x}$. Then we can partition the space of $\boldsymbol{w}$ and $\alpha$ into $\mathcal{O}((\Delta + 1)^{nd_0})$ connected components, in each of which the function $f(\boldsymbol{x}, \boldsymbol{w}; \alpha)$ is a rational function in $\alpha$ and $\boldsymbol{w}$ of degree at most $2(\Delta + 3)$.*

*Proof.* First, recall that $Z = \text{GCN}(X, A) = \hat{A}\text{ReLU}(\hat{A}XW^{(0)})W^{(1)}$, where $\hat{A} = \tilde{D}^{-1/2}\tilde{A}\tilde{D}^{-1/2}$ is the row-normalized adjacent matrix, and the matrices $\tilde{A} = [\tilde{A}_{i,j}] = A + I_n$ and $\tilde{D} = [\tilde{D}_{i,j}]$ are calculated as

$$A_{i,j} = (\delta_{i,j} + \alpha)^\Delta,$$

$$\tilde{D}_{i,j} = 0 \text{ if } i \neq j, \text{ and } \tilde{D}_{i,i} = \sum_{j=1}^{n} \tilde{A}_{i,j} \text{ for } i \in [n].$$

Here, recall that $\boldsymbol{\delta} = [\delta_{i,j}]$ is the distance matrix. We first proceed to analyze the output $Z$ step by step as follow:

- Consider the matrix $T^{(1)} = XW^{(0)}$ of size $n \times d_0$. It is clear that each element of $T^{(1)}$ is a polynomial of $W^{(0)}$ of degree at most 1.

- Consider the matrix $T^{(2)} = \hat{A}T^{(1)}$ of size $n \times d_0$. We can see that each element of matrix $\hat{A}$ is a rational function of $\alpha$ of degree at most $\Delta$. Moreover, by definition, the the denominator of each rational functions are strictly positive. Therefore, each element of matrix $T^{(2)}$ is a rational function of $W^{(0)}$ and $\alpha$ of degree at most $\Delta + 1$.

- Consider the matrix $T^{(3)} = \text{ReLU}(T^{(2)})$ of size $n \times d_0$. By definition, we have

$$T_{i,j}^{(3)} = \begin{cases} T_{i,j}^{(2)}, & \text{if } T_{i,j}^{(2)} \geq 0 \\ 0, & \text{otherwise.} \end{cases}$$

This implies that there are $n \times d_0$ boundary functions of the form $\mathbb{I}_{T_{i,j}^{(2)} \geq 0}$ where $T_{i,j}^{(2)}$ is a rational function of $W^{(0)}$ and $\alpha$ of degree at most $\Delta + 1$ with strictly positive denominators. From Theorem G.2, the number of connected components given by those $n \times d_0$ boundaries are $\mathcal{O}\left((\Delta + 1)^{nd_0}\right)$. In each connected components, the form of $T^{(3)}$ is fixed, in the sense that each element of $T^{(3)}$ is a rational functions in $W^{(0)}$ and $\alpha$ of degree at most $\Delta + 1$.

- Consider the matrix $T^{(4)} = T^{(3)} W^{(1)}$. In connected components defined above, it is clear that each element of $T^{(4)}$ is either $0$ or a rational function in $W^{(0)}, W^{(1)}$, and $\alpha$ of degree at most $\Delta + 2$.

- Finally, consider $Z = \hat{A} T^{(4)}$. In each connected components defined above, we can see that each element of $Z$ is either $0$ or a rational function in $W^{(0)}, W^{(1)}$, and $\alpha$ of degree at most $\Delta + 3$.

In summary, we proved that the space of $\boldsymbol{w}, \alpha$ can be partitioned into $\mathcal{O}((\Delta + 1)^{nd_0})$ connected components, over each of which the output $Z = \mathrm{GCN}(X, A)$ is a matrix with each element is a rational function in $W^{(\mathbf{0})}, W^{(\mathbf{1})}$, and $\alpha$ of degree at most $\Delta + 3$. It means that in each piece, the loss function would be a rational function of degree at most $2(\Delta + 3)$, as claimed. $\qquad\square$

**Theorem G.5.** *Consider the loss function class $\mathcal{L}_r^{GCN}$ defined above. For a problem instance $\boldsymbol{x}$, the dual loss function $\ell_{\boldsymbol{x}}^*(\alpha) := \min_{\boldsymbol{w} \in \mathcal{W}} f_{\boldsymbol{x}}(\alpha, \boldsymbol{w})$, where $f_{\boldsymbol{x}}(\alpha, \boldsymbol{w})$ admits piecewise polynomial structure (Lemma G.4). If we assume the piecewise polynomial structure satisfies Assumption 2, then for any $\delta \in (0, 1)$, w.p. at least $1 - \delta$ over the draw of $m$ problem instances $S \sim \mathcal{D}^m$, where $\mathcal{D}$ is some problem distribution over $\mathcal{X}$, we have*

$$|\mathbb{E}_{S \sim \mathcal{D}}[\ell_{\hat{\alpha}_{ERM}}(S)] - \mathbb{E}_{S \sim \mathcal{D}}[\ell_{\alpha^*}(S)]| = \mathcal{O}\left( \sqrt{\frac{nd_0 \log \Delta + d \log(\Delta F) + \log(1/\delta)}{m}} \right).$$

# H A DISCUSSION ON HOW TO CAPTURE THE LOCAL FLATNESS PROPERTIES OF BLA BLA

Our definition of dual utility function $u_{\boldsymbol{x}}^*(\alpha) = \max_{\boldsymbol{w} \in \mathcal{W}} f_{\boldsymbol{x}}(\alpha, \boldsymbol{w})$ implicitly assumes an ERM oracle. As discussed in Appendix B, this ERM oracle assumption makes the function $u_{\boldsymbol{x}}^*(\alpha)$ well-defined and simplifies the analysis. However, one may argue that assuming the ERM oracle will make the behavior of tuned hyperparameters much different, compared to when using common optimization in deep learning. The difference potentially stems from the fact that the global optimum found by ERM oracle might have a sharp curvature, compared to the local optima found by other optimization algorithms, which tend to have flat local curvature due to their implicit biases.

In this section, we consider the following simplified scenario where the ERM oracle also finds the near-optimum that is locally flat, and explain how our framework could potentially be useful in this case. Instead of defining $u_{\boldsymbol{x}}^*(\alpha) = \max_{\boldsymbol{w} \in \mathcal{W}} f_{\boldsymbol{x}}(\alpha, \boldsymbol{w})$, we define $u_{\boldsymbol{x}}^*(\alpha) = \max_{\boldsymbol{w} \in \mathcal{W}} f_{\boldsymbol{x}}'(\alpha, \boldsymbol{w})$, where the surrogate function $f_{\boldsymbol{x}}'(\alpha, \boldsymbol{w})$ is defined as follows.

**Definition 14** (Surrogate function construction). Assume that $f_{\boldsymbol{x}}(\alpha, \boldsymbol{w})$ admits piecewise polynomial structure, meaning that:

1. The domain $\mathcal{A} \times \mathcal{W}$ of $f_{\boldsymbol{x}}$ is divided into $N$ connected components by $M$ polynomials $h_{\boldsymbol{x},1}, \ldots, h_{\boldsymbol{x},M}$ in $\alpha, \boldsymbol{w}$, each of degree at most $\Delta_b$. The resulting partition $\mathcal{P}_{\boldsymbol{x}} = \{R_{\boldsymbol{x},1}, \ldots, R_{\boldsymbol{x},N}\}$ consists of connected sets $R_{\boldsymbol{x},i}$, each formed by a connected component $C_{\boldsymbol{x},i}$ and its adjacent boundaries.

2. Within each $R_{\boldsymbol{x},i}$, $f_{\boldsymbol{x}}$ takes the form of a polynomial $f_{\boldsymbol{x},i}$ in $\alpha$ and $\boldsymbol{w}$ of degree at most $\Delta_p$.

Defining the function surrogate $f_{\boldsymbol{x}}'(\alpha, \boldsymbol{w})$ as follow:

1. The domain $\mathcal{A} \times \mathcal{W}$ of $f_{\boldsymbol{x}}'(\alpha, \boldsymbol{w})$ is partitioned into $N$ connected components by $M$ polynomials $h_{\boldsymbol{x},1}, \ldots, h_{\boldsymbol{x},M}$ in $\alpha, \boldsymbol{w}$ similar to $f_{\boldsymbol{x}}$. This results in a similar partition $\mathcal{P}_{\boldsymbol{x}} = \{R_{\boldsymbol{x},1}, \ldots, R_{\boldsymbol{x},N}\}$.

2. In each region $R_{\boldsymbol{x},i}$, $f_{\boldsymbol{x}}'$ is defined as

$$f_{\boldsymbol{x}}'(\alpha, \boldsymbol{w}) = f_{\boldsymbol{x},i}'(\alpha, \boldsymbol{w}) = f_{\boldsymbol{x},i}(\alpha, \boldsymbol{w}) - \eta \|\nabla_{\boldsymbol{w},\boldsymbol{w}}^2 f_{\boldsymbol{x}}(\alpha, \boldsymbol{w})\|_F^2,$$

for some fixed $\eta > 0$. We can see that $\|\nabla^2_{\boldsymbol{w},\boldsymbol{w}} f_{\boldsymbol{x}}(\alpha, \boldsymbol{w})\|^2_F$ is a polynomial of $\alpha, \boldsymbol{w}$ of degree at most $2\Delta_p$. Therefore, $f'_{\boldsymbol{x}}(\alpha, \boldsymbol{w})$ is also a polynomial of degree at most $2\Delta_p$ in the region $R_{\boldsymbol{x},i}$.

From the above construction, we can see that $f'_{\boldsymbol{x}}(\alpha, \boldsymbol{w})$ also admits piecewise polynomial structure, where the input domain partition $\mathcal{P}_{\boldsymbol{x}}$ is the same as $f_{\boldsymbol{x}}(\alpha, \boldsymbol{w})$. In each region $R_{\boldsymbol{x},i}$, the function $f'_{\boldsymbol{x}}(\alpha, \boldsymbol{w})$ is also a polynomial in $\alpha, \boldsymbol{w}$ of degree at most $2\Delta_p$. Therefore, our framework is still applicable in this case. Moreover, construction above naturally introduces an extra hyperparameter $\eta$, which is the magnitude of curvature regularization. This makes the analysis more challenging, but for simplicity, we here assume that $\eta$ is fixed and good enough for balancing the effect of regularization.

We can see that by defining $u^*_{\boldsymbol{x}}(\alpha) = \max_{\boldsymbol{w} \in \mathcal{W}} f'_{\boldsymbol{x}}(\alpha, \boldsymbol{w})$, we can somehow capture the generalization behavior of tuned hyperparameter $\alpha$, when the solution $w^*$ of $\max_{\boldsymbol{w} \in \mathcal{W}} f'_{\boldsymbol{x}}(\alpha, \boldsymbol{w})$ is: (1) near optimal w.r.t $\max_{\boldsymbol{w} \in \mathcal{W}} f_{\boldsymbol{x}}(\alpha, \boldsymbol{w})$, and (2) locally flat.

However, the example above is an oversimplified scenario. To truly understand the behavior of data-driven hyperparameter tuning without ERM oracle, we need a better analysis to capture the behavior of $u^*_{\boldsymbol{x}}(\alpha)$ in such a scenario. This analysis should consider the joint interaction between the model, data, and the optimization algorithm, and remains an interesting direction for future work.

