# OpenReview forum: "Provable Data-driven Hyperparameter Tuning for Deep Neural Networks"
_ICLR.cc/2025/Conference — Submitted to ICLR 2025_

### Official Review · Reviewer_w2Ap · 2024-11-02

**Soundness:** 3
**Presentation:** 2
**Contribution:** 2
**Rating:** 3
**Confidence:** 3

**Summary:**

The first formal study of hyperparameter tuning with discontinuous and oscillating optimization landscape considered, introducing a new technique utilizing techniques from differential geometry and constrained optimization.

**Strengths:**

Solid paper.

The subject studied is an important one. We have seen major empirical breakthroughs in empirical worlds, including training, hyper paramter tuning, fine-tuning, etc. Theories have been hard to follow up with these progresses. Work like the paper should definitely be encouraged.

The authors apply many advanced mathematics for the topic. Writing on the theory side is clear and pretty. The authors go very deep into analyze the circumstances described by their theoretical assumptions.

**Weaknesses:**

The settings are not well-explained. "We assume that we have a series of deep learning tasks, and we have to tune hyperparameters to do well on average over the distribution of tasks." Why not fine tuning for each different task? Why is it necessary to introduce the difficulty of task distribution for the problem of hyper parameter tuning? These questions naturally arise and there's a lack of explanation.

Some sentences are hard to parse. "A major difficulty is that the loss as a function of the hyperparameter is very volatile and furthermore, it is given implicitly by an optimization problem over the model parameters. This is unlike previous work in data-driven design, where one can typically
explicitly model the algorithmic behavior as a function of the hyperparameters." I'm confused by two sentences. A volatile function is still a function of the hyperparameters. It's likely the meaning is not well conveyed.

Maths are not necessarily relevant to the topic. The introduced techniques from differential geometry, algebraic geometry are not necessarily relevant to the real difficulties of the hyper parameter tuning problems, but might just be useful because the assumptions are taken in their favor. This leads to the question, how relevant are the assumptions to the reality? The authors argue that the landscape of hyper parameter tuning is very volatile, which gives me the impression that algebraic geometry and differential geometry are too smooth to be applicable. The authors could have presented a visualization of neural network hyperparameter landscape that confirms their assumptions. This could make the whole paper much more convincing if the actual landscape is shown to obviously satisfy the assumptions in the paper. For example, this blog https://sohl-dickstein.github.io/2024/02/12/fractal.html gives a very good presentation of beautiful fractals made by neural network training. Sometimes, blogs are much better at faithfully presenting information and more polished than papers. It feels like the papers' assumptions on discontinuities are much less discontinuous than reality.

There's a lack of discussion of implications that could be useful for the empirical community. I couldn't find relevant empirical experiments done in this paper. As the nature of the subject is quite empirical, it's essential to have empirical support and evidence of significance for improving existing results.

**Questions:**

Included in weaknesses already.

Why the abstract doesn't say "we use both algebraic and differential geometry"?

---

> ### Author Response · Authors · 2024-11-23
> **Response to Reviewer w2Ap [1/3]**
>
> We thank the reviewer for spending time reviewing our paper and raising some good points. We appreciate that the __reviewer finds our paper solid, and important, and should be encouraged due to a lack of theoretical understanding of the topic__. We are glad that __the reviewer found the theory side of our paper clear and pretty__. We address the concern of the reviewer as follows
>
> ### Questions/clarity of settings/assumptions/experiments.
>
> 1. __Extra elaboration on the setting. On introducing the task distribution for the problem of hyperparameter tuning?__:
>    1.  As stated by the reviewer and mentioned in the title, we focus on the data-driven setting, which assumes that there is an application-specific problem (task) distribution $\mathcal{D}$ from which the problem instance (task) $x$ comes. In this setting, we tune the hyperparameter for the problem distribution $\mathcal{D}$, not for a single problem instance $x$. We note that this is not an uncommon setting in machine learning, and has been investigated in both theoretical and empirical sides (see [1,2,3,4,5,6,7,8,9,10] for a non-exhaustive list). __This setting naturally captures cross-validation, but is more general and also applies to multitask hyperparameter tuning [13].__
>
>    2. On introducing the task distribution:  By assuming a distribution $\mathcal{D}$ over tasks and the availability of task samples $x1, . . . , x_N$ from $\mathcal{D}$, we can provide a generalization guarantee for the hyperparameter tuned using those available tasks. Note that this setting makes sense if we have to solve multiple related tasks repeatedly [9, 10], but also captures cross-validation as a special case (where random folds of validation sets correspond to different samples drawn from a fixed training set).
>
>    3. However, we agree with the reviewer that this point should be clarified more carefully. Hence, we made the following changes to emphasize our setting and contribution:
>       1. __Title change__: We are changing the title of our paper to "__Sample complexity of data-driven tuning model hyperparameters in neural networks with piecewise polynomial dual functions__". It emphasizes that:
>          1. We __focus on analyzing the sample complexity__ when tuning hyperparameter specifically in data-driven setting,
>
>          2. We specifically __focus on tuning model hyperparameters__ (not optimization hyperparameters for example), and
>
>          3. We focus on the case where **the dual utility function $f_{x}(\alpha, w)$ admits polynomial piecewise structure**. However, we note that __this case is not uncommon__ when tuning model hyperparameter in data-driven setting, as shown in many prior works [1,2,3,4,5,6,7,8,9].
>
>        Please let us know what you think about this title.
>
>       2. __Main body changes__: We made extra clarification in the main body (l.95-101, l.147-157) and a detailed discussion in Appendix B to justify the positioning of our paper.
>       3. __Our problem is challenging and requires novel techniques__:  We note that our setting requires technical novelty compared to prior work in statistical data-driven algorithm hyperparameter tuning [1, 2, 3, 4, 13, 15]. As far as we are concerned, in most prior work [1,2,3,4], the hyperparameter tuning process does not involve the parameter $w$, meaning that given any fixed hyperparameter $\alpha$, the behavior of the algorithm is determined. In some other cases that involve parameter $w$, we can have a precise analytical characterization of how the optimal parameter behaves for any fixed hyperparameter [13] or at least a uniform approximate characterization [15]. However, our setting does not belong to those cases and requires a novel proof approach to handle the challenging case of tuning hyperparameter tuning of neural networks (see Appendix B in our revised draft for a detailed discussion).
>
> 2. __Clarifying "A major difficulty is that the loss as a function of the hyperparameter is very volatile and it is given implicitly by an optimization problem over the model parameters. This is unlike previous work in data-driven design, where one can typically explicitly model the algorithmic behavior as a function of the hyperparameters."__:
>    1. The second sentence means that in prior work [1,2,3,4,5,6,7,8,9], the structure of the function $u^*_{x} (\alpha)$ is simple, which is a closed-form piecewise polynomial/rational/. . . function of the hyperparameter $\alpha$ and the main challenge is establishing this structure.
>    2. In contrast, there are many cases where $u^*_{x}(\alpha)$ cannot be written as a function of $\alpha$ explicitly, but is implicitly defined as in our case. A natural question now would be: can we still perform data-driven hyperparameter tuning by establishing learning-theoretic guarantees in this case? That is the meaning of the first sentence and the motivation of our main results (Theorem 5.1, Lemma 4.2).
>
> We incorporated this discussion into a revised draft (Appendix B).

---

> ### Author Response · Authors · 2024-11-23
> **Response to Reviewer w2Ap [2/3]**
>
> 3. "__Differential and algebraic might only be useful because the assumptions are taken in their favor. What about the blog https://sohl-dickstein.github.io/2024/02/12/fractal.html? It looks like the discontinuities are much less discontinuous in reality.__"
>
>    We thank the reviewer for referring to the interesting blog. However, we want to clarify the following points:
>     1. Our focus is on __tuning model hyperparameters in data-driven settings__, where the dual utility function $f_{x}(\alpha, w)$ admits piecewise polynomial structures. Of course, one might think of hyperparameter tuning in DNNs in a different way, like tuning the optimization algorithm hyperparameter instead, and we agree that our analysis is not applicable in this case. But as the reviewer stated ". . . Theories have been hard to follow up with these progresses. Work like the paper should definitely be encouraged.  ", the theoretical analysis for hyperparameter tuning is challenging and it is good to study a specific setting, under some (reasonable) assumption first to have some initial result and understanding of the problem from a learning-theoretic lens.
>
>         To clarify this point, we __changed our title to "Sample complexity of data-driven tuning model hyperparameters in neural networks with piecewise polynomial dual functions"__ as mentioned above, to emphasize that we are focusing on studying the sample complexity only, focusing on a specific case of hyperparameter only. We also __incorporated multiple changes in our revised draft to clarify this point__ ( l.95-101, l.147-157, and a detailed discussion in Appendix B).
>
>    2. "__Differential and algebraic might only be useful because the assumptions are taken in their favor__":
>
>         1. Actually, the idea of using differential and algebraic geometry is inspired by the observation that the hyperparameter and parameter loss landscape $f_x(\alpha, w)$ often admits a piecewise polynomial structure.
>         2. For the simpler problem of parameter tuning it is known that the piecewise polynomial structure holds. We show that as we vary both parameter & hyperparameter the same structure holds (see section 6 on applications), but even if that is the case it is not obvious it implies generalization guarantees for hyperparameter tuning. We show that this is the main technical challenge (Sections 4 and 5). When this particular piecewise structure holds true, Assumption 1 holds almost everywhere (see l.321-323, due to a fundamental result in differential geometry (Sard’s theorem, E.10)).
>     3. "What about the blog https://sohl-dickstein.github.io/2024/02/12/fractal.html?": though this blog provides a nice visualization of the hyperparameter loss landscape, we emphasize that it is of optimization algorithm hyperparameter and is not applicable in our case. Moreover, it only provides visualization in a few example instances without any theoretical evidence, while the piecewise structure in our application is proven to hold true for any network and input instance.
>
> 4. __Lack of experiments.__: We thank the reviewer for the comment. However, the main purpose of our work is theoretical.  We note that Learning Theory is an explicit area of interest in the Call of Papers, and this paper is also listed as having Learning Theory as its Primary Area.
>
> ### Additional questions
>  1. "__Why not saying that "we use both algebraic and differential geometry__"?: Thank you for pointing it out, we have just added it in the abstract.
>
> ### Summary
> Again, we thank the reviewer for constructive feedback. We made several modifications (in the title and main body) to address the reviewer’s concern and emphasize the scope and setting of our study. We are happy to answer further questions raised by the reviewer. __We respectfully request that the reviewer reevaluate our paper in light of our rebuttal.__

---

> ### Author Response · Authors · 2024-11-23
> **Response to Reviewer w2Ap [3/3]**
>
> ### References
>
> [1] Balcan et al., How much data is sufficient to learn high-performing algorithms? Generalization guarantees for data-driven algorithm design, STOC’21
>
> [2] Balcan et al., Learning-Theoretic Foundations of Algorithm Configuration for Combinatorial Partitioning Problems, COLT’17
>
> [3] Balcan et al., Learning to Link, ICLR’20
>
> [4] Bartlett et al., Generalization Bounds for Data-Driven Numerical Linear Algebra, COLT’22
>
> [5] Balcan et al., Structural Analysis of Branch-and-Cut and the Learnability of Gomory Mixed Integer Cuts. NeurIPS’22
>
> [6] Balcan et al., Sample Complexity of Tree Search Configuration: Cutting Planes and Beyond. NeurIPS’21
>
> [7] Balcan et al., Dispersion for Data-Driven Algorithm Design, Online Learning, and Private Optimization. FOCS’18
>
> [8] Cheng and Basu, Learning Cut Generating Functions for Integer Programming NeurIPS’24
>
> [9] Cheng et al., Sample Complexity of Algorithm Selection Using Neural Networks and Its Applications to Branch-and-Cut NeurIPS’24
>
> [10] Amos et al. Meta Optimal Transport, ICML’23
>
> [13] Balcan and Sharma Provably Tuning Elastic Across Instance, NeurIPS’22
>
> [15] Balcan et al. New bounds for hyperparameter tuning of regression problems across instances, NeurIPS'24

---

> ### Comment · Reviewer_w2Ap · 2024-11-25
> **Experiments are still needed.**
>
> Thank the authors for their well written responses. Many confusion have been cleared.
>
> Still, I believe experiments are necessary. The type of topic in this paper needs experiments. This comes from my own experience of doing machine learning theory. The gap of learning theory and practice is so large that it's our responsibilities to show that the theories are relevant to the real world. And rarely in learning theory one can develop deep and profound theoretical results that have broad implications across different disciplines, which would be enough excuse for a lack of experiments. This paper doesn't qualify as such.
>
> Furthermore, it doesn't take much time to do these experiments. The absence of them might suggest the assumptions are not well suited to describe real world circumstances.

---

### Official Review · Reviewer_GxVs · 2024-11-04

**Soundness:** 2
**Presentation:** 3
**Contribution:** 2
**Rating:** 8
**Confidence:** 4

**Summary:**

This paper proves statistical learning generalization bounds for the task of optimizing architectural hyperparameters (ie those static during training) when allowed multiple problem instances. In particular, using machinery derived from piecewise decompositions of the associated utility function, the authors are able to (under some assumptions on geometric regularity of the decomposition) derive PAC-style sample complexity results for two architectural optimization applications of interest: (1) learning an optimal interpolation between piecewise-polynomial activations and (2) optimizing a certain parameterized polynomial adjacency kernel in graph convolutional networks. The proofs rely on recent statistical learning results for problems with piecewise-structured utilities, and adapt/apply these techniques to more specific and involved settings.

**Strengths:**

I think the overall approach to getting generalization bounds for this type of architecture search (which is fairly understudied) is solid. Finding piecewise/local structure (w.r.t. $\alpha$) in the costs of optimized networks, deriving learning complexity results for these piecewise structures, and bootstrapping up to a generalization bound for the overall problem feels like a very natural high-level process. Since the main statistical learning theory workhorse (the [Balcan '21] paper) expresses complexity in terms of oscillation, the authors proceed to use certain problem structures (in particular, piecewise constant, piecewise poly) to control these oscillations in a reasonable way -- the latter setting requires some clever geometric reasoning to pull off. To me, the main strength of this paper is the overall perspective taken on architecture search and the method of translating geometric structure of the meta-loss landscape to learning complexity results. Furthermore, the approximation method to relax Assumption 2 to Assumption 1 is very smooth.

**Weaknesses:**

I will list several considerations that are mainly in terms of presentation/contextualization of the results. I am sorry in advance for how long this section is, but I wanted to be helpful and thorough :)

1. This paper sets out to prove generalization bounds for the empirical risk minimizer $\hat{\alpha}$, and it does so. This is a worthwhile theoretical goal on its own (and I really like how you did it!), but I think you should be careful to separate this goal from the results that would be applicable and impactful directly to practitioners. For one, (as is often the case in statistical learning), the analysis is not structured to yield an efficient, implementable algorithm for your hyperparameter tuning nor any clues for how to design one (i.e. unless we can figure out how to find the piecewise structure dynamically and make use of it, one is still stuck doing grid search, hypergradient descent, or bandit methods). In fact, I would argue that the way you've set up your analysis is counterproductive in terms of designing such an algorithm (again, I recognize that algorithm design is not the goal of your paper, but I think the presentation would benefit from being more clear about this). For one concrete example of this point, note that your initial definition of the utility function $u_\alpha$ on p. 3 implicitly assumes finding a *global optimizer* of the loss w.r.t. the network weights, which will be wildly non convex and NP-hard for DL applications. Because of this, it cannot be claimed that $u_{\alpha_1} > u_{\alpha_2}$ implies $\alpha_1$ is a better architectural choice than $\alpha_2$ since there is no telling that an efficient optimization method (like GD) will find local minima that prefer $\alpha_1$ to $\alpha_2$. In a sense, this subtlety is the whole point of papers like [Li '21 (Geometry-Aware...)] -- a useful architecture optimization algorithm needs to improve the architecture **in a way that GD can find**. To reiterate, I understand that your paper does not set out to design such an algorithm, but I think to call the architecture optimization problem "learnable" at all requires some sort of recognition of this subtlety in a way that is clear to the reader. My philosophy is that the theoretical setting should capture the interesting phenomena you wish to explain and ignore those you don't: if you want to work in a setting that is farther from practical considerations, you should make that clear and perhaps reconsider the phrase "for deep neural networks" in the title?

2. I think this paper would benefit from being more careful about presentation. To start, the phrase "hyperparameter tuning" is often (and perhaps more ubiquitously) used to refer to tuning hyperparameters of an optimization algorithm (such as learning rate), whereas your setting is focused on a static choice of $\alpha$ during the NN training and the applications are therefore architectural. This is a bit of my own bias as an optimization theorist, but I think that provable hyperparameter tuning in this dynamic sense is quite different (and, perhaps wrongly, more studied, see "meta-optimization", "learning to learn", and many adaptive LR papers) than provable NAS, but the abstract and much of your exposition is written in a way that is vague about this difference -- it confused me in the beginning and might confuse others similarly. Perhaps more importantly, I think the claim that you provide the "first precise sample complexity bounds for applications" and "first analysis for the learnability of parameterized algorithms involving both parameters and hyperparameters" should be treated with more care. For one example, I would call Prop 4 of https://jmlr.org/papers/volume18/16-558/16-558.pdf a quantitative sample complexity bound and a (constructive) analysis of learnability -- it's not clear at first glance if your and their results are comparable or one stronger than the other, but at the very least this problem has been looked at before. I am fairly sure that you are the first to present a sample complexity bound for your two applications (and this is nontrivial through your analysis, since it requires unveiling particular piecewise structure of the applications), but I see no reason why one couldn't get some result in these applications as a corollary of Theorem 1 of the Hyperband paper. Again, we would have to see if your geometric analysis gives any advantage over such a bound -- I am not saying that your bound is worse or equivalent, I am saying that it's a slightly cavalier thing to say that your bound is the "first" in such a setting where prior results seem directly applicable.

3. In terms of the proof methodology, I would like to know more about how to control the # of piecewise components in the piecewise decompositions you use. Your Lemma D.1/E.8 are in a sense the exponential worst-case bounds, but perhaps there is more advantage to be had from a closer look into when the number of pieces is smaller? As an example, here is a cool result https://arxiv.org/pdf/1901.09021 showing that, for piecewise-linear activation functions, while the worst-case number of linear regions is exponential in the # of neurons, the average case is actually linear (proven on initialization, experimentally verified during training). Such investigations shed significant light on the relationship between complexity and expressivity as measured by # of pieces, and would really help contextualize the strength/pessimism of your bounds (and may even directly strengthen them in certain settings!). I am not sure if there are more useful results in the literature, but I think it's worth revisiting.

**Questions:**

To recap the suggestions that I kind of roped into the "Weaknesses" section, I would advise that you:
1. be more specific about the differences between your theoretical setup and the practical setting (i.e. local optima w.r.t. NN weights instead of global) and maybe highlight which parts of your analysis help toward designing practical hyperparameter-tuning algorithms
2. be more thorough and precise about the particular type of hyperparameter-tuning you study (i.e. exemplify differences with the dynamic perspective on hyperparameter-tuning) and the relationships with prior results (such as Hyperband paper and others you cite)
3. look a bit deeper into how '# of pieces' has been studied by the deep learning theorists as a measure of complexity in order to gain insight, contextualize your results better, or perhaps even strengthen them

To finish, I want to ask a question that was lingering in my head while thinking about your paper. I feel that the [Balcan '21] paper is the main statistical learning workhorse of these results, and it all boils down to the use of 'oscillation of the utility function' as the measure of learning complexity. My geometric intuition tells me that the suitable condition is specified in terms of curvature of the meta-loss landscape, perhaps through smoothness assumptions such as Lipschitz gradients (ie bounded Hessians) -- at a high level, bounded second derivatives implies reduced oscillation. One could appeal to something like the differential part of Danskin's theorem (which is similar to your Lemma E.2) to convert smoothness of the loss to smoothness of the utility and proceed that way... do you think that a plan of attack along these lines could be more direct and result in proofs that are sharper, more transparent, or more general? I feel that the polynomial formulation may be an indirect way of doing a morally similar thing -- while the polynomial structure allows you to use Bezout and co., perhaps it puts too much emphasis on things like degree and zero sets and obscures the more fundamental object of oscillation. Do you think there are sufficient analytic/geometric tools to carry out such an approach? And do you think there is something particularly special about the (piecewise-)polynomial structure w.r.t. generalization bounds, or do you view it more as a mechanical tool that we know how to prove things about?

Thank you for reading this review! Have a lovely day.

---

> ### Author Response · Authors · 2024-11-23
> **Discussion with Reviewer GxVs [1/6]**
>
> We thank the reviewer for constructive feedback and suggestions. Indeed, we find the reviewer's comments enjoyable and on point. We are glad that the reviewer finds our paper solid and likes our approaches. We will address the reviewer's concerns as follows.
>
> ## On reviewer’s comments/suggestions/questions on the paper
> 1. "__The paper provides generalization bounds for ERM $\hat{\alpha}_{ERM}$ in the data-driven setting. This is a worthwhile theoretical goal on its own (and I really like how you did it!), but I think you should be careful to separate this goal from the results that would be applicable and impactful directly to practitioners__":
>
>      __A__: We thank the reviewer for the question. We want to clarify there are actually two ERMs in our formulation, one for parameter (network weights) and another for hyperparameter tuning. The latter can be replaced by a different optimization algorithm due to uniform convergence properties (see points below) and the former is important for a clean theoretical abstraction for hyperparameter tuning that is independent of the training algorithm used.
>
>    1. If the reviewer talking about why we only prove generalization bound for $\hat{\alpha}_{ERM}$: We note that by providing the pseudo-dimension upper-bound for the utility function class $\mathcal{U}$, we have the uniform convergence guarantee that applies to any $\alpha$, i.e. we have the high probability bound for
>
>    $$\left| \sum_{i = 1}^N u_{\alpha}(x_i) - E_{x \sim \mathcal{D}} u_{\alpha}(x) \right|$$
>
>     It means that this generalization guarantee for any output $\hat{\alpha} = \mathcal{A}(x_1, \dots, x_n)$, where $\mathcal{A}$ is an algorithm that take the problem instances $x_1, …, x_n$ as the inputs, and output a hyperparameter $\hat{\alpha}$. However, if you want to compete with the best hyperparameter, i.e.  $\alpha^* = \min_{\alpha}E_{x \sim \mathcal{D}}u_{\alpha}(x)$, you should consider the empirical risk minimizer $\alpha_{ERM}$.
>
>    2. __If the reviewer talking about the ERM oracle that we use when defining $u^*_{x}(\alpha) = \min_{w \in \mathcal{W}}f_x(\alpha, w)$__: It is true that we are assuming an ERM oracle here, and this does not imply an efficient algorithm to compute $u^*_{x}(\alpha)$.  However, as the reviewer stated, it is a worthwhile theoretical goal on its own, because we want to know if our setting is reasonable in a learning-theoretic sense in the first place, before even considering if it can be efficiently learnable. After determining the learnability of the problem, we can employ many common techniques to improve the computational tractability of the learning problem, for example using a surrogate loss with better properties, or using an approximate function class, etc. which might be of interest to the practitioners.
>
> 	However, we agree that this point should be more clear to the reader, and we made changes in the main body and the title to emphasize as suggested by the reviewer to clarify this point. The changes are listed in the details below.

---

> ### Author Response · Authors · 2024-11-23
> **Discussion with Reviewer GxVs [2/6]**
>
> 2.  "__Need to be more careful about the presentation as people often think of hyperparameter tuning in DNNs as tuning optimization algorithm hyperparameters rather than model hyperparameters like NAS. Maybe consider changing the title.__":
>
>       __A__: It is true that our framework does not capture the case where the hyperparameter tuned is of optimization algorithms, and we are assuming an ERM oracle for our analysis. __We agree with the reviewer that changing the title and more clarification will make the paper more clear in this point. Here is our modification__:
>
>      1. __Title change__: We are changing the title of our paper to "__Sample complexity of data-driven tuning model hyperparameters in neural networks with piecewise polynomial dual functions__". It emphasizes that:
>          1. We __focus on analyzing the sample complexity__ when tuning hyperparameter specifically in data-driven setting,
>
>          2. We specifically __focus on tuning model hyperparameters__ (not optimization hyperparameters for example), and
>
>          3. We focus on the case where **the dual utility function $f_{x}(\alpha, w)$ admits polynomial piecewise structure**. However, we note that __this case is not uncommon__ when tuning model hyperparameter in data-driven setting, as shown in many prior works [1,2,3,4,5,6,7,8,9].
>
>        Please let us know what you think about this title.
>
>       2. __Main body changes__: We made extra clarification in the main body (l.95-101, l.147-157) and a detailed discussion in Appendix B to justify the positioning of our paper.
>       3. __Our problem is challenging and requires novel techniques__:  We note that our setting requires technical novelty compared to prior work in statistical data-driven algorithm hyperparameter tuning [1, 2, 3, 4, 13, 15]. As far as we are concerned, in most prior work [1,2,3,4], the hyperparameter tuning process does not involve the parameter $w$, meaning that given any fixed hyperparameter $\alpha$, the behavior of the algorithm is determined. In some other cases that involve parameter $w$, we can have a precise analytical characterization of how the optimal parameter behaves for any fixed hyperparameter [13] or at least a uniform approximate characterization [15]. However, our setting does not belong to those cases and requires a novel proof approach to handle the challenging case of tuning hyperparameter tuning of neural networks (see Appendix B in our revised draft for a detailed discussion).

---

> ### Author Response · Authors · 2024-11-23
> **Discussion with Reviewer GxVs [3/6]**
>
> 3. "__It might be cavalier to say that this bound is the first in such a setting where prior works might be applicable.	I see
> no reason why one couldn’t get some results in these applications as a corollary of Theorem 1 of the Hyperband paper.
> . . . What about Proposition 4 in Hyperband?__"
>
>       __A__: The setting of Hyperband is significantly different from ours, especially in the following points:
>       1. Most results (including Thm 1 and Prop 4) in Hyperband assume finitely many distinct hyperparameter values (arms) and guarantees with respect to the best arm in that set. Even their infinite arm setting considers a distribution over the hyperparameter space from which n arms are sampled. It is assumed that n is large enough to sample a good arm with high probability without actually showing that this holds for any concrete hyperparameter loss landscape. It is not clear why this assumption will hold in our case. In sharp contrast, we seek optimality over the entire continuous hyperparameter range for concrete loss functions that satisfy a piecewise polynomial dual structure.
>
>       2. The notion of “sample complexity” in Hyperband is very different from ours. Intuitively, their goal is to find the best hyperparameter from learning curves over fewest training epochs, assuming the test loss converges to a fixed value for each hyperparameter after some epochs. By ruling out (successively halving) hyperparameters that are unlikely to be optimal early, they speed up the search process (by avoiding full training epochs for suboptimal hyperparameters). In contrast, we focus on model hyperparameters and assume the network can be trained to optimality for any value of the hyperparameter. We ignore the computational efficiency aspect and focus on the data (sample) efficiency aspect which is not captured in Hyperband analysis.
>
>      3. Learning setting: Hyperband assumes the problem instance is fixed, and aims to accelerate the random search of hyperparameter configuration for that problem instance with constrained budgets (formulated as a pure-exploration non-stochastic infinite-armed bandit). In contrast, our results assume a problem distribution $\mathcal{D}$ (data-driven setting) and bounds the sample complexity of learning a good hyperparameter for the problem distribution $\mathcal{D}$.
>
>     __Conclusion__. The Hyperband paper and our work do not compete but complement each other, as the two papers see the hyperparameter tuning problem from different perspectives and our results cannot be compared to theirs. We only claim that our analysis is unique in the data-driven setting. As the reviewer suggested, we (1) changed the title of our work to "Sample complexity of data-driven tuning model hyperparameters in neural networks with piecewise polynomial dual functions", and (2) added clarification about this point in the revised draft, removed "the first analysis" claim not to confuse future readers. The changes are also reflected in Discussion 2 above and in our revised draft.

---

> ### Author Response · Authors · 2024-11-23
> **Discussion with Reviewer GxVs [4/6]**
>
> 4. "__A more sophisticated way to control the number of piecewise components (for example, Complexity of Linear Regression in Deep Networks) could help.__"
>
>      __A__. We thank the reviewer for pointing out this interesting paper. However, we want to clarify the following crucial points:
>      1. The piecewise structure in the paper __Complexity of Linear Regression in Deep Networks__ is fundamentally different from the piecewise structure we are considering. Concretely, the piecewise linear structure in that paper is of the input space, obtained when one fixes the parameters of the neural networks. In contrast, we consider the piecewise polynomial structure of the function $f_x(\alpha, w)$ of the space of hyperparameter α and parameter w, obtained when one fixes the input problem instance $x$. Therefore, the technique introduced in that paper does not apply to our scenario.
>      2. Furthermore, to the best of our knowledge, there is no other technique that gives a more refined bound for the number of connected components (and boundaries) that is applicable in our scenario (the piecewise polynomial structure in the parameter and hyperparameter space).
>
>     Nevertheless, we agree with the reviewer that a more sophisticated result controlling the number of pieces and boundaries could help improve our Theorem 6.1, 6.2 in Section 6, Applications. That is also why we express our main result (Theorem 4.2, 5.1), which gives generalization guarantees in the form of a number of regions and boundaries. We are not ruling out the existence of other more advanced connected components that could be helpful, and we are open to hearing about the reviewer’s suggestions!
>
>     Furthermore, we believe that developing average-case analyses for parameter space partitioning (analogous to the paper Complexity of Linear Regression in Deep Networks, but in parameter space instead of input space) could be an interesting direction in the future. However, we think that providing generalization guarantees using such results would require fundamentally different techniques than those presented in our paper.

---

> ### Author Response · Authors · 2024-11-23
> **Discussion with Reviewer GxVs [5/6]**
>
> ## On other comments/questions
> 1. __Is oscillation the main statistical learning workhorse?__:
>
>     __A__: Prior work provided sample complexity if the duals have bounded oscillations. We build on that result, however, the main technical challenge is to show that bounded oscillation holds for us (see Appendix B in the revised draft, or discussion above for details). We actually can present our supporting lemmas (Lemma 3.1, 3.3, bounding the pseudo-dimension with the discontinuities and local maxima of dual function) without mentioning oscillations at all but: (1) it will make the presentation less elegant, (2) we want to give credit to prior work and establish a clear relation with it.
>
>     __What is the true workhorse?__: the main challenge in our work is to control the number of discontinuities and local maxima of the dual function $u^*_{x} (\alpha) = \max_w f_x(\alpha, w)$. The __novel technical tools__ are:
>      1. Differential geometry:
>         1. allows us to determine the potential shape (smooth 1-manifolds) of (α, w) that solves $\max_w f_x(\alpha, w)$ when $\alpha$ changes,
>
>         2. allows us to rewrite $u^∗_{x} (\alpha)$ as the pointwise maximum of $f_x(\alpha, w)$ along the monotonic curves (Definition 12), and
>
>         3. allows us to control the discontinuities/local maxima of $u^∗_{x} (\alpha)$ using the property of $f_(\alpha, w)$ along the monotonic curves. __We note that most of our supporting results (Definition 12, Lemma E.16, Proposition E.12) are not even readily available in the machine learning literature, which also implies the novelty and challenge of our work.__
>      2. Algebraic geometry: combining with point (3) above, allowing us to give upper bounds on the number of discontinuities and local extrema, leveraging the piecewise polynomial structure of $f_x(\alpha, w)$.
>      3. Tools from constrained optimization (Lagrangian method).
> We added this discussion in the main body (l.147-157) and in the appendix (Appendix B). Moreover, the proof methodology is also not trivial, but another separate contribution of our work. We believe that this proof methodology is helpful in future theoretical work on data-driven algorithm design in general, not restricted to tuning model hyperparameters in DNNs.
>
> 2. "__Could the local curvature of meta-loss landscape control the oscillations?__":
>
>  __A__: It is an interesting point, but we doubt that it is true. For example, consider the function $v_\epsilon(\alpha) = \epsilon sin(\alpha)$, which has the bounded derivative everywhere and the bound can be made arbitrarily small (by decreasing ϵ). However, for any $\epsilon$, the function has infinite oscillations. Moreover, in our case, the function $u^*_{x}(\alpha)$ does not typically have nice properties on the curvature, for example, the local smoothness does not typically hold at the piece boundaries e.g. when using a ReLU activation function.
>
>  In our analysis, we show that the number of discontinuities (which includes non-smooth points) and local maxima in the meta-loss landscape are related to oscillations. We do not rule out that the information on the curvature can somehow be used to get generalization, and we think that it is an interesting question for potentially interesting restricted settings (e.g. smooth activations).
>
> 3. "__Is the polynomial formulation an indirect way of doing a morally similar thing? Is there something particularly special about the piecewise polynomial structure w.r.t generalization bound? Do you view it more as a mechanical tool that we know how to prove things around__"
>
>    __A__:  We expect that our techniques can be extended beyond the polynomial formulation, but that could require further technical work e.g. appropriate extensions of Bezout’s and Warren’s theorems to more general functions.
>
> ## Summary
> Overall, we thank the reviewer for the constructive feedback to improve the presentation of the paper, and we think that it is a very enjoyable and productive discussion! We have made several modifications (title and main body) to address the reviewer’s concerns, and we are happy to answer/discuss more points raised by the reviewer if needed. __We respectfully request that the reviewer reevaluate our paper in light of our comments above.__ Have a lovely day!

---

> ### Author Response · Authors · 2024-11-23
> **Discussion with Reviewer GxVs [6/6]**
>
> ### References
>
> [1] Balcan et al., How much data is sufficient to learn high-performing algorithms? Generalization guarantees for data-driven algorithm design, STOC’21
>
> [2] Balcan et al., Learning-Theoretic Foundations of Algorithm Configuration for Combinatorial Partitioning Problems, COLT’17
>
> [3] Balcan et al., Learning to Link, ICLR’20
>
> [4] Bartlett et al., Generalization Bounds for Data-Driven Numerical Linear Algebra, COLT’22
>
> [5] Balcan et al., Structural Analysis of Branch-and-Cut and the Learnability of Gomory Mixed Integer Cuts. NeurIPS’22
>
> [6] Balcan et al., Sample Complexity of Tree Search Configuration: Cutting Planes and Beyond. NeurIPS’21
>
> [7] Balcan et al., Dispersion for Data-Driven Algorithm Design, Online Learning, and Private Optimization. FOCS’18
>
> [8] Cheng and Basu, Learning Cut Generating Functions for Integer Programming NeurIPS’24
>
> [9] Cheng et al., Sample Complexity of Algorithm Selection Using Neural Networks and Its Applications to Branch-and-Cut NeurIPS’24
>
> [13] Balcan et al. Provably Tuning Elastic Across Instance, NeurIPS’22
>
> [15] Balcan et al. New bounds for hyperparameter tuning of regression problems across instances, Neurips’23

---

> ### Comment · Reviewer_GxVs · 2024-11-25
> **Response to Authors**
>
> I would like to thank the authors for a very thorough response and adjustment to the paper! The change in title and addition of some clarifying information regarding the positioning of your results in relation to existing perspectives and approaches addresses my main concern about presentation. I think this work is valuable to the community for being a particular instance of a more general question (how the structure of a base learning problem can inform us about complexity of meta-learning) that makes strong use of the assumptions (polynomial structure and remaining static for each learning instance, i.e. model hyperparam not optimization param) to answer the question well. Now that it's phrased as such, I am happy to increase my score to a 6! :)
>
> However, I think my question about the effects of assuming a global minimizer to each learning task (the inner ERM oracle that learns model parameters) still stands. To me, it is not just a question about efficiency, tractability, or even finding/analyzing an algorithm that implements it -- rather, I feel that the nature of how the meta-loss changes w.r.t. hyperparameter $\alpha$ can be qualitatively different when evaluating the meta-loss at e.g. stationary points vs a global minimizer. Imagine an optimization problem with very reasonable local minima and one super sharp, difficult to find global minimum (such as a spurious one that interpolates the training data but generalizes poorly). If the model is parameterized in a way that changing the hyperparameter $\alpha$ (e.g. an interpolation parameter for the activation function) affects this spurious global minimum a lot but leaves the more reasonable local minima alone, you would see a very different behavior of the meta-loss landscape.
>
> I know that this is vague, perhaps a precise formulation would be something like: Consider a setting where each learning problem is to fit a degree-$n$ polynomial $\sum_j a_jx^j$ to some noisy data (the coefficients $a_j$ are some nonconvex, but simple function $g$ of the model parameters $\theta$, and there is some model hyperparameter $\alpha$ that tweaks these so that $a_j = g(\theta_j, \alpha)$). For $n$ data points, there may be polynomials $(a_j)_j$ that are quite nice and appropriately represent the data in a reasonable way, but the global minimum may be some $(\tilde{a}_j)_j$ that is very spurious and tries to interpolate the data (depending on choice of $g$, it may not be able to do so) -- these are expressed via a quite reasonable $\theta$ and a very bizarre $\tilde{\theta}$, respectively. If we are looking at a reasonable local minimum, changing the model hyperparameter $\alpha$ to $\alpha'$ may change the loss $L(\theta, \alpha)$ very smoothly and nicely, but change $L(\tilde{\theta}, \alpha)$ in a completely different way. Once you deploy your machinery on controlling discontinuities, extrema, and oscillations w.r.t. $\alpha$, you may find that the meta-learning problem has a very different complexity when the meta-loss is $\alpha \mapsto L(\tilde{\theta}, \alpha)$ as opposed to $\alpha \mapsto L(\theta, \alpha)$.  Still vague, but I hope this gets my point across.
>
> To me, the potential for this instability is one of the reasons that hyperparameter tuning can be so finicky -- an approximate solution to the inner learning problem can behave very differently to changes in the hyperparameter than an exact one. I am not asking you to introduce any surrogate losses or track implicit biases of optimization procedures or anything like that, but I would appreciate any clarity you are able to provide for how to think about this. Do you think your machinery would have some robustness to this, and more broadly are there any techniques that could be used in follow-up works to capture it? It would be nice to include a sentence or footnote or something pointing out the subtlety surrounding this assumption, since otherwise I feel the reader has to work hard to notice it.

---

> > ### Author Response · Authors · 2024-11-27
> > **Response to Reviewer GxVs**
> >
> > We thank the reviewer for their constructive feedback. We understand the point raised by the reviewer, where the implicit bias (flatness seeking) property of optimization algorithms for the inner optimization problem might potentially have strong effects on the tuned hyperparameter of the outer optimization problem. We acknowledge this point, but as the reviewer pointed out, providing theoretical analysis of hyperparameter tuning is a very challenging problem and it is good to start with some reasonable formulation.
> >
> > Moreover, we suggest that __our framework might still be useful for analyzing the generalization of hyperparameter tuning, where the local flatness is also considered__. Consider the following (over)simplified scenario: instead of optimizing $f_x(\alpha, w)$ for the inner optimization, we optimize a surrogate $f’x(\alpha, w)$. The surrogate has the same discontinuity structure as the original function, but in each region $R_{x, i}$ where $f_x(\alpha, w)$ admits the polynomial form $f_{x, i}(\alpha, w)$, the value of the surrogate is $f_{x, i}(\alpha, w)$ minus a curvature regularization term (Hessian norm). __See Appendix H for the detailed construction__. We can see that by optimizing this surrogate function instead, we can capture the locally flat behavior suggested by the reviewer within our analytical framework. Because our main result is general, we can instantiate a generalization guarantee for this case, because note that the regularization term is also a polynomial of $\alpha, w$, meaning that $f’_x(\alpha, w)$ also admits piecewise polynomial structure. We are happy to incorporate this discussion with the reviewer to the final version, as well as a discussion on how to better model the phenomenon pointed out by the reviewer (since the above is just a simplified scenario).

---

> > > ### Comment · Reviewer_GxVs · 2024-11-27
> > >
> > > Thank you for the thorough addressing of my concerns. I now have no significant criticisms, and I am happy to increase my score to 8. I think this paper’s methods are a useful and flexible contribution to the field. Happy Thanksgiving!

---

> > > > ### Author Response · Authors · 2024-11-28
> > > > **Response to Reviewer GxVs**
> > > >
> > > > We thank the reviewer for constructive feedback and the positive evaluation of our paper. We believe that through the discussion of the reviewer, the presentation of our work was greatly improved. Thank you and happy Thanksgiving!

---

### Official Review · Reviewer_4928 · 2024-11-04

**Soundness:** 2
**Presentation:** 3
**Contribution:** 2
**Rating:** 6
**Confidence:** 2

**Summary:**

The paper proposes a learning theoretic approaches to hyperparameter optimization. They show that using results from learning theory, it is possible to estimate the error of hyperparameter optimization in the case where the function $f(x, \alpha, \omega)$, representing the neural network, is piecewise constant or piecewise polynomial. The authors prove that this assumption is true for two instances of hyperparameter optimization for deep neural networks, providing corresponding learning guarantees.

**Strengths:**

- Applying a learning theoretic approach to hyperparameter optimization is original, challenging and novel.
- Hyperparameter optimization is a very heuristic research domain and it is refreshing to see some efforts towards more principled understanding and characterisation of the problem.
- The paper presents an impressive piece of theoretical work.
- The paper is well structured

**Weaknesses:**

I would like to state that I was not able to check the proofs completely since I am not an expert in learning theory. I think that this paper would at least require an examination of these proofs by an expert in the field before acceptance.

### Major

- My main problem is about the positioning of the paper. The paper claims to provide guarantees for hyperparameter optimization for deep neural networks, but in reality, it provides guarantees:
	- In a setting that is unusual for hyperparameter optimization, i.e. the setting where $\alpha_{ERM}$ is obtained after a sampling  of several whole datasets from $\mathcal{D}$ (see l.163), whereas usually hyperparameter optimization is performed for one single dataset. So, this setting does not correspond to reality, diminishing the impact of the work. In addition, the paper focuses on the optimization of one single hyperparameter, which misses the stakes and challenges of hyperparameter optimization that are more about the large number of hyperparameters and their interactions.
	- It is not applicable to "hyperparameter optimization of deep neural nets" but rather to (i) the interpolation parameter of activation function in a (debatable, see questions) application of one hyperparameter opt algorithm called DARTS (2) kernel parameter of graph neural nets. They are two very specific and not-so-common instances of hyperparameter optimization, and each of them required significant theoretical work to prove that they match the piecewise constant / polynomial assumptions.
*  l.177 $u_{\alpha}$ is computed by solving an optimization problem. This problem is stochastic by nature, whereas $u_{\alpha}$ is presented as a function (see questions)
- Some problems in the proofs:
	- **(potentially serious)** I did not find any proof of Lemma 3.3 in the Appendix, whereas it seems central, and the authors clearly state that Lemma 3.1 is not applicable in the case of Lemma 3.3 (so it would need a proof even more).
	- **(presentation)** Warren's theorem used in the proof of Th. 6.1 but no references, we don't know what it is. It is not a standard Theorem.
	- **(presentation)** l.926, 1424, 1476 "standard learning theory results gives us..." which standard learning theory result ??
### Minor

- The presentation could be improved:
	- l.56: random search methods [...] only work for a discrete and finite grid" this is not true.
	- l.93-95 sentence not correct
	- l.249 state that the results holds thanks to Th. 2.1
	- l. 258 piece function not defined, $c_i$ is introduced and no no longer used (why not defining as in 5?)
	- l.282 Theorem 4.1 you meant Lemma 4.1 right ?
	- l. 308 Notation $R_{x,t}$ different as before
	- l.309 "behaves regularly" not defined
	- l. 310 preimage introduced but not defined (defined below but you should define it prior to using it)
	- Assumption 1 is really difficult to grasp. The authors say that it is "relatively mild" but it is not at all conveyed by the presentation.

**Questions:**

- What is the link between assumtion 1 and deep neural networks?
- Can you calrify the link with DARTS? To my understanding, what they do is not what is stated in the paper. They use weights to encode the probability of using $o_1$ and $o_2$, to make the NAS differentiable. It is not an interpolation.
- l.177 $u_{\alpha}$ is computed by solving an optimization problem. This problem is stochastic by nature, whereas $u_{\alpha}$ is presented as a function. How do you cope with this stochasticity in your analysis ?

---

> ### Author Response · Authors · 2024-11-23
> **Reply to Reviewer 4928 [1/3]**
>
> We thank the reviewer for constructive feedback. We appreciate that the reviewer finds our paper __novel, well-structured, and impressive piece of theoretical work__. Some main concerns of the reviewer are about the __position of the paper__, and the __clarification of the proofs__, which we address below.
>
> ## On major clarification/paper positioning
>
> ### A. Paper positioning:
> The author raises some good points about how we should position our work. Though to some degree we agree with the reviewer, we also want to clarify the following points:
>
> 1. On our hyperparameter tuning setting:
>    1.  It is true that we are focusing on the __data-driven hyperparameter tuning settings__, as stated in the title. In this setting, one can think of tuning $\alpha_{ERM}$ using multiple problem instances $x_1, \dots, x_N$ drawn from an application-specific problem distribution $\mathcal{D}$. The problem instance $x_i$ could be a dataset as a reviewer stated, but it could also be something more simple, such as random validation folds from a fixed training set (usual cross-validation).
>
>        __This setting is not uncommon__ in machine learning in machine learning (see [1,2,3,4,5,6,7,8,9,12,17] for a non-exhaustive list of examples, including clustering/semi-supervised learning/decision trees ...). __This setting naturally captures cross-validation, but is more general and also applies to multitask hyperparameter tuning__ [12].
>     2. __To position the paper better as the reviewer suggested, we made the following changes__ (marked in red in our revised draft):
>           1. __Title change__: We are changing the title of our paper to "__Sample complexity of data-driven tuning model hyperparameters in neural networks with piecewise polynomial dual functions__". It emphasizes that:
>                1. We __focus on analyzing the sample complexity__ when tuning hyperparameter specifically in data-driven setting,
>                2. We specifically __focus on tuning model hyperparameters__ (not optimization hyperparameters for example), and
>                3. We focus on the case where **the dual utility function $f_{x}(\alpha, w)$ admits polynomial piecewise structure**. However, we note that __this case is not uncommon__ when tuning model hyperparameter in data-driven setting, as shown in many prior works [1,2,3,4,5,6,7,8,9].
>           2. __Main body changes__: We made extra clarification in the main body (l.95-101, l.147-157) and a detailed discussion in Appendix B to justify the positioning of our paper.
>     3. __Our problem is challenging and requires novel techniques__:  We note that our setting requires technical novelty compared to prior work in statistical data-driven algorithm hyperparameter tuning [1, 2, 3, 4, 13, 15]. As far as we are concerned, in most prior work [1,2,3,4], the hyperparameter tuning process does not involve the parameter $w$, meaning that given any fixed hyperparameter $\alpha$, the behavior of the algorithm is determined. In some other cases that involve parameter $w$, we can have a precise analytical characterization of how the optimal parameter behaves for any fixed hyperparameter [13], or at least a uniform approximate characterization [15]. However, our setting does not belong to those cases and requires a novel proof approach to handle the challenging case of tuning hyperparameter tuning of neural networks (see Appendix B in our revised draft for a detailed discussion).
>
> 2. "Is the result only applicable to (1) interpolation hyperparameter of activation function, and (2) kernel parameter of graph neural networks?"
>      1. We note that our main results (Theorem 5.1, Theorem 4.2) are applicable for model hyperparameter tuning problems for which the dual function $f_x(\alpha, w)$ admits a piecewise constant/polynomial structure. We have worked out implications in two interesting cases, but we expect it will apply in other settings as well.
>      2. Prior work [11, 14] has shown that the network function $f_x(\alpha)$ as a function of just the parameter on a fixed instance $x$ has a piecewise polynomial structure. So we expect our techniques to be useful for any hyperparameter for which dual function $f_x(\alpha, w)$ also possesses a piecewise polynomial structure.
>      3. However, we agree with the reviewer that __our result cannot capture all the scenarios of hyperparameter tuning in DNNs__, but rather focus on model hyperparameters in the data-driven setting. As suggested by the reviewer, we made __changes in the title and main body to clarify this point (see above for details).__
>      4. As the reviewer stated, ". . . Hyperparameter optimization is a very heuristic research domain and it is refreshing to see some efforts towards more principled understanding and characterization of the problem . . . ". Though our work focuses on specific scenarios, we believe that it still benefits future research on theoretical understanding of hyperparameter tuning.

---

> ### Author Response · Authors · 2024-11-23
> **Reply to Reviewer 4928 [2/3]**
>
> ### B. "$u_{\alpha}(x)$ is computed by solving an optimization problem, but it is also presented as a function, how could it be?"
>
> We do not quite understand this question, but will answer it with any potential meaning we can think of:
> 1. If it is about the stochasticity of problem instance $x$: it is true that the problem instance $x \sim D$ is drawn from the problem distribution $\mathcal{D}$ over the set of problem instance $\mathcal{X}$. However, given any realization of problem instance $x$, the function $u_x(\alpha)$ is defined deterministically as $u^*_x(\alpha)$ = \min_{w \in \mathcal{W}} f_{x}(\alpha, w). In other words, given a fixed problem instance $x$, there would be no randomness in the definition of $u^*_x(\alpha)$.
>
> 2. If it is about the stochasticity of optimization algorithm for solving $u^*_x(\alpha)$:
>     1. By defining $u_x(\alpha) = \min_{w \in \mathcal{W}}f_x(\alpha, w)$, we assume that we are using an ERM oracle here. We will make sure to emphasize this point again in the revised draft. Besides, we note that it is quite common in machine learning theory (see [18] for example).
>     2. Besides, as the reviewer stated, a theoretical understanding of hyperparameter tuning is challenging, and applying a learning theoretic approach is even more original, challenging, and novel. We believe the reviewer will agree that taking initial steps towards a challenging direction requires some original foundation/assumption.
>     3. We added the clarification about the ERM oracle and its necessity in the revised draft (l.948-l.951).
>
> We hope that the above is what the reviewer is talking about, and we are happy to be clarified by the reviewer if that is not the case.
>
> ### C. Other concerns of the proofs:
> 1. "Missing proof of Lemma 3.3."  Sorry for confusing the reviewer. The proof of Lemma 3.3 is straightforward and can be directly derived from the oscillation definition (Definition 1), which is why we did not incorporate it in the main paper. As the reviewer requested, we reincorporated it into the revised version (See Appendix C).
> 2. "What is Warren’s theorem in the proof of Theorem 6.1.". We were referring to the Lemma E.8. (Warren). To make it more clear, we added an explicit reference to that lemma in the proof.
> 3. "l.926, 1424, 1476, which standard learning theory results?." Sorry for confusing the reviewer, we are talking about the results summarized in Appendix C. Additional background on learning theory. We added the references to that appendix section as requested.
>
> ## On other minor comments on the presentation
> 1. "l93-95 typos": thank you for pointing it out. We were missing the part "... admits a specific piecewise structure." We have added that part in the revised version.
> 2. "l.282 Theorem 4.1 you meant Lemma 4.1 right?": that is correct, sorry for the typo. We fixed it in the revised draft.
> 3. "Why Assumption 1 is mild? What is the intuition behind it?":
>    1. __Intuition__: Assumption 1 is about the regularity of the boundary functions, which are frequently mentioned as "general positions" in algebraic geometry literature. Roughly speaking, it says that the intersections of boundary functions behave regularly, for example, the intersection of two hyperplanes in 3-dimension space should be a line, or the intersection of two lines in 2-dimension should be a point, etc.
>    2. __Why it is mild?__: as mentioned in l.322-349, due to Sard’s Theorem (Theorem E.10), the set of non-regular values (basically determining where the non-regularity of boundary functions occur) has Lebesgue measure zero. It generally means that Assumption 1 almost always holds.
> 4. Other clarification (e.g. preimages, . . . ): We made changes in the main draft to clarify the points raised by the reviewer.

---

> ### Author Response · Authors · 2024-11-23
> **Reply to Reviewer 4928  [3/3]**
>
> ## Other questions
> 1. "What is the link between Assumption 1 and DNNs?": We will break it down as follows:
>     1. The objects (problem instances, piece, and boundary functions) considered in Assumption 1: As in our data-driven setting, there is a problem distribution $\mathcal{D}$ where the problem instance $x$ comes from. Going back to Assumption 1, it puts a condition on the boundary and piece functions $f_{x, i}$, $h_{x, i}$, induced by the structure of the utility function $u^*_x(\alpha)$ (or $f_x(\alpha, w)$), for a realization of problem instance $x$. In the case of DNNs (like two examples that we consider in Section 6: Application), the piece and boundary functions dictate the piecewise polynomial structure of hyperparameter $\alpha$ and parameter $w$ loss landscape.
>     2. "The link between Assumption 1 and DNNs": in the case of DNNs, Assumption 1 essentially assumes that the piece and boundary function $f_{x, i}$, $h_{x, i}$ dictating the loss landscape of hyperparameter $\alpha$ and parameter $w$ are in general position. See the discussion above about the intuition of Assumption 1, and why it is mild.
>
> 2. "Clarifying the link with DARTS [12]?": as the reviewer commented, it is true that the activation function interpolation setting is not exactly what the DARTS paper does, but rather a simplified version of DARTS. Instead of using probabilistic interpolation as in DARTS as pointed out by the reviewer, we consider a linear interpolation to simplify the setting. We never claim to solve the DARTS setting, but only a simplified setting motivated by DARTS, as mentioned in l.429-431, l.846. We emphasized this point again in the revised draft. (l.848-851)
>
> ## Summary
> Overall, we thank the reviewer for constructive feedback and for raising good points on how we should clarify/position our contribution. As the reviewer suggested, we __made changes in both the title and the main body of the paper to clarify our contribution and scope__, as well as fix the typos pointed out by the reviewer. We hope that our answers and changes address the reviewer's concerns, and we __kindly request that the reviewer reevaluate our paper in light of our rebuttal__. After all, as the reviewer stated, there is a lack of theoretical understanding in this hyperparameter tuning direction, and we believe that our work would serve as a good starting point and would benefit future research for theoretical understanding of hyperparameter tuning.
>
> ### References
>
> [1] Balcan et al., How much data is sufficient to learn high-performing algorithms? Generalization guarantees for data-driven algorithm design, STOC’21
>
> [2] Balcan et al., Learning-Theoretic Foundations of Algorithm Configuration for Combinatorial Partitioning Problems, COLT’17
>
> [3] Balcan et al., Learning to Link, ICLR’20
>
> [4] Bartlett et al., Generalization Bounds for Data-Driven Numerical Linear Algebra, COLT’22
>
> [5] Balcan et al., Structural Analysis of Branch-and-Cut and the Learnability of Gomory Mixed Integer Cuts. NeurIPS’22
>
> [6] Balcan et al., Sample Complexity of Tree Search Configuration: Cutting Planes and Beyond. NeurIPS’21
>
> [7] Balcan et al., Dispersion for Data-Driven Algorithm Design, Online Learning, and Private Optimization. FOCS’18
>
> [8] Cheng and Basu, Learning Cut Generating Functions for Integer Programming NeurIPS’24
>
> [9] Cheng et al., Sample Complexity of Algorithm Selection Using Neural Networks and Its Applications to Branch-and-Cut NeurIPS’24
>
> [11] Anthony and Bartlett, Neural Network Learning: Theoretical Foundations, Cambridge University Press
>
> [12] Liu et al., Darts: Differentiable architecture search, ICLR’19
>
> [13] Balcan et al. Provably Tuning Elastic Across Instance, NeurIPS’22
>
> [14] Bartlett, P., Maiorov, V., Meir, R. (1998). Almost linear VC dimension bounds for piecewise polynomial networks. NeurIPS'11
>
> [15] Balcan et al. New bounds for hyperparameter tuning of regression problems across instances, Neurips’23
>
> [16] Balcan et al. Learning to branch, ICML’18
>
> [17] Balcan and Sharma, Data-driven Semi-supervised Learning, NeurIPS’22
>
> [18] Suggala, Netrapalli, Online non-convex learning: Following the perturbed leader is optimal, ALT’20

---

> ### Author Response · Authors · 2024-11-27
> **Follow-up on Reviewer 4928**
>
> __We are reaching out to follow up on our response and to check if you have any further questions__. In our rebuttal, we addressed your concerns regarding the data-driven settings, its benefits, as well as clarifying the scope of our work. We also discussed/fixed your comments on the proofs, and presentation, as well as other minor issues. We hope that this resolves the weaknesses outlined in your review and __would appreciate a prompt response for confirmation, or any additional questions, clarifications__. Thank you again for your thoughtful feedback.

---

> ### Comment · Reviewer_4928 · 2024-11-28
>
> Dear authors, I appreciate that you took into account my concerns about the presentation and positioning. Even if the setting seems a bit far from practical hyperparameter optimization, I think such theoretical works should be emphasized. Especially nowadays, when most of the focus is captured by intensive empirical works.
>
> Hence, I increase my rating to 6. I can not do more because I am not proficient enough in learning theory and was not able to check the proofs appropriately, though I am reassured by the review of the reviewer GxVs.

---

> > ### Author Response · Authors · 2024-11-29
> > **Response to Reviewer 4928**
> >
> > We thank the reviewer for understanding and positively revaluing our work. We are glad our response resolved the reviewer's concerns. Happy Thanksgiving!

---

### Author Response · Authors · 2024-11-23
**General response**

We thank all the reviewers for their constructive feedback. We are glad that the __reviewers consider our work novel, challenging, and impressive__ (reviewer 4928), and __solid__ (reviewers GxVs, w2AP), and __appreciate our theoretical contributions__ (reviewer GxVs, w2AP).

A main concern of the reviewers is the positioning/scope of our paper (reviewers 4928, GxVs, w2Ap). As suggested by reviewers 4928, GxVs, __we made the following major changes in our paper__":

   1. __Title change__: Sample complexity of data-driven tuning model hyperparameters in neural networks with piecewise polynomial dual functions, to emphasize that (1) we focus on analyzing generalization guarantee, (2) we consider tuning model hyperparameter (not applicable with optimization hyperparameter such as learning rate), and (3) we focus on a special case that the dual functions $f_x(\alpha, w)$ admits piecewise polynomial structure, which is inspired by prior work.

   2. __Main body change__: we add discussions in the main body (l.96-101, 190-193) and a detailed discussion in Appendix B to further justify the technical challenges we have to overcome.

  3. __Challenge and novelty of our contribution__: We also clarify the novelty and challenge of our main contributions (Lemma 4.2, Theorem 5.2) in l.147-156 (with a detailed discussion in Appendix B).

We hope that the changes make the positioning of our paper more clear and highlight our contribution. __We kindly request the reviewers to reevaluate in light of our rebuttal.__

---

### Meta-Review · Area_Chair_FuB2 · 2024-12-22

**Metareview:**

The paper presents a framework for data driven hyper-parameter tuning and establishes sample complexity results for two specific settings of the framework. The overall exposition is clear, and the assumptions and technical results are clearly presented. The reviewers mostly liked the advance and acknowledged the motivation behind the work. There were several concerns raised by the reviewers including alignment with real hyper-parameter settings, the focus on one single parameter which is arguably unrealistic, no computationally efficient algorithm for the proposed framework, and related optimization challenges. The authors have addressed some of these concerns, and clarified and improved aspects of their exposition, which have been acknowledged by the reviewers. Some aspects did stay unresolved and addressing these will strengthen the paper, e.g., do the techniques and results extend to k hyperparameters, do we get exponential dependence on k; can we characterize the number of regions in realistic neural networks, possibly both worst case and average case analysis, etc.

There was an additional thread of discussion on empirical evaluation, where the authors and a reviewer had different perspectives. While empirical evaluations should be included if helps the storyline of a paper, the AC does not think it should be necessity for every ML paper and should not be a necessary criterion for acceptance.

**Additional Comments On Reviewer Discussion:**

The reviewers engaged with the authors during the discussion phase. The discussions led to both increase in scores as well as vastly differing perspectives on certain aspects.

---

### Decision · Program_Chairs · 2025-01-22

Reject